# A Theoretical Analysis of Mamba's Training Dynamics: Filtering Relevant Features for Generalization in State Space Models

**Mugunthan Shandirasegaran**
New Jersey Institute of Technology
ms3537@njit.edu

**Hongkang Li**
University of Pennsylvania
lihk@seas.upenn.edu

**Songyang Zhang**
University of Louisiana at Lafayette
songyang.zhang@louisiana.edu

**Meng Wang**
Rensselaer Polytechnic Institute
wangm7@rpi.edu

**Shuai Zhang**
New Jersey Institute of Technology
sz457@njit.edu

## Abstract

The recent empirical success of Mamba and other selective state space models (SSMs) has renewed interest in non-attention architectures for sequence modeling, yet their theoretical foundations remain underexplored. We present a first-step analysis of generalization and learning dynamics for a simplified but representative Mamba block: a single-layer, single-head selective SSM with input-dependent gating, followed by a two-layer MLP trained via gradient descent (GD). Our study adopts a structured data model with tokens that include both class-relevant and class-irrelevant patterns under token-level noise and examines two canonical regimes: majority-voting and locality-structured data sequences. We prove that the model achieves guaranteed generalization by establishing non-asymptotic sample complexity and convergence rate bounds, which improve as the effective signal increases and the noise decreases. Furthermore, we show that the gating vector aligns with class-relevant features while ignoring irrelevant ones, thereby formalizing a feature-selection role similar to attention but realized through selective recurrence. Numerical experiments on synthetic data justify our theoretical results. Overall, our results provide principled insight into when and why Mamba-style selective SSMs learn efficiently, offering a theoretical counterpoint to Transformer-centric explanations.

## 1 Introduction

Transformers (Vaswani et al., 2017) have become the mainstream framework in large language models (Achiam et al., 2023; Guo et al., 2025; Brown et al., 2020; Touvron et al., 2023). However, due to the quadratic time and memory complexity introduced by the attention mechanism with respect to input length (Gu & Dao, 2023; Dao & Gu, 2024), Transformers are inefficient when handling long input sequences. Recently, State Space Models (SSMs) (Gu & Dao, 2023; Dao & Gu, 2024; Zhu et al., 2024; Wang et al., 2024a; Behrouz & Hashemi, 2024; Liu et al., 2024; Wang et al., 2024b) have shown competitive or superior performance to Transformers across domains such as language (Gu & Dao, 2023), vision (Zhu et al., 2024; Liu et al., 2024), graphs (Wang et al., 2024a; Behrouz & Hashemi, 2024), audio (Yadav & Tan, 2024), and reinforcement learning (Lu et al., 2023). SSMs have brought many advantages absent in Transformer-based models, such as linear computational complexity and hardware-friendly properties that enable efficient parallelization. Among these models, Mamba (Gu & Dao, 2023) proposes a selection mechanism, which parameterizes the SSM with the input, which allows the model to dynamically retain or discard relevant and irrelevant infor-

mation. This enables the Mamba model to achieve performance comparable to Transformer-based models in long-text modeling as well as tasks such as visual classification and dense prediction (Zhu et al., 2024; Liu et al., 2024), but in a more efficient manner.

Although recent work has primarily focused on the empirical performance of Mamba and its architectural comparisons with other models, the theoretical understanding of Mamba remains less investigated. In addition, recent empirical evidence shows that Mamba's success is highly sensitive to hyperparameter tuning (Okpekpe & Orvieto, 2025). Such dependence on fragile optimization choices raises fundamental questions about why and when Mamba succeeds. These include fundamental inquiries such as:

- *Under what conditions can a Mamba be trained to achieve satisfactory generalization?*

- *How is the selection mechanism implemented through Mamba's components?*

Existing theoretical studies on Mamba or related SSMs mainly focus on the expressive power and the mechanisms of optimal parameters. Orvieto et al. (2024) and Nishikawa & Suzuki (2025) prove SSMs augmented with MLPs are universal approximators of regular functionals and can mimic token selection dynamically. Muca Cirone et al. (2024) and Huang et al. (2025) show that Mamba has stronger expressive power than its diagonal SSM predecessor, especially in approximating discontinuous functions. Li et al. (2024c) and Li et al. (2025c) respectively prove that two simplified SSMs, H3 and GLA, implicitly perform weighted preconditioned GD at the global minima of in-context learning problems when input with context examples. However, these works do not explain whether the selection mechanisms and advantages of Mamba can actually be obtained through practical training. Moreover, these studies do not analyze the generalization ability of Mamba models.

**Contributions of this paper.** In this work, we study a nonlinear neural model composed of a one-layer Mamba block and a two-layer perceptron, which is simplified but sufficiently representative to reflect the gating structure in Mamba. By assuming the presence of the class-relevant features that influence the label and class-irrelevant features that do not, we respectively formulate majority-voting and locality-structured data, whose labels depend on the proportion and the spatial/temporal locality of a certain class-relevant feature in the data. To the best of our knowledge, this work provides the first theoretical analysis of Mamba's training dynamics with input-dependent gating, together with generalization guarantees under the two structured data regimes. The highlights of our technical contributions include:

**First,** we develop a general theoretical framework for analyzing gated architectures trained with gradient descent on structured data. Our analysis explains how the selection mechanism within Mamba interacts with data structure to enable efficient learning and guaranteed generalization, complementing prior results that focus mainly on attention-based models.

**Second,** we provide a theoretical characterization of the gating mechanism in Mamba. We show that the gating parameter vector is trained to amplify class-relevant features while ignoring class-irrelevant ones, thereby formalizing the intuition that the gating network dynamically allocates capacity to informative patterns.

**Third,** we establish the sample complexity and the required number of iterations for two canonical data types: majority-voting and locality-structured data sequences. For majority-voting data, these bounds scale with the gap between the class-relevant and confusion features; for locality-structured data, they depend on the concentration of class-relevant tokens. In both regimes, stronger signal and lower token-level noise yield faster convergence and better generalization.

## 1.1 RELATED WORK

**State Space Models (SSMs).** Building upon the early S4 models (Gu et al., 2021; Gupta et al., 2022; Smith et al., 2023), Mamba (Gu & Dao, 2023; Dao & Gu, 2024) introduced input-dependent gating to dynamically select relevant features, achieving remarkable performance in NLP and CV. Recent works extending SSMs beyond 1D sequences have highlighted the importance of input ordering and scanning. For example, VMamba (Liu et al., 2024) introduces SS2D, employing multiple scanning routes to bridge sequential structure with the non-sequential nature of vision inputs, while Graph Mamba (Wang et al., 2024a; Behrouz & Hashemi, 2024) adapts SSMs to non-Euclidean domains by leveraging graph connectivity. Collectively, these works show that the effectiveness of

SSMs is tightly linked to input ordering and scanning strategies, a challenge that also motivates our theoretical analysis.

**Theoretical Analysis of SSMs.** Theoretical understanding of Mamba is still in its early stages and has so far centered primarily on approximation theory, such as connections to attention-like mechanisms (Dao & Gu, 2024; Nishikawa & Suzuki, 2025), expressive capacity (Cohen-Karlik et al., 2025; Huang et al., 2025; Muca Cirone et al., 2024; Bao et al., 2025), long-range dependency modeling (Ma & Najarian, 2025; Yu & Erichson, 2025), and the comparison with Transformers (Jelassi et al., 2024). Beyond approximation theory, several recent works have begun examining optimization and generalization aspects of SSMs. Honarpisheh et al. (2025) provide a generalization-error bound based on Rademacher complexity; Slutzky et al. (2024) study implicit bias under a teacher–student setting and show that gradient flow can converge to a low-rank solution, though their model does not incorporate Mamba's input-dependent gating. These analyses provide valuable intuition about the representational strengths and weaknesses of Mamba blocks. However, such results remain largely structural: they establish only the existence of desirable representations, without explaining whether or how these capabilities arise during training, particularly under Mamba's unique mechanism. Motivated by this gap, we focus on studying how Mamba interacts with structured data, with particular emphasis on the role of its gating mechanism in shaping training dynamics and generalization.

**Feature Learning Framework.** Recent theoretical studies of deep learning have shifted focus from the NTK framework (Jacot et al., 2018; Allen-Zhu et al., 2019b; Arora et al., 2019) to the feature-learning framework, where data is modeled as a combination of features and the central question is how neural networks align with these features. Much of the recent work has concentrated on transformers (Li et al., 2023a; 2024b; 2023b; 2025b;a; Liao et al., 2025), feedforward neural networks (Bakshi et al., 2019; Arora et al., 2019; Wen & Li, 2021; Sun et al., 2025), graph neural networks (Zhang et al., 2023; Li et al., 2024a), and Mixture-of-Experts Chowdhury et al. (2023). Due to the inherent complexity of non-convex optimization and modern architectures, prior works on feature learning have, to the best of our knowledge, focused primarily on shallow networks. In this work, we extend the structural data model to analyze the training dynamics of a shallow yet representative Mamba block, with particular emphasis on how its data-dependent gating mechanism shapes learning and generalization.

## 2 PRELIMINARIES

**Structured state space models (S4).** For the $t$-th token, e.g., at time step $t$, let $\boldsymbol{x}_t \in \mathbb{R}^d$ be the input, $\boldsymbol{H}_t \in \mathbb{R}^{N \times d}$ denote the corresponding hidden state, and $\boldsymbol{y}_t \in \mathbb{R}^d$ denote the output. Let $\boldsymbol{A} \in \mathbb{R}^{N \times N}$ and $\boldsymbol{b}, \boldsymbol{c} \in \mathbb{R}^N$ be model parameters. The discrete-time SSM is given by

$$\boldsymbol{H}_t = \overline{\boldsymbol{A}}\boldsymbol{H}_{t-1} + \overline{\boldsymbol{b}}\,\boldsymbol{x}_t^\top, \qquad y_t = \boldsymbol{H}_t^\top \boldsymbol{c}, \tag{1}$$

where $\overline{\boldsymbol{A}} = \exp(\Delta \boldsymbol{A})$ and $\overline{\boldsymbol{b}} = \boldsymbol{A}^{-1}(\exp(\Delta \boldsymbol{A}) - \boldsymbol{I})\,\boldsymbol{b}$ with $\Delta > 0$ as the sampling step.

**Mamba.** To overcome the data-independence of S4, recent work introduced *selective state space models* (Gu & Dao, 2023), where key parameters are made input-dependent. Concretely, given input tokens $\boldsymbol{x}_t \in \mathbb{R}^d$, the recurrence parameters are defined as

$$\boldsymbol{b}_t = \boldsymbol{W}_B^\top \boldsymbol{x}_t, \qquad \Delta_t = \log\left(1 + e^{\boldsymbol{w}_\Delta^\top \boldsymbol{x}_t}\right), \qquad \boldsymbol{c}_t = \boldsymbol{W}_C^\top \boldsymbol{x}_t, \tag{2}$$

with learnable projections $\boldsymbol{W}_B, \boldsymbol{W}_C \in \mathbb{R}^{d \times N}$ and a gating vector $\boldsymbol{w}_\Delta \in \mathbb{R}^d$. The discretization then yields two input-dependent gates,

$$\overline{\boldsymbol{b}}_t = \sigma(\boldsymbol{w}_\Delta^\top \boldsymbol{x}_t)\,\boldsymbol{b}_t, \qquad \bar{a}_t = 1 - \sigma(\boldsymbol{w}_\Delta^\top \boldsymbol{x}_t), \tag{3}$$

which respectively control the input update and the carry-over of past states. With hidden state $\boldsymbol{H}_t \in \mathbb{R}^{N \times d}$, the recurrence becomes

$$\boldsymbol{H}_t = \bar{a}_t \boldsymbol{H}_{t-1} + \overline{\boldsymbol{b}}_t \boldsymbol{x}_t^\top. \tag{4}$$

Mamba output at token $t$ is given by:

$$\boldsymbol{y}_t(\boldsymbol{X}) := \boldsymbol{H}_t^\top \boldsymbol{c}_t = \sigma(\boldsymbol{w}_\Delta^\top \boldsymbol{x}_t)\,(\boldsymbol{W}_B^\top \boldsymbol{x}_t)^\top (\boldsymbol{W}_C^\top \boldsymbol{x}_t)\,\boldsymbol{x}_t + \left(1 - \sigma(\boldsymbol{w}_\Delta^\top \boldsymbol{x}_t)\right)\boldsymbol{H}_{t-1}^\top \boldsymbol{c}_t$$

$$= \sum_{s=1}^t \left(\prod_{j=s+1}^t \left(1 - \sigma(\boldsymbol{w}_\Delta^\top \boldsymbol{x}_j)\right)\right) \cdot \sigma(\boldsymbol{w}_\Delta^\top \boldsymbol{x}_s)\,(\boldsymbol{W}_B^\top \boldsymbol{x}_s)^\top (\boldsymbol{W}_C^\top \boldsymbol{x}_t)\,\boldsymbol{x}_s. \tag{5}$$

**Connection and Difference with Transformer.** The Mamba formulation reveals a natural analogy to attention mechanisms (Dao & Gu, 2024; Sieber et al., 2024). In particular, the input-dependent matrices $\boldsymbol{W}_C$ and $\boldsymbol{W}_B$ can be interpreted as counterparts to queries and keys in the self-attention, while the gating term $\sigma(\boldsymbol{w}_\Delta^\top \boldsymbol{x}_t)$ acts as a dynamic weight controlling how past information contributes to the current output (Dao & Gu, 2024). This structure yields a formulation closely related to gated linear attention (Yang et al., 2024; Li et al., 2025c; Lu et al., 2025), thereby highlighting a connection between SSM and Transformer models. Meanwhile, Mamba departs from these architectures: its gating mechanism is defined through *multiplicative interactions*, effectively involving products of successive terms. This nonlinearity makes the analysis of Mamba substantially different and more challenging than that of gated linear attention. Unlike additive attention-style weighting, Mamba's gating introduces input-dependent multiplicative modulation in the selection mechanism. This alters how information is propagated through the model and results in training dynamics that differ from attention-based architectures.

## 3 PROBLEM FORMULATION

Following existing works (Brutzkus & Globerson, 2021; Zhang et al., 2023; Li et al., 2023a), we consider a binary classification problem with training data $\{(\boldsymbol{X}^{(n)}, z^{(n)})\}_{n=1}^N$ sampled i.i.d. from an unknown distribution $\mathcal{D}$, where $z^{(n)} \in \{+1, -1\}$ is the label. The goal is to learn a model that maps $\boldsymbol{X}$ to $z$ for any $(\boldsymbol{X}, z) \sim \mathcal{D}$. Each input takes the form $\boldsymbol{X}^{(n)} = [\boldsymbol{x}_1^{(n)}, \ldots, \boldsymbol{x}_L^{(n)}] \in \mathbb{R}^{d \times L}$ with $L$ tokens, where each token is $d$-dimensional. Tokens can be image patches (Dosovitskiy et al., 2021; Touvron et al., 2021) or subwords (Sennrich et al., 2016; Kudo & Richardson, 2018).

Learning is performed using a simplified Mamba block formulated by (5), followed by a two-layer MLP. Formally, the model output can be expressed as

$$F(\boldsymbol{X}) = \frac{1}{L} \sum_{l=1}^L \sum_{i=1}^m v_i \phi\left(\boldsymbol{W}_{O(i,\cdot)} \boldsymbol{y}_l(\boldsymbol{X})\right), \tag{6}$$

where $\phi(\cdot)$ denotes the ReLU function, and $\boldsymbol{W}_O \in \mathbb{R}^{m \times d}$, with $\boldsymbol{W}_{O(i,\cdot)}$ being the $i$-th row of $\boldsymbol{W}_O$. Here, $\boldsymbol{y}_l(\boldsymbol{X})$ corresponds to the $l$-th token output of Mamba, as defined in (5). In addition, $v_i$ represents the output-layer weight for the $i$-th hidden unit.

**Model Training**. Let $\boldsymbol{\Psi} = (\boldsymbol{v}, \boldsymbol{W}_O, \boldsymbol{w}_\Delta, \boldsymbol{W}_B, \boldsymbol{W}_C)$ denote the set of model parameters. The training process is to minimize the empirical risk $f_N(\boldsymbol{\Psi})$,

$$\min_{\boldsymbol{\Psi}} \ f_N(\boldsymbol{\Psi}) = \frac{1}{N} \sum_{n=1}^N \ell(\boldsymbol{X}^{(n)}, z^{(n)}; \boldsymbol{\Psi}), \tag{7}$$

where $\ell(\boldsymbol{X}^{(n)}, z^{(n)}; \boldsymbol{\Psi})$ is the hinge loss function, i.e.,

$$\ell(\boldsymbol{X}^{(n)}, z^{(n)}; \boldsymbol{\Psi}) = \max\{0, 1 - z^{(n)} \cdot F(\boldsymbol{X}^{(n)})\}. \tag{8}$$

The empirical risk minimization problem in (7) is solved via gradient descent (GD). For the theoretical analysis, we consider the full batch gradient update with a learning rate of $\eta$ at each iteration $t = 1, 2, \ldots, T$. Each entry of $\boldsymbol{W}_O \in \mathbb{R}^{m \times d}$ is independently initialized from $\mathcal{N}(0, c_0^2)$, and $\boldsymbol{w}_\Delta$ is initialized to 0. Similarly, each entry of $\boldsymbol{v} \in \mathbb{R}^m$ is independently sampled from $\{+\frac{1}{\sqrt{m}}, -\frac{1}{\sqrt{m}}\}$ with equal probability. $\boldsymbol{v}$ is fixed during training, as in other theoretical works (Allen-Zhu & Li, 2022; Arora et al., 2019; Karp et al., 2021; Allen-Zhu et al., 2019a; Li et al., 2023a; 2024b).

**Generalization**. The generalization error of the learned model $\boldsymbol{\Psi}$ is evaluated using the population risk $f(\boldsymbol{\Psi})$, defined as

$$f(\boldsymbol{\Psi}) = f(\boldsymbol{v}, \boldsymbol{W}_O, \boldsymbol{w}_\Delta, \boldsymbol{W}_B, \boldsymbol{W}_C) = \mathbb{E}_{(\boldsymbol{X}, z) \sim \mathcal{D}} \ell(\boldsymbol{X}, z). \tag{9}$$

## 4 THEORETICAL RESULTS

### 4.1 KEY TAKEAWAYS AND INSIGHTS OF THE FINDINGS

Before formally presenting our data assumptions and theoretical results, we first summarize key insights derived from our theoretical findings. We consider a data model where tokens are noisy

Table 1: Some important notations

| $\boldsymbol{y}_l$ | Mamba block output at token position $l$ | $N$ | Number of samples in a batch |
|---|---|---|---|
| $d$ | Embedding dimension | $m$ | The number of neurons in $\boldsymbol{W}_O$ |
| $\eta$ | Learning rate for gradient descent | $L$ | Length of the squence |
| $\Delta L_{\boldsymbol{o}_+}^+$ | Concentration of class-relevant tokens | $\alpha_r$ | Average fraction of class-relevant tokens |
| $\Delta L_{\boldsymbol{o}_+}^-$ | Dispersion of the confusion tokens | $\alpha_c$ | Average fraction of confusion tokens |

versions of *class-relevant* patterns that determine the data label and *class-irrelevant* patterns that do not affect the label. Some important parameters are summarized in Table 1.

**(T1). Convergence and sample complexity analysis of GD to achieve guaranteed generalization.** We introduce a theoretical framework for analyzing gated architectures with structured data. Compared with existing results on attention-based models, our framework captures the role of the gating mechanism inside the Mamba block and structured weight interactions, explaining how gradient descent (GD) exploits data structure to improve learning efficiency. Based on this analysis, we show that a model trained with GD achieves guaranteed generalization with high probability over the randomness of the data and the GD updates.

**(T2). Theoretical characterization of the gating mechanism in Mamba.** We prove that during training, the gating network learns to prioritize class-relevant features while ignoring irrelevant ones. In the majority-voting regime, the gating vector $\boldsymbol{w}_\Delta$ becomes increasingly aligned with class-relevant directions: gradients along those directions grow, while those along irrelevant features remain negligible. In the locality-structured data regime, learning emphasizes the elimination of irrelevant features. Their directions are consistently pushed downward by negative updates, while the directions of relevant features remain nearly unchanged. This occurs because class-relevant and confusion tokens appear in equal proportion, so the model cannot amplify the former and instead reduces the influence of the latter. These dynamics strengthen informative tokens and weaken uninformative ones, inducing effective sparsity in the activations and formalizing the intuition that Mamba allocates capacity to the most important patterns in the data.

**(T3). A larger fraction or a higher local concentration of class-relevant features accelerates learning.** We show that both the number of iterations and the sample complexity required for generalization depend on the discriminative structure of the data and the token-level noise $\tau$. For majority-voting data, these quantities scale as $(\alpha_r - \alpha_c)^{-2}$, so learning is faster when the fraction of class-relevant tokens is larger. For locality-structured data, the number of iterations scales as $\left[\left(\frac{1}{2}\right)^{\Delta L_{\boldsymbol{o}_+}^+} - \left(\frac{1}{2}\right)^{\Delta L_{\boldsymbol{o}_+}^-}\right]^{-1}$, while the sample complexity scales as $\left[\left(\frac{1}{2}\right)^{\Delta L_{\boldsymbol{o}_+}^+} - \left(\frac{1}{2}\right)^{\Delta L_{\boldsymbol{o}_+}^-}\right]^{-2}$. Here, $\Delta L_{\boldsymbol{o}_+}^+$ denotes the separation between class-relevant features $\boldsymbol{o}_+$ in positive samples (capturing their locality), and $\Delta L_{\boldsymbol{o}_+}^-$ denotes the separation between confusion features $\boldsymbol{o}_+$ in negative samples (capturing the locality of confusing patterns). Thus, when $\Delta L_{\boldsymbol{o}_+}^+ \gg \Delta L_{\boldsymbol{o}_+}^-$, the locality of class-relevant features dominates, which reduces both the number of iterations and the sample complexity needed for convergence, implying faster learning when class-relevant tokens are more concentrated locally. Finally, in both regimes, smaller token-level noise $\tau$ further accelerates learning.

## 4.2 DATA MODEL

Consider an arbitrary set of orthonormal vectors $\mathcal{O} = \{\boldsymbol{o}_+, \boldsymbol{o}_-, \boldsymbol{o}_3, \ldots, \boldsymbol{o}_d\}$ in $\mathbb{R}^d$, where $\boldsymbol{o}_+$ and $\boldsymbol{o}_-$ are discriminative features and the remaining vectors $\boldsymbol{o}_j, j \geq 3$, are class-irrelevant (filler) features. Depending on the class label, either $\boldsymbol{o}_+$ or $\boldsymbol{o}_-$ serves as the class-relevant pattern, while the other acts as a confusion pattern. Each token $\boldsymbol{x}_l^{(n)}$ in $\boldsymbol{X}^{(n)}$ is a noisy version of one of the input patterns (features), i.e., $\boldsymbol{x}_l^{(n)} = \boldsymbol{o} + \boldsymbol{\xi}$, where $\boldsymbol{o} \in \mathcal{O}$ and $\boldsymbol{\xi}$ is the Gaussian noise. We consider two different data types: majority-voting and locality-structured data.

**Majority Voting Data.** For the majority voting data type, the label is determined by a majority vote over the class-relevant patterns. Let $\alpha_r$ and $\alpha_c$ denote the average fractions of class-relevant tokens and confusion tokens over the distribution $\mathcal{D}$, respectively. In positive samples, noisy variants of $\boldsymbol{o}_+$

are class-relevant, while noisy variants of $\boldsymbol{o}_-$ act as confusion tokens. In negative samples, the roles are reversed. All other tokens correspond to class-irrelevant features.

**Locality-structured Data.** For the locality-structured data type, each sequence contains two $\boldsymbol{o}_+$ tokens and two $\boldsymbol{o}_-$ tokens, while all other tokens correspond to class-irrelevant features. In positive samples, the two $\boldsymbol{o}_+$ tokens are close to each other, while the two $\boldsymbol{o}_-$ tokens are far apart; formally, $\Delta L_{\boldsymbol{o}_+}^+ \ll \Delta L_{\boldsymbol{o}_-}^+$, where $\Delta L_{\boldsymbol{o}_+}^+$ and $\Delta L_{\boldsymbol{o}_-}^+$ denote the distances between the two $\boldsymbol{o}_+$ and $\boldsymbol{o}_-$ tokens, respectively. In negative samples, the pattern is reversed: $\Delta L_{\boldsymbol{o}_-}^- \ll \Delta L_{\boldsymbol{o}_+}^-$.

In addition, we consider a **balanced dataset** sampled from the unknown distribution $\mathcal{D}$. Let $\mathcal{N}_+ = \{(\boldsymbol{X}^{(n)}, z^{(n)}) : z^{(n)} = +1,\, n \in [N]\}$ and $\mathcal{N}_- = \{(\boldsymbol{X}^{(n)}, z^{(n)}) : z^{(n)} = -1,\, n \in [N]\}$ denote the sets of positively and negatively labeled samples, respectively. Then the class balance satisfies $\big|\, |\mathcal{N}_+| - |\mathcal{N}_-| \,\big| = O(\sqrt{N})$.

**Interpreting the Data Model in Practice.** Our theoretical data models are motivated by common patterns observed in practical machine learning tasks.

On the one hand, the **majority-voting** data model captures a widely adopted assumption (Li et al., 2023a; 2024b) in theoretical analysis, whereby the label is determined by the aggregate contribution through majority vote. For example, in image classification tasks (Krizhevsky et al., 2012; Simonyan & Zisserman, 2014; He et al., 2016), the class label is often driven by multiple discriminative patches corresponding to foreground objects (class-relevant tokens). In contrast, background patches may contain other objects or patterns that are not associated with the target class (confusing tokens), along with random patches that are entirely unrelated (class-irrelevant tokens) (Dosovitskiy et al., 2021; Touvron et al., 2021).

On the other hand, the **locality-structured** data corresponds to tasks where semantic meaning is concentrated in spatially or temporally localized clusters, while background features are more dispersed. This structure is most familiar in vision tasks such as object detection and localization (Ren et al., 2015; Carion et al., 2020; Zhou et al., 2016) and image captioning (Vinyals et al., 2016; Xu et al., 2015; Radford et al., 2021), where the decisive content is often confined to a small region of the image. For example, in an image labeled "dog in a park," the prediction relies primarily on the contiguous region containing the dog rather than on scattered background textures. A similar principle holds in audio and speech recognition (Yadav & Tan, 2024; Gulati et al., 2020), where short phonetic segments capture the information needed to recognize words, and in genomics (Alipanahi et al., 2015; Zhou & Troyanskaya, 2015), where functional elements such as sequence motifs and regulatory regions are localized to short windows of DNA. In these settings, the local structure of nearby tokens strongly correlates with the label.

Together, the majority-voting and locality-structured models offer complementary perspectives on when selective recurrence can most effectively support learning from structured real-world data.

### 4.3 FORMAL THEORETICAL RESULTS

#### 4.3.1 THEORETICAL RESULTS FOR MAJORITY-VOTING DATA

We next present a lemma characterizing how the gating vector aligns with different features under the majority-voting data.

**Lemma 4.1** (Gating Vector Alignment for Majority Voting Data). *With initialization where each entry of $\boldsymbol{W}_O$ is drawn independently from $\mathcal{N}(0, \xi^2)$ and $\boldsymbol{w}_\Delta^{(0)} = 0$. With a sufficient number of training samples and iterations, we have*

$$\left\langle \boldsymbol{w}_\Delta^{(T)}, \boldsymbol{o}_+ \right\rangle \geq \frac{\eta T}{8L^2} \Theta((\alpha_r L - \alpha_c L)^2) \tag{10}$$

$$\left\langle \boldsymbol{w}_\Delta^{(T)}, \boldsymbol{o}_- \right\rangle \geq \frac{\eta T}{8L^2} \Theta((\alpha_r L - \alpha_c L)^2) \tag{11}$$

$$\langle \boldsymbol{w}_\Delta^{(T)}, \boldsymbol{o}_j \rangle \leq \widetilde{\mathcal{O}}\left(1/\mathrm{poly}(d)\right), \quad \forall j \geq 3. \tag{12}$$

Lemma 4.1 establishes that after sufficient training, the gating vector $\boldsymbol{w}_\Delta$ aligns positively with the class-relevant features $\boldsymbol{o}_+$(10) and $\boldsymbol{o}_-$ as shown in (11), while its alignment with irrelevant features

remains strictly negative as shown in (12). In other words, the selection mechanism implicitly acts as a feature selector, amplifying relevant tokens and ignoring irrelevant ones. Lemma 4.1 serves as an informal version of Lemmas B.5 and B.6.

**Remark 1:** With majority voting data, the gating vector aligns with discriminative features, i.e., $\boldsymbol{o}_+$ and $\boldsymbol{o}_-$. As a result, the model's output focuses primarily on these features, giving more weight to tokens that carry discriminative features while reducing the influence of less important tokens. Since the number of class-relevant tokens is greater than the number of confusing ones, e.g., in a positive sample, the tokens containing $\boldsymbol{o}+$ outnumber those containing $\boldsymbol{o}-$, the model can correctly assign the label through this majority effect. Furthermore, as the difference between the counts of class-relevant features and confusing features (i.e., $\alpha_r - \alpha_c$ increases, the gating vector converges much faster. Overall, this gating mechanism allows the model to use its training samples more efficiently because it learns to emphasize the most relevant feature early on and ignore irrelevant features.

We now present the theorem establishing the generalization guarantee for Mamba under the majority-voting data.

**Theorem 1** (Generalization for Majority Voting Data). *Suppose the model width satisfies $m \geq d^2 \log q$ for some constant $q > 0$, and the token noise level is bounded as $\tau < \mathcal{O}\left(\frac{1}{d}\right)$. Then, with probability at least $1 - N^{-d}$, if the number of training samples $N$ satisfies*

$$N \geq \Omega\left(\frac{d}{\eta^2(\alpha_r - \alpha_c)^2}\right),\tag{13}$$

*and the number of iterations $T$ satisfies*

$$T = \Theta\left(\frac{1}{\eta(\alpha_r - \alpha_c)^2}\right),\tag{14}$$

*the resulting model achieves guaranteed generalization, i.e.,*

$$f\big(\boldsymbol{v}^{(0)}, \boldsymbol{W}_O^{(T)}, \boldsymbol{w}_\Delta^{(T)}, \boldsymbol{W}_B^{(0)}, \boldsymbol{W}_C^{(0)}\big) = 0.\tag{15}$$

Theorem 1 establishes the sample complexity, as shown in (13), and the convergence rate, as given in (14), that are required to guarantee desirable generalization when training the model in (6) using GD for the majority-voting data type. In other words, the model achieves good generalization once a sufficient number of samples is available, as specified in (13), and training has proceeded for a sufficient number of iterations, as specified in (14).

**Remark 2:** With majority-voting data, the Mamba architecture can effectively capture the underlying data distribution by first identifying discriminative features through its gating mechanism and then aggregating them via a data-dependent recurrent mechanism. In this sense, Mamba behaves similarly to the Transformer (Li et al., 2023a), suggesting a close connection between the two models despite their architectural differences. According to the results of Lemma 4.1, the model further benefits from a faster convergence rate and reduced sample complexity when the gap between class-relevant and confusing features is larger.

### 4.3.2 THEORETICAL RESULTS FOR LOCALITY-STRUCTURED DATA

We next present a lemma characterizing how the gating vector aligns with different features under the locality-structured data.

**Lemma 4.2** (Gating Vector Alignment for Locality-structured Data). *With initialization where each entry of $\boldsymbol{W}_O$ is drawn independently from $\mathcal{N}(0, \xi^2)$ and $\boldsymbol{w}_\Delta^{(0)} = 0$. With a sufficient number of training samples and iterations, we have*

$$\langle \boldsymbol{w}_\Delta^{(T)}, \boldsymbol{o}_+ \rangle \geq -\widetilde{\mathcal{O}}\left(1/\mathrm{poly}(d)\right),\tag{16}$$

$$\langle \boldsymbol{w}_\Delta^{(T)}, \boldsymbol{o}_- \rangle \geq -\widetilde{\mathcal{O}}\left(1/\mathrm{poly}(d)\right),\tag{17}$$

$$\left\langle \boldsymbol{w}_\Delta^{(T)}, \boldsymbol{o}_j \right\rangle \leq \frac{-\eta T c'^3}{16L}\left[\left(\frac{1}{2}\right)^{\Delta L_{\boldsymbol{o}_+}^+ - 2} - \left(\frac{1}{2}\right)^{\Delta L_{\boldsymbol{o}_+}^- - 2}\right]\left[\left(\frac{1}{2}\right)^{\Delta L_{\boldsymbol{o}_+}^+} + \left(\frac{1}{2}\right)^{\Delta L_{\boldsymbol{o}_-}^-}\right].\tag{18}$$

Lemma 4.2 establishes that after sufficient training, the gating vector $\boldsymbol{w}_\Delta$ remains close to zero for class-relevant features $\boldsymbol{o}_+$ as shown in (16) and $\boldsymbol{o}_-$ as shown in (17), however its alignment with irrelevant features remains strongly negative as shown in (18). Through this mechanism, the gating favors class-relevant features to select the most informative feature for learning. Lemma 4.2 serves as an informal version of Lemmas C.5 and C.6.

**Remark 3:** The gating vector behaves differently from majority voting, though the overall insights remain similar. We can no longer guarantee that $\boldsymbol{w}_\Delta$ will always grow in the direction of discriminative features, because we assume that the number of class-relevant features can be comparable to the number of confusing features. This assumption is introduced to highlight the role of data locality in shaping the gating vector, which is more challenging to analyze in isolation since majority voting can readily reinforce it; however, their combined effect better reflects real-world data. Although this direct growth no longer holds, the gating vector consistently decreases in the direction of irrelevant features. At a higher level, this can be seen as a synergistic interaction: the recurrent mechanism captures locality and suppresses irrelevant features, which pushes the gating vector to decrease along those directions, while the gating itself further amplifies this suppression. From another perspective, by making the model pay less attention to irrelevant features, the gating vector effectively shifts more attention toward discriminative features.

We now present the theorem establishing the generalization guarantee for Mamba under the locality-structured data.

**Theorem 2** (Generalization for Locality-structured Data). *Suppose the model width satisfies $m \geq d^2 \log q$ for some constant $q > 0$, and the token noise level is bounded as $\tau < \mathcal{O}\left(\frac{1}{d}\right)$. Then, with probability at least $1 - N^{-d}$, if the number of training samples $N$ satisfies*

$$N \geq \Omega\left(\frac{L^2 d}{\eta^2 \left[(1/2)^{\Delta L_{o_+}^+} - (1/2)^{\Delta L_{o_+}^-}\right]^2}\right), \tag{19}$$

*and the number of iterations $T$ satisfies*

$$T = \Theta\left(\frac{L^2}{\eta\left[(1/2)^{\Delta L_{o_+}^+} - (1/2)^{\Delta L_{o_+}^-}\right]}\right), \tag{20}$$

*the resulting model achieves guaranteed generalization, i.e.,*

$$f\left(\boldsymbol{v}^{(0)}, \boldsymbol{W}_O^{(T)}, \boldsymbol{w}_\Delta^{(T)}, \boldsymbol{W}_B^{(0)}, \boldsymbol{W}_C^{(0)}\right) = 0. \tag{21}$$

Theorem 2 shows that good generalization on locality-structured data is guaranteed if the sample complexity meets (19) and training proceeds for at least (20) iterations.

**Remark 4:** We establish that Mamba can also effectively learn this type of data through its ability to exploit locality, in contrast to Transformers, where no such guarantee is provided in (Li et al., 2023a). In our analysis, $\Delta L_{o_+}^+$ captures the distance between class-relevant tokens, reflecting the locality of class-relevant features, while $\Delta L_{o_+}^-$ captures the locality of confusing features. The effectiveness of learning is governed by the separation between these two quantities. In particular, when $\Delta L_{o_+}^+ \gg \Delta L_{o_+}^-$, the locality of class-relevant features dominates that of confusing features. That is, when $\Delta L_{o_+}^+ \gg \Delta L_{o_+}^-$, the locality of class-relevant features dominates that of confusing ones, which reduces both the sample complexity and the number of iterations required for convergence, allowing Mamba to learn more effectively and efficiently.

## 4.4 TECHNICAL NOVELTY AND CHALLENGES

**Differences with Existing Works.** Our work is mainly inspired by prior feature-learning analyses of (Bakshi et al., 2019; Arora et al., 2019; Brutzkus & Globerson, 2021; Li et al., 2023a; 2025b). Building on these foundations, we develop a framework specifically tailored to *gated* architectures with structured data. Unlike these existing models, Mamba introduces an input-dependent gating mechanism, absent from other network architectures, which acts as a dynamic selection operator and requires new analytical techniques to capture its learning dynamics. Moreover, while the majority-voting data model has been previously studied in the context of Transformers (Li et al., 2023a), we show that Mamba can also learn this type of data with comparable performance. Furthermore,

we find that Mamba is particularly effective at capturing the inherent locality of the data, which motivates us to introduce a new *locality-structured* data model. For both regimes, we establish generalization guarantees within the framework of selective state space models, thereby advancing our understanding of this class of architectures and clarifying their distinctions from Transformers. A proof sketch can be found in Appendix A.2.

**Technical Challenges.** Our analysis faces several unique technical challenges stemming from the structure of selective SSMs. Unlike attention-based models, where interactions are primarily additive, Mamba's gating mechanism introduces multiplicative recurrences across tokens, with dynamics that are explicitly sensitive to token order. These multiplicative effects accumulate over time, substantially complicating the training analysis. To capture this behavior, we systematically track the gradient updates of the gating vector $\boldsymbol{w}_\Delta$, decomposing the contributions from different token positions and analyzing how token placement influences training dynamics.

Specifically, in the **majority-voting data**, the gradient decomposition of the gating includes off-diagonal terms $\beta_{s,s+1}^{(l)}$ that exhibit additional multiplicative decay due to the recursive gating structure, whereas the diagonal term $\beta_{s,s}^{(l)}$ is independent of token position. Hence, it is important to carefully consider competing token contributions to prove that the gating vector is indeed aligned with class-relevant feature directions.

Instead, in the **locality-structured data**, the variation introduced by the number of class-relevant and confusion tokens in positive and negative samples is negligible, as our data model assumes an equal number of class-relevant and confusion tokens. Consequently, we need to rely on $\Delta L_{\boldsymbol{o}_+}^+$ and $\Delta L_{\boldsymbol{o}_-}^+$ to ensure that the lucky neuron $\boldsymbol{W}_{O(i,\cdot)}$ learns the class-relevant feature. Moreover, since the number of class-relevant and confusion tokens is balanced, updates along the class-relevant feature direction for the gating vector remain close to zero. To demonstrate how the gate filters information, we show that gradient updates along class-irrelevant features are driven strongly negative. To prove this, in addition to the terms considered in the majority-voting setting, we must also bound the positively contributing terms that hinder the gate's ability to suppress irrelevant features. Specifically, we bound these opposing terms as $\mathcal{O}\left((1 - \sigma(p_2))^{\Delta L_{\boldsymbol{o}_+}^-}\right) + \mathcal{O}\left((1 - \sigma(p_2))^{\Delta L_{\boldsymbol{o}_-}^+}\right)$ ensuring that their effect remains minimal. This reveals that the gate effectively suppresses irrelevant features while preserving class-relevant features for this data model.

# 5 NUMERICAL EXPERIMENTS

We verify our theoretical results through synthetic experiments based on the data models described in Section 4.2. Due to the space limit, we defer the experiment details and additional numerical results to Appendix A.3

**Faster convergence with larger majority-voting gap.** Fig. 1 illustrates that increasing the majority-voting gap $\alpha_r - \alpha_c$ consistently reduces the number of epochs across various sizes of training samples. These findings are consistent with our theoretical results in (13) and (14).

**Gating mechanism amplifies relevant features in majority-voting data.** Fig. 2 shows the cosine similarity between the gating vector $\boldsymbol{w}_\Delta$ and both class-relevant and class-irrelevant features. The similarity with class-relevant features steadily increases, while that with class-irrelevant features remains essentially unchanged. This empirically confirms Lemma 4.1, demonstrating that the gate prioritizes informative features while ignoring irrelevant ones.

**Locality affects the learning.** Fig. 3 illustrate the effect of class-relevant token separation $\Delta L$ on the convergence in the locality-structured data. Larger $\Delta$ slows convergence across different training sample sizes, which is consistent with our results in (19) and (20).

**Gating mechanism suppresses irrelevant features.** Fig. 4 illustrates that while the cosine similarity is negative for both types of features, it stays close to zero for class-relevant features but becomes largely negative for class-irrelevant ones. This contrast drives the gating mechanism to prioritize class-relevant features, consistent with Lemma 4.2.

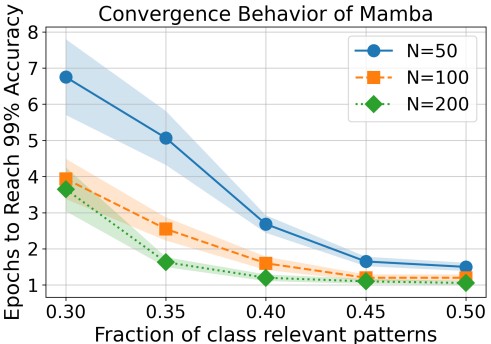

Figure 1: Convergence vs. majority-voting gap.

Figure 2: Alignment of $\boldsymbol{w}_\Delta$ for majority-voting data.

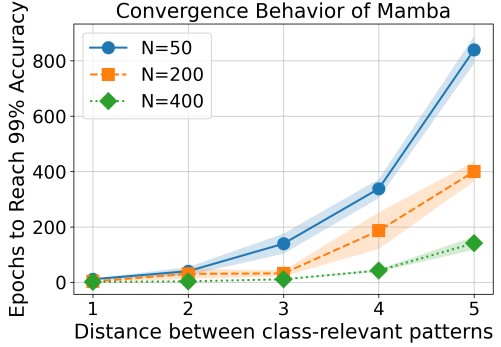

Figure 3: Convergence under locality-structured data.

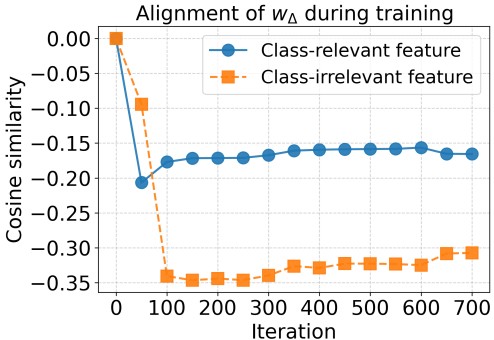

Figure 4: Alignment of $\boldsymbol{w}_\Delta$ for locality-structured data.

## 6 CONCLUSION

Encouraged by the emergence and successful applications of the Transformer alternative architecture Mamba, this paper provides a novel theoretical generalization analysis of Mamba by considering its unique gated selection mechanism. Focusing on a data model with class-relevant and class-irrelevant tokens, we establish the non-asymptotic sample complexity and the convergence rate required to achieve desirable test accuracy. Our analysis further shows that the gating parameter vector filters out the class-relevant features while ignoring irrelevant ones. To the best of our knowledge, this is the first theoretical analysis of Mamba's training dynamics with its input-dependent gating mechanism, together with generalization guarantees.

Finally, we note some limitations of our work. First, our theoretical analysis focuses on a simplified Mamba setting that abstracts away practical components such as depth, multiple heads, residual connections, and layer normalization. Second, our data model, while standard in theoretical studies, also simplifies real-world sequence structures. Extending the analysis to more realistic multi-layer and multi-head Mamba architectures, richer data models, and alternative designs such as gated Transformers or hybrid Mamba–Transformer frameworks remains an important direction for future work.

ACKNOWLEDGMENTS

This work was supported in part by the National Science Foundation (NSF) under Grants #2349879, #2349878, #2425811, and #2430223. Part of Hongkang's work was completed while he was a Ph.D. student at Rensselaer Polytechnic Institute (RPI) and was supported in part by the Army Research Office (ARO) under Grant W911NF-25-1-0020, as well as by the Rensselaer–IBM Future

of Computing Research Collaboration (`http://airc.rpi.edu`). We also thank the anonymous reviewers for their constructive and insightful comments.

LLM USAGE DISCLOSURE

We used large-language models (ChatGPT) to aid in polishing the writing of this paper. For numerical experiments, we employed AI-assisted coding tools (GitHub Copilot and ChatGPT) to support code development.

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

# A  NOTATIONS, PROOF SKETCH AND ADDITIONAL EXPERIMENTS

## A.1  NOTATIONS

### A.1.1  LUCKY NEURON DEFINITION

Let

$$\mathcal{K}_+ = \{i \in [m] : v_i > 0\}, \quad \mathcal{K}_- = \{i \in [m] : v_i < 0\} \tag{22}$$

denote the sets of neurons with positive and negative output layer weights, respectively.

We define the sets of lucky neurons at initialization as:

$$\mathcal{W}(0) = \left\{i \in \mathcal{K}_+ : \boldsymbol{W}_{O(i,\cdot)}(0)\boldsymbol{o}_+ > 0\right\}, \tag{23}$$

$$\mathcal{U}(0) = \left\{i \in \mathcal{K}_- : \boldsymbol{W}_{O(i,\cdot)}(0)\boldsymbol{o}_- > 0\right\}, \tag{24}$$

where $\boldsymbol{o}_+$ and $\boldsymbol{o}_-$ denote the class-relevant features for the positive and negative classes, respectively.

### A.1.2  LOSS FUNCTION

The loss function for the $n^{\text{th}}$ sample is defined as

$$
\begin{aligned}
\ell(\boldsymbol{X}^{(n)}, z^{(n)}) &= \max\{0, 1 - z^{(n)} \cdot F(\boldsymbol{X}^{(n)})\} \\
&= \max\left\{0, 1 - z^{(n)} \cdot \frac{1}{L} \sum_{l=1}^{L} \sum_{i=1}^{m} v_i \phi\left(\boldsymbol{W}_{O(i,\cdot)} \boldsymbol{y}_l^{(n)}\right)\right\}.
\end{aligned}
\tag{25}
$$

The empirical loss is denoted by $\hat{\mathcal{L}}$ and is given by

$$\hat{\mathcal{L}} = \frac{1}{N} \sum_{n=1}^{N} \ell(\boldsymbol{X}^{(n)}, z^{(n)}). \tag{26}$$

The population loss is denoted by $\mathcal{L}$ and is defined as

$$\mathcal{L} = \mathbb{E}_{(\boldsymbol{X},z)\sim\mathcal{D}}\ell(\boldsymbol{X}, z). \tag{27}$$

With additional important notations can be found in Table 2.

## A.2  PROOF SKETCH

The major idea of our proof is to analyze how GD gradually aligns both the hidden-layer weights and gating vector with class-relevant features while ignoring the irrelevant ones. A key tool in our analysis is the notion of a *lucky neuron*, i.e., a hidden layer neuron whose initialization is well aligned with a class-relevant feature. For the majority-voting data model, the signal driving this alignment is proportional to the gap between the fractions of class-relevant and confusion tokens, $\Theta(\alpha_r - \alpha_c)$, as established by Lemmas B.1–B.4. Lucky neurons move consistently toward their class-relevant feature, while the magnitude of unlucky ones remains small (upper-bounded by the inverse square root of the number of samples). For the locality-structured data model, we prove that the update in the class-relevant feature direction for the gating vector remains close to zero because an equal number of class-relevant and confusion tokens are present in the data. We then show that the gating vector consistently decreases along irrelevant feature directions, thereby enabling the gate to effectively select the class-relevant feature.

Due to these properties, the training dynamics can be simplified to show that the network output in (6) changes linearly with the iteration number $t$. In particular, we prove that, for a new positive sample (w.l.o.g.) during inference, the learned model's output is strictly positive. From this analysis, we derive the sample complexity and the required number of iterations for achieving zero generalization error for both data types, as shown in (13) and (14) for the majority-voting setting in Theorem 1, and similarly in (19) and (20) for the locality-structured setting in Theorem 2.

Table 2: Summary of notations

| | |
|---|---|
| $F(\boldsymbol{X}^{(n)})$ | The final model output for $\boldsymbol{X}^{(n)}$ |
| $\alpha_r$ | The average fractions of class-relevant tokens |
| $\alpha_c$ | The average fractions of confusion tokens |
| $\Delta L_{\boldsymbol{o}_+}^+$ | Separation between class-relevant features $\boldsymbol{o}_+$ in positive samples |
| $\Delta L_{\boldsymbol{o}_+}^-$ | Separation between confusion features $\boldsymbol{o}_+$ in negative samples |
| $\Delta L_{\boldsymbol{o}_-}^-$ | Separation between class-relevant features $\boldsymbol{o}_-$ in negative samples |
| $\Delta L_{\boldsymbol{o}_-}^+$ | Separation between confusion features $\boldsymbol{o}_-$ in positive samples |
| $\mathcal{O}$ | The set of class-relevant and class-irrelevant patterns |
| $\mathcal{K}_+$ | The set of lucky neurons with respect to $\boldsymbol{W}^{(0)}$ |
| $\mathcal{K}_-$ | The set of lucky neurons with respect to $\boldsymbol{U}^{(0)}$ |
| $\mathcal{N}$ | The set of training data |
| $\mathcal{N}_+$ | The set of training data with positive labels |
| $\mathcal{N}_-$ | The set of training data with negative labels |
| $\mathcal{W}(t)$ | Set of lucky neurons for the positive class at iteration $t$ |
| $\mathcal{U}(t)$ | Set of lucky neurons for the negative class at iteration $t$ |
| $\mathcal{O}(\cdot), \Omega(\cdot), \Theta(\cdot)$ | We use the standard convention: $f(x) = \mathcal{O}(g(x))$ (resp. $\Omega(g(x))$, $\Theta(g(x))$) means $f(x)$ grows at most (resp. at least, on the order of) $g(x)$. |
| $\widetilde{\mathcal{O}}(\cdot)$ | Soft-$\mathcal{O}$ notation: hides polylog factors |
| $\mathrm{poly}(d)$ | An unspecified polynomial in $d$ |
| $\gtrsim, \lesssim$ | $f(x) \gtrsim g(x)$ (resp. $f(x) \lesssim g(x)$) abbreviates $f(x) \geq \Omega(g(x))$ (resp. $f(x) \leq \mathcal{O}(g(x))$). |

## A.3 ADDITIONAL NUMERICAL EXPERIMENTS

**Experiment settings.**

The data dimension and token embedding size are both set to $d = 32$, which also corresponds to the number of feature directions. Unless otherwise stated, experiments in the main text use exactly the model defined in Eq. (6) to match our theoretical setting. We also use the model without convolution, and keep $\mathbf{W}_B = \mathbf{W}_C = I$ frozen as in Eq. (15). The total number of neurons in the hidden layer $\boldsymbol{W}_O$ is set to $m = 50$. For simplicity, we fix the ratio of different features to be the same across all data. The sequence length is set to $L = 30$.

We run 100 independent trials and consider only the successful trials to compute the mean epochs for convergence for a given fraction of class-relevant patterns. An experiment is deemed successful if the test accuracy reaches $99\%$ and the test loss falls below $10^{-3}$. For this experiment, we fixed the fraction of the confusion tokens at $0.10$ and varied the fraction of class-relevant features.

**MLP weights selectively align with only class-relevant features.** Fig. 5 tracks the average cosine similarity between each neuron $\boldsymbol{W}_{O(i,\cdot)}$ and both class-relevant & class-irrelevant features. The alignment increases for class-relevant features and stays essentially unchanged for irrelevant features, which is consistent with our findings in Lemmas B.1 and B.3 in the Appendix.

**Mamba outperforms Transformer and local attention on locality-structured data.** Intuitively, locality-structured data favors models that exploit local biases. Global attention performs only marginally better than random guessing, whereas both local attention and Mamba learn meaningful patterns, with Mamba achieving the best performance.

**Additional Results on MLP Weight Alignment.**

Figure 7 illustrates the alignment of sampled neurons with the class-relevant feature. We observe that, with a good initialization, a subset of neurons, denoted as lucky neurons, consistently increases in the direction of the class-relevant feature, while another subset, denoted as unlucky neurons, remains almost unchanged, which supports our findings in Lemmas B.3 and C.3.

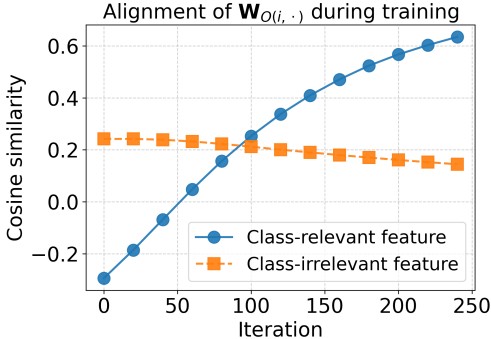
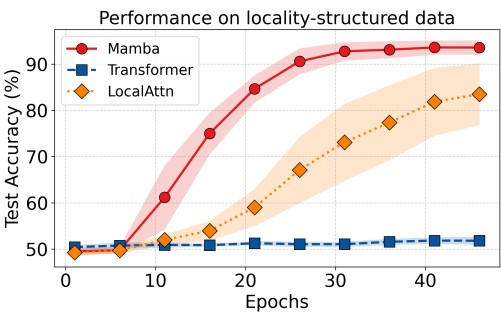

Figure 5: Average alignment of $\mathbf{W}_{O(i,\cdot)}$ during training.

Figure 6: Mamba outperforms on locality data.

In contrast, Figure 8 shows the alignment of sampled neurons with the class-irrelevant feature. In this case, we observe that all neurons, both lucky and unlucky, remain nearly unchanged in the direction of the class-irrelevant feature, which further supports our findings in Lemmas B.4 and C.4.

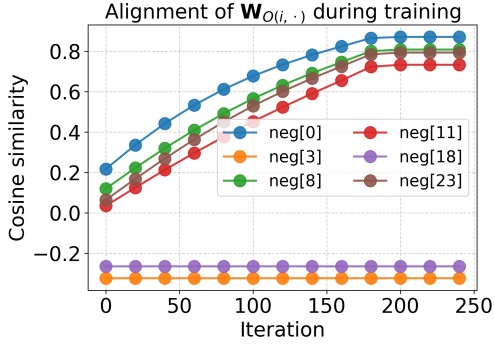
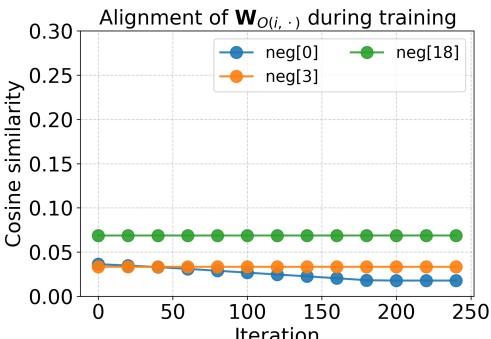

Figure 7: Alignment of $\mathbf{W}_{O(i,\cdot)}$ with class-relevant feature directions during training on the majority-voting data.

Figure 8: Alignment of $\mathbf{W}_{O(i,\cdot)}$ with class-irrelevant feature directions during training on the majority-voting data.

### A.3.1 ADDITIONAL EXPERIMENTS

To further strengthen the empirical connection between our theoretical analysis and practical Mamba architectures, we conducted additional experiments using the multi-layer, multi-head Mamba model from Dao & Gu (2024), trained on synthetic datasets that follow the same structured data models as in our theory.

We first evaluated the Mamba2 block, which includes residual connections and RMSNorm. We focused on a 2-block Mamba model with 4 heads and report the cosine similarity of the learned gating vectors and MLP weights with class-relevant and class-irrelevant features in Figures 9 and 10. For a deeper 5-block Mamba model with the same configuration, we summarize the final alignment values in Table 3, which exhibit the same qualitative trends predicted by our analysis.

Table 3: Cosine similarity alignment in the 5-block Mamba model

| Component | Class-relevant | Class-irrelevant |
|---|---|---|
| Gating vector | 0.53 | 0.00 |
| MLP weights | 0.73 | 0.00 |

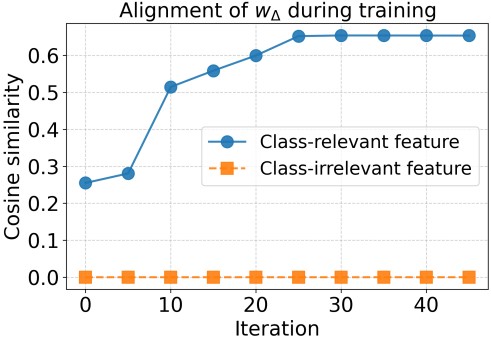 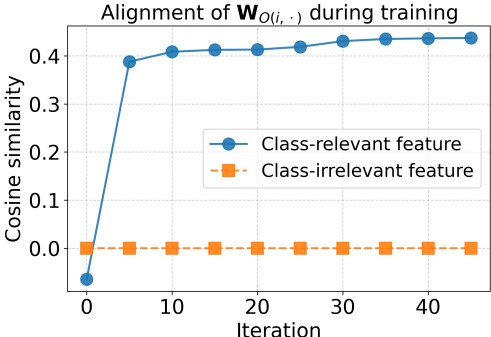

Figure 9: Alignment of the gating vector in the 2-block Mamba model.

Figure 10: Alignment of the MLP weights in the 2-block Mamba model.

Next, we examined the effect of the gating mechanism by comparing models trained with and without gating across both structured data regimes. On the majority-voting data, the gated model consistently outperforms the ungated variant (Figure 11). On the locality-structured data, gating becomes essential: the ungated model fails to learn the task, whereas the gated model converges reliably (Figure 12).

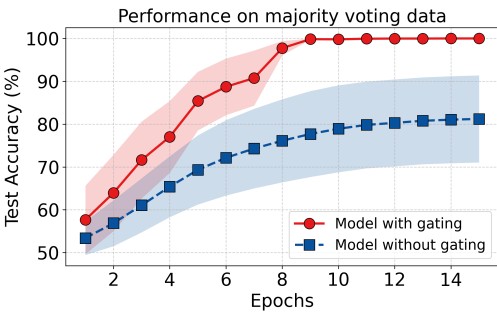 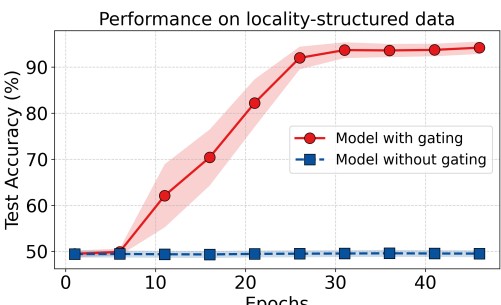

Figure 11: Test accuracy with and without gating on the majority-voting data.

Figure 12: Test accuracy with and without gating on the locality-structured data.

We also conducted two controlled ablations. First, we varied the feature dimension $d \in \{32, 64, 128\}$ and observed that the qualitative behavior of the model remained consistent across all three settings (Figures 13–15). Second, we varied the data distribution parameter $\alpha_c$, the fraction of confusion tokens in the majority-voting data. Across all three choices of $\alpha_c$, the empirical results remained closely aligned with the theoretical predictions (Figures 16–18).

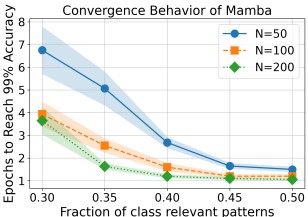 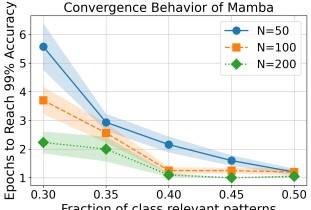

Figure 13: Ablation with feature dimension $d = 32$.

Figure 14: Ablation with feature dimension $d = 64$.

Figure 15: Ablation with feature dimension $d = 128$.

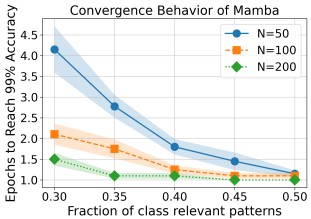 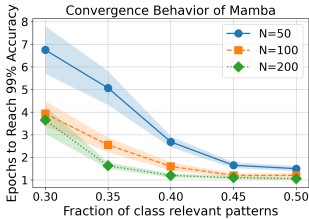 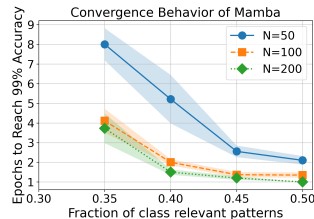

Figure 16: Ablation with confusion fraction $\alpha_c = 0.17$.

Figure 17: Ablation with confusion fraction $\alpha_c = 0.20$.

Figure 18: Ablation with confusion fraction $\alpha_c = 0.23$.

## B    MAJORITY-VOTING DATA

### B.1    USEFUL LEMMAS

Lemma B.1 provides bounds on the gradient updates of lucky neurons $i \in \mathcal{W}(t)$ in the directions of both class-relevant features ($\boldsymbol{o}_+$, $\boldsymbol{o}_-$) and irrelevant features.

**Lemma B.1.** *Suppose $p_1 \leq \langle \boldsymbol{w}_\Delta^{(t)}, \boldsymbol{o}_+ \rangle \leq q_1$ and $p_1 \leq \langle \boldsymbol{w}_\Delta^{(t)}, \boldsymbol{o}_- \rangle \leq q_1$. Then, for any lucky neuron $i \in \mathcal{W}(t)$ at iteration $t$, the following bounds hold:*

*(L1.1) A lower bound on the gradient of $\hat{\mathcal{L}}$ with respect to $\boldsymbol{W}_{O(i,\cdot)}$ at iteration $t$, in the direction of $\boldsymbol{o}_+$, is given by*

$$\left\langle -\frac{\partial \hat{\mathcal{L}}}{\partial \boldsymbol{W}_{O(i,\cdot)}^{(t)}}, \boldsymbol{o}_+ \right\rangle \geq \frac{1}{\sqrt{mL}} \cdot \sigma(p_1)\Theta(\alpha_r L - \alpha_c L) - \mathcal{O}\left(\sqrt{\frac{d \log N}{mN}}\right) - \mathcal{O}(\tau). \qquad (28)$$

*(L1.2) An upper bound on the gradient of $\hat{\mathcal{L}}$ with respect to $\boldsymbol{W}_{O(i,\cdot)}$ at iteration $t$, in the direction of $\boldsymbol{o}_+$, is given by*

$$\left\langle -\frac{\partial \hat{\mathcal{L}}}{\partial \boldsymbol{W}_{O(i,\cdot)}^{(t)}}, \boldsymbol{o}_+ \right\rangle \leq \frac{1}{\sqrt{mL}} \cdot \sigma(q_1)\Theta(\alpha_r L - \alpha_c L) + \mathcal{O}\left(\sqrt{\frac{d \log N}{mN}}\right) + \mathcal{O}(\tau). \qquad (29)$$

*(L1.3) A lower bound on the gradient of $\hat{\mathcal{L}}$ with respect to $\boldsymbol{W}_{O(i,\cdot)}$ at iteration $t$, in the direction of $\boldsymbol{o}_-$, is given by*

$$\left\langle -\frac{\partial \hat{\mathcal{L}}}{\partial \boldsymbol{W}_{O(i,\cdot)}^{(t)}}, \boldsymbol{o}_- \right\rangle \geq -\frac{1}{\sqrt{mL}} \cdot \sigma(q_1)\Theta(\alpha_r L - \alpha_c L) - \mathcal{O}\left(\sqrt{\frac{d \log N}{mN}}\right) - \mathcal{O}(\tau). \qquad (30)$$

*(L1.4) An upper bound on the gradient of $\hat{\mathcal{L}}$ with respect to $\boldsymbol{W}_{O(i,\cdot)}$ at iteration $t$, in the direction of $\boldsymbol{o}_-$, is given by*

$$\left\langle -\frac{\partial \hat{\mathcal{L}}}{\partial \boldsymbol{W}_{O(i,\cdot)}^{(t)}}, \boldsymbol{o}_- \right\rangle \leq \mathcal{O}\left(\sqrt{\frac{d \log N}{mN}}\right) + \mathcal{O}(\tau). \qquad (31)$$

*(L1.5) An upper bound on the gradient of $\hat{\mathcal{L}}$ with respect to $\boldsymbol{W}_{O(i,\cdot)}$ at iteration $t$, in the direction of $\boldsymbol{o}_j$, is given by*

$$\left\langle -\frac{\partial \hat{\mathcal{L}}}{\partial \boldsymbol{W}_{O(i,\cdot)}^{(t)}}, \boldsymbol{o}_j \right\rangle \leq \mathcal{O}\left(\sqrt{\frac{d \log N}{mN}}\right) + \mathcal{O}(\tau), \quad for\ j \neq 1, 2. \qquad (32)$$

Lemma B.2 shows that, for unlucky neurons associated with the positive class, the gradients in the directions of both class-relevant and irrelevant features are small.

**Lemma B.2.** *For any unlucky neuron $i \in \mathcal{K}_+ \setminus \mathcal{W}(t)$ at iteration $t$, the following bounds hold:*

*(L2.1) An upper bound on the gradient of $\hat{\mathcal{L}}$ with respect to $\boldsymbol{W}_{O(i,\cdot)}$ at iteration $t$, in the direction of $\boldsymbol{o}_+$, is given by*

$$\left\langle -\frac{\partial \hat{\mathcal{L}}}{\partial \boldsymbol{W}_{O(i,\cdot)}^{(t)}}, \boldsymbol{o}_+ \right\rangle \leq \mathcal{O}\left( \sqrt{\frac{d \log N}{mN}} \right) + \mathcal{O}(\tau). \tag{33}$$

*(L2.2) An upper bound on the gradient of $\hat{\mathcal{L}}$ with respect to $\boldsymbol{W}_{O(i,\cdot)}$ at iteration $t$, in the direction of $\boldsymbol{o}_-$, is given by*

$$\left\langle -\frac{\partial \hat{\mathcal{L}}}{\partial \boldsymbol{W}_{O(i,\cdot)}^{(t)}}, \boldsymbol{o}_- \right\rangle \leq \mathcal{O}\left( \sqrt{\frac{d \log N}{mN}} \right) + \mathcal{O}(\tau). \tag{34}$$

*(L2.3) An upper bound on the gradient of $\hat{\mathcal{L}}$ with respect to $\boldsymbol{W}_{O(i,\cdot)}$ at iteration $t$, in the direction of $\boldsymbol{o}_j$, is given by*

$$\left\langle -\frac{\partial \hat{\mathcal{L}}}{\partial \boldsymbol{W}_{O(i,\cdot)}^{(t)}}, \boldsymbol{o}_j \right\rangle \leq \mathcal{O}\left( \sqrt{\frac{d \log N}{mN}} \right) + \mathcal{O}(\tau), \quad \text{for } j \neq 1, 2. \tag{35}$$

Lemmas B.3 and B.4, by symmetry, state the analogous results for lucky and unlucky neurons associated with the negative class.

**Lemma B.3.** *Suppose $p_1 \leq \langle \boldsymbol{w}_\Delta^{(t)}, \boldsymbol{o}_+ \rangle \leq q_1$ and $p_1 \leq \langle \boldsymbol{w}_\Delta^{(t)}, \boldsymbol{o}_- \rangle \leq q_1$. Then, for any lucky neuron $i \in \mathcal{U}(t)$ at iteration $t$, the following bounds hold:*

*(L3.1) A lower bound on the gradient of $\hat{\mathcal{L}}$ with respect to $\boldsymbol{W}_{O(i,\cdot)}$ at iteration $t$, in the direction of $\boldsymbol{o}_-$, is given by*

$$\left\langle -\frac{\partial \hat{\mathcal{L}}}{\partial \boldsymbol{W}_{O(i,\cdot)}^{(t)}}, \boldsymbol{o}_- \right\rangle \geq \frac{1}{\sqrt{m}L} \cdot \sigma(p_1)\Theta(\alpha_r L - \alpha_c L) - \mathcal{O}\left( \sqrt{\frac{d \log N}{mN}} \right) - \mathcal{O}(\tau). \tag{36}$$

*(L3.2) An upper bound on the gradient of $\hat{\mathcal{L}}$ with respect to $\boldsymbol{W}_{O(i,\cdot)}$ at iteration $t$, in the direction of $\boldsymbol{o}_-$, is given by*

$$\left\langle -\frac{\partial \hat{\mathcal{L}}}{\partial \boldsymbol{W}_{O(i,\cdot)}^{(t)}}, \boldsymbol{o}_- \right\rangle \leq \frac{1}{\sqrt{m}L} \cdot \sigma(q_1)\Theta(\alpha_r L - \alpha_c L) + \mathcal{O}\left( \sqrt{\frac{d \log N}{mN}} \right) + \mathcal{O}(\tau). \tag{37}$$

*(L3.3) A lower bound on the gradient of $\hat{\mathcal{L}}$ with respect to $\boldsymbol{W}_{O(i,\cdot)}$ at iteration $t$, in the direction of $\boldsymbol{o}_+$, is given by*

$$\left\langle -\frac{\partial \hat{\mathcal{L}}}{\partial \boldsymbol{W}_{O(i,\cdot)}^{(t)}}, \boldsymbol{o}_+ \right\rangle \geq -\frac{1}{\sqrt{m}L} \cdot \sigma(q_1)\Theta(\alpha_r L - \alpha_c L) - \mathcal{O}\left( \sqrt{\frac{d \log N}{mN}} \right) - \mathcal{O}(\tau). \tag{38}$$

*(L3.4) An upper bound on the gradient of $\hat{\mathcal{L}}$ with respect to $\boldsymbol{W}_{O(i,\cdot)}$ at iteration $t$, in the direction of $\boldsymbol{o}_+$, is given by*

$$\left\langle -\frac{\partial \hat{\mathcal{L}}}{\partial \boldsymbol{W}_{O(i,\cdot)}^{(t)}}, \boldsymbol{o}_+ \right\rangle \leq \mathcal{O}\left( \sqrt{\frac{d \log N}{mN}} \right) + \mathcal{O}(\tau). \tag{39}$$

*(L3.5) An upper bound on the gradient of $\hat{\mathcal{L}}$ with respect to $\boldsymbol{W}_{O(i,\cdot)}$ at iteration $t$, in the direction of $\boldsymbol{o}_j$, is given by*

$$\left\langle -\frac{\partial \hat{\mathcal{L}}}{\partial \boldsymbol{W}_{O(i,\cdot)}^{(t)}}, \boldsymbol{o}_j \right\rangle \leq \mathcal{O}\left( \sqrt{\frac{d \log N}{mN}} \right) + \mathcal{O}(\tau), \quad \text{for } j \neq 1, 2. \tag{40}$$

**Lemma B.4.** *For any unlucky neuron $i \in \mathcal{K}_- \setminus \mathcal{U}(t)$ at iteration $t$, the following bounds hold:*

*(L4.1) An upper bound on the gradient of $\hat{\mathcal{L}}$ with respect to $\boldsymbol{W}_{O(i,\cdot)}$ at iteration $t$, in the direction of $\boldsymbol{o}_-$, is given by*

$$\left\langle -\frac{\partial \hat{\mathcal{L}}}{\partial \boldsymbol{W}_{O(i,\cdot)}^{(t)}}, \boldsymbol{o}_- \right\rangle \leq \mathcal{O}\left(\sqrt{\frac{d \log N}{mN}}\right) + \mathcal{O}(\tau). \tag{41}$$

*(L4.2) An upper bound on the gradient of $\hat{\mathcal{L}}$ with respect to $\boldsymbol{W}_{O(i,\cdot)}$ at iteration $t$, in the direction of $\boldsymbol{o}_+$, is given by*

$$\left\langle -\frac{\partial \hat{\mathcal{L}}}{\partial \boldsymbol{W}_{O(i,\cdot)}^{(t)}}, \boldsymbol{o}_+ \right\rangle \leq \mathcal{O}\left(\sqrt{\frac{d \log N}{mN}}\right) + \mathcal{O}(\tau). \tag{42}$$

*(L4.3) An upper bound on the gradient of $\hat{\mathcal{L}}$ with respect to $\boldsymbol{W}_{O(i,\cdot)}$ at iteration $t$, in the direction of $\boldsymbol{o}_j$, is given by*

$$\left\langle -\frac{\partial \hat{\mathcal{L}}}{\partial \boldsymbol{W}_{O(i,\cdot)}^{(t)}}, \boldsymbol{o}_j \right\rangle \leq \mathcal{O}\left(\sqrt{\frac{d \log N}{mN}}\right) + \mathcal{O}(\tau), \quad \text{for } j \neq 1, 2. \tag{43}$$

Lemma B.5 establishes bounds for the gradient updates of $\boldsymbol{w}_\Delta$ in the class-relevant feature directions.

**Lemma B.5.** *Suppose $r_1^* \leq \langle \boldsymbol{W}_{O(i,\cdot)}^{(t+1)^\top}, \boldsymbol{o}_+ \rangle \leq s_1^*$. Let $|\mathcal{W}(t)| = \rho_t^+$ and $|\mathcal{U}(t)| = \rho_t^-$. Then, at iteration $t$, the following bounds hold:*

*(L5.1) A lower bound on the gradient of $\hat{\mathcal{L}}$ with respect to $\boldsymbol{w}_\Delta$ at iteration $t$, in the direction of $\boldsymbol{o}_+$, is given by*

$$\left\langle -\frac{\partial \hat{\mathcal{L}}}{\partial \boldsymbol{w}_\Delta^{(t)}}, \boldsymbol{o}_+ \right\rangle \geq \frac{r_1^*}{2\sqrt{mL}} \cdot \rho_t^+ \cdot \Theta(\alpha_r L) - \frac{\sqrt{m}s_1^*}{4L} \cdot \Theta(\alpha_c L) - \mathcal{O}\left(\sqrt{\frac{d \log N}{mN}}\right) - \mathcal{O}(\tau). \tag{44}$$

*(L5.2) A lower bound on the gradient of $\hat{\mathcal{L}}$ with respect to $\boldsymbol{w}_\Delta$ at iteration $t$, in the direction of $\boldsymbol{o}_-$, is given by*

$$\left\langle -\frac{\partial \hat{\mathcal{L}}}{\partial \boldsymbol{w}_\Delta^{(t)}}, \boldsymbol{o}_- \right\rangle \geq \frac{r_1^*}{2\sqrt{mL}} \cdot \rho_t^- \cdot \Theta(\alpha_r L) - \frac{\sqrt{m}s_1^*}{4L} \cdot \Theta(\alpha_c L) - \mathcal{O}\left(\sqrt{\frac{d \log N}{mN}}\right) - \mathcal{O}(\tau) \tag{45}$$

Lemma B.6 establishes bounds for the gradient updates of $\boldsymbol{w}_\Delta$ in the directions of irrelevant features.

**Lemma B.6.** *An upper bound on the gradient of $\hat{\mathcal{L}}$ with respect to $\boldsymbol{w}_\Delta$ at iteration $t$, in the direction of $\boldsymbol{o}_j$, is given by*

$$\left\langle -\frac{\partial \hat{\mathcal{L}}}{\partial \boldsymbol{w}_\Delta^{(t)}}, \boldsymbol{o}_j \right\rangle \leq \mathcal{O}\left(\sqrt{\frac{d \log N}{mN}}\right) + \mathcal{O}(\tau), \quad \text{for } j \neq 1, 2. \tag{46}$$

### B.2 PROOF OF CONVERGENCE

*Proof of Theorem 1.* The proof starts with the base case at $t = 0$ and proceeds to analyze the training dynamics in a deductive manner, providing additional details in deriving the corresponding convergence and sample complexity bounds.

(**S1**) Warm-up (Base case): Training dynamics at the first iteration $t = 0$.

Recall that we set $\boldsymbol{w}_\Delta^{(0)} = \boldsymbol{0}$. Then, we have

$$\langle \boldsymbol{w}_\Delta^{(0)}, \boldsymbol{o}_+ \rangle = 0 \quad \text{and} \quad \langle \boldsymbol{w}_\Delta^{(0)}, \boldsymbol{o}_- \rangle = 0.$$

**(S1.1)** Training dynamics of $\boldsymbol{W}_{O(i,\cdot)}$ at the first iteration $t = 0$.

From Lemma B.1, identify $p_1 = 0$ and $q_1 = 0$. Let $\alpha_r$ and $\alpha_c$ denote the average fraction of label-relevant tokens and confusion tokens, respectively. Then, for any lucky neuron $i \in \mathcal{W}(0)$, we obtain

$$\frac{1}{2\sqrt{m}L}\Theta(\alpha_r L - \alpha_c L) - \widetilde{\mathcal{O}}\left(\frac{1}{\mathrm{poly}(d)}\right) \leq \left\langle -\frac{\partial\hat{\mathcal{L}}}{\partial\boldsymbol{W}_{O(i,\cdot)}^{(0)}}, \boldsymbol{o}_+ \right\rangle$$
$$\leq \frac{1}{2\sqrt{m}L}\Theta(\alpha_r L - \alpha_c L) + \widetilde{\mathcal{O}}\left(\frac{1}{\mathrm{poly}(d)}\right). \tag{47}$$

$$\text{and} \quad \left\langle -\frac{\partial\hat{\mathcal{L}}}{\partial\boldsymbol{W}_{O(i,\cdot)}^{(0)}}, \boldsymbol{o}_j \right\rangle \leq \widetilde{\mathcal{O}}\left(\frac{1}{\mathrm{poly}(d)}\right) \quad \text{for } j \neq 1. \tag{48}$$

Recall that we set the number of samples in a batch $N = \mathrm{poly}(d)$.

Recall that the initialization is

$$\boldsymbol{W}_{O(i,\cdot)}(0) = \delta_1\boldsymbol{o}_+ + \delta_2\boldsymbol{o}_- + \cdots + \delta_d\boldsymbol{o}_d, \quad \delta_j \stackrel{\text{i.i.d.}}{\sim} \mathcal{N}(0, \xi^2) \quad j = 1, 2, \cdots, d. \tag{49}$$

Then, after one gradient descent step, we have

$$\delta_1 + \frac{\eta}{2\sqrt{m}L}\Theta(\alpha_r L - \alpha_c L) - \widetilde{\mathcal{O}}\left(\frac{1}{\mathrm{poly}(d)}\right)$$
$$\leq \left\langle \boldsymbol{W}_{O(i,\cdot)}^{\top}{}^{(1)}, \boldsymbol{o}_+ \right\rangle$$
$$\leq \delta_1 + \frac{\eta}{2\sqrt{m}L}\Theta(\alpha_r L - \alpha_c L) + \widetilde{\mathcal{O}}\left(\frac{1}{\mathrm{poly}(d)}\right) \tag{50}$$

$$\text{and} \quad \left\langle \boldsymbol{W}_{O(i,\cdot)}^{\top}{}^{(1)}, \boldsymbol{o}_j \right\rangle \leq \widetilde{\mathcal{O}}\left(\frac{1}{\mathrm{poly}(d)}\right) \quad \text{for } j \neq 1. \tag{51}$$

By applying Lemma B.3, for any lucky neuron $i \in \mathcal{U}(0)$, we obtain

$$\delta_2 + \frac{\eta}{2\sqrt{m}L}\Theta(\alpha_r L - \alpha_c L) - \widetilde{\mathcal{O}}\left(\frac{1}{\mathrm{poly}(d)}\right)$$
$$\leq \left\langle \boldsymbol{W}_{O(i,\cdot)}^{\top}{}^{(1)}, \boldsymbol{o}_- \right\rangle$$
$$\leq \delta_2 + \frac{\eta}{2\sqrt{m}L}\Theta(\alpha_r L - \alpha_c L) + \widetilde{\mathcal{O}}\left(\frac{1}{\mathrm{poly}(d)}\right) \tag{52}$$

$$\text{and} \quad \left\langle \boldsymbol{W}_{O(i,\cdot)}^{\top}{}^{(1)}, \boldsymbol{o}_j \right\rangle \leq \widetilde{\mathcal{O}}\left(\frac{1}{\mathrm{poly}(d)}\right) \quad \text{for } j \neq 2. \tag{53}$$

For any unlucky neuron $i \in \mathcal{K}_- \setminus \mathcal{U}(0)$, Lemma B.4 gives

$$\left\langle \boldsymbol{W}_{O(i,\cdot)}^{\top}{}^{(1)}, \boldsymbol{o}_j \right\rangle \leq \widetilde{\mathcal{O}}\left(\frac{1}{\mathrm{poly}(d)}\right) \quad \text{for } \forall j. \tag{54}$$

**(S1.2)** Training dynamics of $\boldsymbol{w}_\Delta$ at the first iteration $t = 0$.

Now consider the gradient update for $\boldsymbol{w}_\Delta$. Define:

$$a = \delta_1 + \frac{\eta}{2\sqrt{mL}}\Theta(\alpha_r L - \alpha_c L) - \tilde{\mathcal{O}}\left(\frac{1}{\text{poly}(d)}\right)$$

$$b = \delta_1 + \frac{\eta}{2\sqrt{mL}}\Theta(\alpha_r L - \alpha_c L) + \tilde{\mathcal{O}}\left(\frac{1}{\text{poly}(d)}\right)$$

Applying Lemma B.5 with $r_1^* = a$, $s_1^* = b$, and $\rho_0^+ = |\mathcal{W}(0)|$, we get

$$\left\langle -\frac{\partial\hat{\mathcal{L}}}{\partial\boldsymbol{w}_\Delta^{(0)}}, \boldsymbol{o}_+ \right\rangle \geq \frac{a}{2\sqrt{mL}} \cdot \rho_0^+ \cdot \Theta(\alpha_r L) - \frac{\sqrt{m}b}{4L} \cdot \Theta(\alpha_c L) - \tilde{\mathcal{O}}\left(\frac{1}{\text{poly}(d)}\right) =: \alpha \qquad (55)$$

Let $\delta_1 = \frac{1}{\text{poly}(d)}$. Since $a - b = \tilde{\mathcal{O}}\left(\frac{1}{\text{poly}(d)}\right)$ that is sufficiently small,

$$\begin{aligned}
\alpha &= \frac{1}{2L}\left[\frac{a}{\sqrt{m}} \cdot \frac{m}{2}\Theta(\alpha_r L) - \frac{\sqrt{m}b}{2}\Theta(\alpha_c L)\right] - \tilde{\mathcal{O}}\left(\frac{1}{\text{poly}(d)}\right) \\
&= \frac{1}{2L}\left[\frac{\sqrt{m}a}{2}(\Theta(\alpha_r L) - \Theta(\alpha_c L))\right] - \tilde{\mathcal{O}}\left(\frac{1}{\text{poly}(d)}\right) \\
&= \frac{\sqrt{m}}{4L} \cdot \frac{\eta}{2\sqrt{mL}}\Theta((\alpha_r L - \alpha_c L)^2) - \tilde{\mathcal{O}}\left(\frac{1}{\text{poly}(d)}\right) \\
&= \frac{\eta}{8L^2}\Theta((\alpha_r L - \alpha_c L)^2) - \tilde{\mathcal{O}}\left(\frac{1}{\text{poly}(d)}\right) > 0
\end{aligned} \qquad (56)$$

From Lemma B.6, we also obtain

$$\left\langle -\frac{\partial\hat{\mathcal{L}}}{\partial\boldsymbol{w}_\Delta^{(0)}}, \boldsymbol{o}_j \right\rangle \leq \tilde{\mathcal{O}}\left(\frac{1}{\text{poly}(d)}\right) =: \gamma \quad \text{for } j \neq 1, 2. \qquad (57)$$

(**S2**) Induction Step: Training dynamics at a general iteration $t$.

Suppose $\langle\boldsymbol{w}_\Delta^{(t)}, \boldsymbol{o}_+\rangle = \alpha^* \geq \alpha \cdot t$, $\langle\boldsymbol{w}_\Delta^{(t)}, \boldsymbol{o}_-\rangle = \beta^* \geq \beta \cdot t$, and $\langle\boldsymbol{w}_\Delta^{(t)}, \boldsymbol{o}_j\rangle = \gamma^* \leq \gamma \cdot t$, where

$$\beta = \frac{a'}{2\sqrt{mL}} \cdot \rho_0^- \cdot \Theta(\alpha_r L) - \frac{\sqrt{m}b'}{4L} \cdot \Theta(\alpha_c L) - \tilde{\mathcal{O}}\left(\frac{1}{\text{poly}(d)}\right) > 0, \qquad (58)$$

$$a' = \delta_2 + \frac{\eta}{2\sqrt{mL}}\Theta(\alpha_r L - \alpha_c L) - \tilde{\mathcal{O}}\left(\frac{1}{\text{poly}(d)}\right), \qquad (59)$$

$$b' = \delta_2 + \frac{\eta}{2\sqrt{mL}}\Theta(\alpha_r L - \alpha_c L) + \tilde{\mathcal{O}}\left(\frac{1}{\text{poly}(d)}\right). \qquad (60)$$

Following the same approach as in (56), we can simplify and obtain

$$\beta = \frac{\eta}{8L^2}\Theta((\alpha_r L - \alpha_c L)^2) - \tilde{\mathcal{O}}\left(\frac{1}{\text{poly}(d)}\right) > 0. \qquad (61)$$

For any lucky neuron $i \in \mathcal{W}(t)$ at the $(t+1)$-th iteration, we have

$$\frac{1}{\sqrt{mL}} \cdot \sigma(\alpha^*)\Theta(\alpha_r L - \alpha_c L) - \tilde{\mathcal{O}}\left(\frac{1}{\text{poly}(d)}\right)$$

$$\leq \left\langle -\frac{\partial\hat{\mathcal{L}}}{\partial\boldsymbol{W}_{O(i,\cdot)}^{(t)}}, \boldsymbol{o}_+ \right\rangle$$

$$\leq \frac{1}{\sqrt{mL}} \cdot \sigma(\alpha^*)\Theta(\alpha_r L - \alpha_c L) + \tilde{\mathcal{O}}\left(\frac{1}{\text{poly}(d)}\right),$$

and

$$\left\langle -\frac{\partial \hat{\mathcal{L}}}{\partial \boldsymbol{W}_{O(i,\cdot)}^{(t)}}, \boldsymbol{o}_j \right\rangle \leq \tilde{\mathcal{O}}\left(\frac{1}{\text{poly}(d)}\right) \quad \text{for } j \neq 1 \tag{62}$$

Next, we have $\sigma(\alpha^*) > \frac{1}{2}$ since $\alpha^* > 0$ when $t = 1$. By a simple induction, this further ensures

$$\left\langle -\frac{\partial \hat{\mathcal{L}}}{\partial \boldsymbol{W}_{O(i,\cdot)}^{(0)}}, \boldsymbol{o}_+ \right\rangle \leq \left\langle -\frac{\partial \hat{\mathcal{L}}}{\partial \boldsymbol{W}_{O(i,\cdot)}^{(1)}}, \boldsymbol{o}_+ \right\rangle \leq \cdots \left\langle -\frac{\partial \hat{\mathcal{L}}}{\partial \boldsymbol{W}_{O(i,\cdot)}^{(t)}}, \boldsymbol{o}_+ \right\rangle \leq \left\langle -\frac{\partial \hat{\mathcal{L}}}{\partial \boldsymbol{W}_{O(i,\cdot)}^{(t+1)}}, \boldsymbol{o}_+ \right\rangle.$$
$$\tag{63}$$

Thus, we obtain the following bound after the second gradient descent step:

$$\delta_1 + \frac{\eta}{2\sqrt{mL}}\Theta(\alpha_r L - \alpha_c L)\left[1 + 2\sigma(\alpha^*)\right] - \tilde{\mathcal{O}}\left(\frac{1}{\text{poly}(d)}\right) =: u$$
$$\leq \left\langle (\boldsymbol{W}_{O(i,\cdot)}^{(2)})^\top, \boldsymbol{o}_+ \right\rangle$$
$$\leq \delta_1 + \frac{\eta}{2\sqrt{mL}}\Theta(\alpha_r L - \alpha_c L)\left[1 + 2\sigma(\alpha^*)\right] + \tilde{\mathcal{O}}\left(\frac{1}{\text{poly}(d)}\right) =: v. \tag{64}$$

Similarly, applying Lemma B.3 to any lucky neuron $i \in \mathcal{U}(1)$ at iteration 2, we get

$$\frac{1}{\sqrt{mL}} \cdot \sigma(\beta^*)\Theta(\alpha_r L - \alpha_c L) - \tilde{\mathcal{O}}\left(\frac{1}{\text{poly}(d)}\right)$$
$$\leq \left\langle -\frac{\partial \hat{\mathcal{L}}}{\partial \boldsymbol{W}_{O(i,\cdot)}^{(1)}}, \boldsymbol{o}_- \right\rangle \tag{65}$$
$$\leq \frac{1}{\sqrt{mL}} \cdot \sigma(\beta^*)\Theta(\alpha_r L - \alpha_c L) + \tilde{\mathcal{O}}\left(\frac{1}{\text{poly}(d)}\right),$$

$$\text{and} \quad \left\langle -\frac{\partial \hat{\mathcal{L}}}{\partial \boldsymbol{W}_{O(i,\cdot)}^{(1)}}, \boldsymbol{o}_j \right\rangle \leq \tilde{\mathcal{O}}\left(\frac{1}{\text{poly}(d)}\right) \quad \text{for } j \neq 2. \tag{66}$$

Applying Lemma B.5 with $r_1^* = u$, and $s_1^* = v$, we obtain

$$\left\langle -\frac{\partial \hat{\mathcal{L}}}{\partial \boldsymbol{w}_\Delta^{(1)}}, \boldsymbol{o}_+ \right\rangle \geq \frac{u}{2\sqrt{mL}} \cdot \rho_1^+ \cdot \Theta(\alpha_r L) - \frac{\sqrt{m}v}{4L} \cdot \Theta(\alpha_c L) - \tilde{\mathcal{O}}\left(\frac{1}{\text{poly}(d)}\right) =: \chi. \tag{67}$$

Since $u - v = \tilde{\mathcal{O}}\left(\frac{1}{\text{poly}(d)}\right)$ that is sufficiently small, we have $\chi \geq 0$.

By applying Lemma B.6, we get

$$\left\langle -\frac{\partial \hat{\mathcal{L}}}{\partial \boldsymbol{w}_\Delta^{(0)}}, \boldsymbol{o}_j \right\rangle \leq \tilde{\mathcal{O}}\left(\frac{1}{\text{poly}(d)}\right) \quad \text{for } j \neq 1, 2. \tag{68}$$

(**S3**) Induction conclusion: Training dynamics when the algorithm ends.

We proceed by induction on $t$: the base case $t = 0$ is established in (S1), and the induction step for general $t$ is shown in (S2). For any lucky neuron $i \in \mathcal{W}(T)$, we obtain

$$\left\langle \boldsymbol{W}_{O(i,\cdot)}^{\top(T)}, \boldsymbol{o}_+ \right\rangle \geq aT, \tag{69}$$

$$\text{and} \quad \left\langle \boldsymbol{W}_{O(i,\cdot)}^{\top(T)}, \boldsymbol{o}_j \right\rangle \leq \tilde{\mathcal{O}}\left(\frac{1}{\text{poly}(d)}\right) \quad \text{for } j \neq 1 \tag{70}$$

For any lucky neuron $i \in \mathcal{U}(T)$, we obtain

$$\left\langle \boldsymbol{W}_{O(i,\cdot)}^{\top}{}^{(T)}, \boldsymbol{o}_{-} \right\rangle \geq aT, \tag{71}$$

$$\text{and} \quad \left\langle \boldsymbol{W}_{O(i,\cdot)}^{\top}{}^{(T)}, \boldsymbol{o}_{j} \right\rangle \leq \tilde{\mathcal{O}}\left(\frac{1}{\text{poly}(d)}\right) \quad \text{for } j \neq 2 \tag{72}$$

Also, we obtain

$$\left\langle \boldsymbol{w}_{\Delta}^{(T)}, \boldsymbol{o}_{+} \right\rangle \geq \alpha T, \tag{73}$$

$$\left\langle \boldsymbol{w}_{\Delta}^{(T)}, \boldsymbol{o}_{-} \right\rangle \geq \beta T, \tag{74}$$

$$\text{and} \quad \left\langle \boldsymbol{w}_{\Delta}^{(T)}, \boldsymbol{o}_{j} \right\rangle \leq \gamma T. \tag{75}$$

(**S4**) Derivation for the generalization bound.

We will demonstrate that once the weights have converged at iteration $T$, the model accurately captures the underlying data distribution, which leads to zero generalization error, as shown in (94).

Consider $z^{(n)} = +1$ as an example. The sequence $\boldsymbol{X}^{(n)} = \begin{bmatrix} \boldsymbol{x}_{1}^{(n)} & \boldsymbol{x}_{2}^{(n)} & \cdots & \boldsymbol{x}_{L}^{(n)} \end{bmatrix}$ has first $\alpha_r L$ tokens correspond to the feature $\boldsymbol{o}_{+}$, while the following $\alpha_c L$ tokens correspond to the feature $\boldsymbol{o}_{-}$.

$$
\begin{aligned}
F(\boldsymbol{X}^{(n)}) &= \frac{1}{L} \sum_{l=1}^{L} \sum_{i=1}^{m} v_i \, \phi\left(\boldsymbol{W}_{O(i,\cdot)} \boldsymbol{y}_l^{(n)}\right) \\
&= \frac{1}{\sqrt{m}L} \sum_{i \in \mathcal{K}^+} \sum_{l=1}^{L} \phi\left(\boldsymbol{W}_{O(i,\cdot)} \boldsymbol{y}_l^{(n)}\right) - \frac{1}{\sqrt{m}L} \sum_{i \in \mathcal{K}^-} \sum_{l=1}^{L} \phi\left(\boldsymbol{W}_{O(i,\cdot)} \boldsymbol{y}_l^{(n)}\right) \\
&\geq \frac{1}{\sqrt{m}L} \sum_{i \in \mathcal{W}(0)} \sum_{l=1}^{L} \phi\left(\boldsymbol{W}_{O(i,\cdot)} \boldsymbol{y}_l^{(n)}\right) - \frac{1}{\sqrt{m}L} \sum_{i \in \mathcal{U}(0)} \sum_{l=1}^{L} \phi\left(\boldsymbol{W}_{O(i,\cdot)} \boldsymbol{y}_l^{(n)}\right) \\
&\quad - \frac{1}{\sqrt{m}L} \sum_{i \in \mathcal{K}^- \setminus \mathcal{U}(0)} \sum_{l=1}^{L} \phi\left(\boldsymbol{W}_{O(i,\cdot)} \boldsymbol{y}_l^{(n)}\right)
\end{aligned}
\tag{76}
$$

The Mamba output $\boldsymbol{y}_l^{(n)}$ is defined as

$$\boldsymbol{y}_l^{(n)} = \sum_{s=1}^{l} \left( \prod_{j=s+1}^{l} \left(1 - \sigma(\boldsymbol{w}_{\Delta}^{\top} \boldsymbol{x}_j^{(n)})\right) \right) \cdot \sigma(\boldsymbol{w}_{\Delta}^{\top} \boldsymbol{x}_s^{(n)}) \cdot (\boldsymbol{x}_s^{(n)\top} \boldsymbol{x}_l^{(n)}) \boldsymbol{x}_s^{(n)}. \tag{77}$$

We now derive a lower bound for

$$\sum_{i \in \mathcal{W}(0)} \sum_{l=1}^{L} \phi(\boldsymbol{W}_{O(i,\cdot)} \boldsymbol{y}_l).$$

To that end, consider the aggregated projection

$$\sum_{i \in \mathcal{W}(0)} \sum_{l=1}^{L} \boldsymbol{W}_{O(i,\cdot)} \boldsymbol{y}_l = \sum_{i \in \mathcal{W}(0)} \sum_{l=1}^{L} \sum_{j=1}^{d} \langle \boldsymbol{W}_{O(i,\cdot)}^{\top}, \boldsymbol{o}_j \rangle \cdot \langle \boldsymbol{y}_l, \boldsymbol{o}_j \rangle. \tag{78}$$

For any $i \in \mathcal{W}(0)$, we know that

$$\langle \boldsymbol{W}_{O(i,\cdot)}^{\top}, \boldsymbol{o}_+ \rangle \geq aT. \tag{79}$$

Hence, let's obtain a lower bound for $\langle \boldsymbol{y}_l, \boldsymbol{o}_+ \rangle$.

We only need to consider the cases where $\boldsymbol{x}_s = \boldsymbol{o}_+$ for some $s$ in the range $1 \leq s \leq l$.

After $T$ iterations, we know

$$\langle \boldsymbol{w}_\Delta, \boldsymbol{o}_+ \rangle \geq \alpha T, \quad \langle \boldsymbol{w}_\Delta, \boldsymbol{o}_- \rangle \geq \beta T, \quad \langle \boldsymbol{w}_\Delta, \boldsymbol{o}_j \rangle \leq \gamma T \quad \text{for } j \neq 1, 2. \tag{80}$$

Therefore, we have

$$\langle \boldsymbol{y}_l, \boldsymbol{o}_+ \rangle = \Theta(\sigma(\langle \boldsymbol{w}_\Delta, \boldsymbol{o}_+ \rangle)) = \Theta(\sigma(\alpha T)), \quad \text{for } l = 1, 2, \ldots, \alpha_r L. \tag{81}$$

We now lower bound the objective

$$\sum_{i \in \mathcal{W}(0)} \sum_{l=1}^{L} \phi(\boldsymbol{W}_{O(i,\cdot)} \boldsymbol{y}_l).$$

Note that

$$\boldsymbol{W}_{O(i,\cdot)} \boldsymbol{y}_l = \sum_{j=1}^{d} \left\langle \boldsymbol{W}_{O(i,\cdot)}^{\top}, \boldsymbol{o}_j \right\rangle \left\langle \boldsymbol{y}_{L_1^+}, \boldsymbol{o}_j \right\rangle,$$

and $\boldsymbol{y}_l$ has only $\boldsymbol{o}_+$ component for $l = 1, 2, \ldots, \alpha_r L$.

Therefore,

$$\boldsymbol{W}_{O(i,\cdot)} \boldsymbol{y}_l = \left\langle \boldsymbol{W}_{O(i,\cdot)}^{\top}, \boldsymbol{o}_+ \right\rangle \langle \boldsymbol{y}_l, \boldsymbol{o}_+ \rangle \geq aT \cdot \Theta(\sigma(\alpha T)) > 0, \quad \text{for } l = 1, 2, \ldots, \alpha_r L.$$

Applying $\phi(z) = z$ for positive $z$, we obtain

$$\phi(\boldsymbol{W}_{O(i,\cdot)} \boldsymbol{y}_l) \geq aT \cdot \Theta(\sigma(\alpha T)), \quad \text{for } l = 1, 2, \ldots, \alpha_r L.$$

Hence,

$$\sum_{i \in \mathcal{W}(0)} \sum_{l=1}^{L} \phi(\boldsymbol{W}_{O(i,\cdot)} \boldsymbol{y}_l) \geq \sum_{i \in \mathcal{W}(0)} aT \cdot \Theta(\sigma(\alpha T)) \cdot \alpha_r L \tag{82}$$

Next, we derive an upper bound for

$$\sum_{i \in \mathcal{U}(0)} \sum_{l=1}^{L} \phi\left(\boldsymbol{W}_{O(i,\cdot)} \boldsymbol{y}_l^{(n)}\right).$$

For any $i \in \mathcal{U}(0)$, we know that

$$0 < \langle \boldsymbol{W}_{O(i,\cdot)}^{\top}, \boldsymbol{o}_- \rangle \leq bT. \tag{83}$$

We now derive an upper bound for $\langle \boldsymbol{y}_l, \boldsymbol{o}_- \rangle$. We only need to consider the cases where $\boldsymbol{x}_s = \boldsymbol{o}_-$ such that $1 \leq s \leq l$.

We have,

$$\langle \boldsymbol{w}_\Delta, \boldsymbol{o}_+ \rangle \leq WT, \qquad \langle \boldsymbol{w}_\Delta, \boldsymbol{o}_- \rangle \leq WT,$$

where

$$W = \frac{\eta}{8L^2} \Theta((\alpha_r L - \alpha_c L)^2) + \tilde{\mathcal{O}}\left(\frac{1}{\text{poly}(d)}\right). \tag{84}$$

$$\langle \boldsymbol{y}_l, \boldsymbol{o}_- \rangle = \Theta(\sigma(\langle \boldsymbol{w}_\Delta, \boldsymbol{o}_- \rangle)) = \Theta(\sigma(WT)), \quad \text{for } l = 1, 2, \ldots, \alpha_c L. \tag{85}$$

$$\sum_{l=1}^{L} \boldsymbol{W}_{O(i,\cdot)}\boldsymbol{y}_l \leq bT \cdot \Theta(\sigma(WT)) \cdot \alpha_c L. \tag{86}$$

$$\sum_{i\in\mathcal{U}(0)}\sum_{l=1}^{L} \phi\left(\boldsymbol{W}_{O(i,\cdot)}\boldsymbol{y}_l^{(n)}\right) \leq \sum_{i\in\mathcal{U}(0)} bT \cdot \Theta(\sigma(WT)) \cdot \alpha_c L. \tag{87}$$

In addition, we have

$$\sum_{i\in\mathcal{K}^-\setminus\mathcal{U}(0)}\sum_{l=1}^{L} \phi\left(\boldsymbol{W}_{O(i,\cdot)}\boldsymbol{y}_l^{(n)}\right) \leq \tilde{\mathcal{O}}\left(\frac{1}{\mathrm{poly}(d)}\right). \tag{88}$$

By (76), we can write

$$F(\boldsymbol{X}^{(n)}) \geq \frac{1}{\sqrt{mL}}\left\{\frac{m}{2}\cdot aT \cdot \Theta(\sigma(\alpha T))\cdot \alpha_r L - \frac{m}{2}\cdot bT\cdot\Theta(\sigma(WT))\cdot\alpha_c L - \tilde{\mathcal{O}}\left(\frac{1}{\mathrm{poly}(d)}\right)\right\}, \tag{89}$$

with

$$a = \frac{\eta}{2\sqrt{mL}}\Theta(\alpha_r L - \alpha_c L) - \tilde{\mathcal{O}}\left(\frac{1}{\mathrm{poly}(d)}\right), \tag{90}$$

$$\text{and}\quad b = \frac{\eta}{2\sqrt{mL}}\Theta(\alpha_r L - \alpha_c L) + \tilde{\mathcal{O}}\left(\frac{1}{\mathrm{poly}(d)}\right). \tag{91}$$

$$\alpha = \frac{\eta}{8L^2}\Theta((\alpha_r L - \alpha_c L)^2) - \tilde{\mathcal{O}}\left(\frac{1}{\mathrm{poly}(d)}\right)$$
$$= W - \tilde{\mathcal{O}}\left(\frac{1}{\mathrm{poly}(d)}\right). \tag{92}$$

Therefore, we conclude that

$$F(\boldsymbol{X}^{(n)}) \geq \frac{\sqrt{m}}{2}\cdot aT\cdot\Theta(\sigma(\alpha T))\cdot(\alpha_r - \alpha_c) - \tilde{\mathcal{O}}\left(\frac{1}{\mathrm{poly}(d)}\right) \tag{93}$$

There, for any positive sample, we can prove that

$$F(\boldsymbol{X}^{(n)}) \geq C, \text{ where } C \text{ is some positive constant.} \tag{94}$$

Similar to the previous analysis, one can show that the negtive sample $\boldsymbol{X}_n$ leads to $F(\boldsymbol{X}^{(n)}) \leq -C$.

(**S4.1**) Derivation for the convergence rate.

Let's find the number of iterations $T$ required such that $F(\boldsymbol{X}^{(n)}) \geq 1$, since the label is $+1$. We require

$$\frac{\sqrt{m}}{2}\cdot aT(\alpha_r - \alpha_c) \geq 1 + \epsilon. \tag{95}$$

Substituting the value of $a \approx b = \frac{\eta}{2\sqrt{mL}}\Theta(\alpha_r L - \alpha_c L)$, the condition becomes

$$\frac{\sqrt{m}aT}{2}(\alpha_r - \alpha_c) = \frac{\sqrt{m}}{2}\cdot\frac{\eta}{2\sqrt{mL}}\Theta(\alpha_r L - \alpha_c L)T\cdot(\alpha_r - \alpha_c)$$
$$= \frac{\eta T}{4}\Theta((\alpha_r - \alpha_c)^2) \geq 1 + \epsilon. \tag{96}$$

Solving for $T$, we obtain

$$T \geq \frac{4(1+\epsilon)}{\eta\Theta((\alpha_r - \alpha_c)^2)} \geq \frac{4}{\eta\Theta((\alpha_r - \alpha_c)^2)}. \tag{97}$$

Now, we additionally require that the sigmoid activation $\sigma(\alpha T)$ be sufficiently large, i.e.,

$$\sigma(\alpha T) \geq 1 - \epsilon. \tag{98}$$

When $z$ is sufficiently large we can approximate

$$\sigma(z) = \frac{1}{1 + e^{-z}} \approx 1 - e^{-z}.$$

Substituting $z = \alpha T$, condition (98) becomes:

$$
\begin{aligned}
\sigma(\alpha T) \approx 1 - e^{-\alpha T} &\geq 1 - \epsilon, \\
e^{-\alpha T} &\leq \epsilon, \\
\alpha T &\geq -\ln(\epsilon) \\
T &\geq -\frac{\ln(\epsilon)}{\alpha}.
\end{aligned}
\tag{99}
$$

Substituting $\alpha = \frac{\eta}{8L^2}\Theta((\alpha_r L - \alpha_c L)^2)$, we get:

$$T \geq -\ln(\epsilon) \cdot \frac{8L^2}{\eta\Theta((\alpha_r L - \alpha_c L)^2)}. \tag{100}$$

$$T \geq -\ln(\epsilon) \cdot \frac{8}{\eta\Theta((\alpha_r - \alpha_c)^2)}. \tag{101}$$

Hence, by combining (97) and (101), we obtain

$$T \geq \max\left\{ \frac{4}{\eta\Theta((\alpha_r - \alpha_c)^2)}, -\ln(\epsilon) \cdot \frac{8}{\eta\Theta((\alpha_r - \alpha_c)^2)} \right\}. \tag{102}$$

By combining (95) and (98) with the expression for the model output $F(\boldsymbol{X}^{(n)})$ in (93), we obtain

$$
\begin{aligned}
F(\boldsymbol{X}^{(n)}) &\geq (1 + \epsilon) \cdot (1 - \epsilon) \\
&\geq 1 - \mathcal{O}(\epsilon^2)
\end{aligned}
\tag{103}
$$

Hence, for sufficiently small $\epsilon > 0$, the model output satisfies $F(\boldsymbol{X}^{(n)}) \geq 1$.

Similarly, for a negative sample, one can show by symmetry that the model output satisfies $F(\boldsymbol{X}^{(n)}) \leq -1$.

(**S4.2**) Derivation for the sample complexity.

Now we derive a sample-complexity bound that guarantees zero generalization error.

Assuming enough samples, we can write for sufficiently small $\lambda \ll 1$

$$\mathcal{O}\left( \sqrt{\frac{d \log N}{mN}} \right) \leq \lambda \cdot \frac{\eta}{2\sqrt{m}L}\Theta(\alpha_r L - \alpha_c L). \tag{104}$$

From this, we can derive a lower bound on the required sample size,

$$
\begin{aligned}
N &\geq \Omega\left( \lambda^{-2} \cdot \frac{4d}{\eta^2\Theta((\alpha_r - \alpha_c)^2)} \right) \\
&\geq \Omega\left( \frac{d}{\eta^2\Theta((\alpha_r - \alpha_c)^2)} \right),
\end{aligned}
\tag{105}
$$

which will be (13) in Theorem 1.

$\square$

## C    LOCALITY-STRUCTURED DATA

### C.1    USEFUL LEMMAS

Lemma C.1 provides bounds on the gradient updates of lucky neurons $i \in \mathcal{W}(t)$ in the directions of both class-relevant features ($\boldsymbol{o}_+$, $\boldsymbol{o}_-$) and irrelevant features.

**Lemma C.1.** *Suppose $p_1 \leq \langle \boldsymbol{w}_\Delta^{(t)}, \boldsymbol{o}_+ \rangle \leq q_1$, $p_1 \leq \langle \boldsymbol{w}_\Delta^{(t)}, \boldsymbol{o}_- \rangle \leq q_1$, and $p_2 \leq \langle \boldsymbol{w}_\Delta^{(t)}, \boldsymbol{o}_j \rangle \leq q_2$ for $j \neq 1, 2$. Then, for any lucky neuron $i \in \mathcal{W}(t)$ at iteration $t$, the following bounds hold:*

**(L1.1)** *A lower bound on the gradient of $\hat{\mathcal{L}}$ with respect to $\boldsymbol{W}_{O(i,\cdot)}$ at iteration $t$, in the direction of $\boldsymbol{o}_+$, is given by*

$$\left\langle -\frac{\partial \hat{\mathcal{L}}}{\partial \boldsymbol{W}_{O(i,\cdot)}^{(t)}}, \boldsymbol{o}_+ \right\rangle \geq \frac{1}{\sqrt{m}L} \cdot \sigma(p_1) \cdot (1 - \sigma(q_1))^2 \left[ (1 - \sigma(q_2))^{\Delta L_{\boldsymbol{o}_+}^+ - 2} - (1 - \sigma(p_2))^{\Delta L_{\boldsymbol{o}_+}^- - 2} \right]$$
$$- \mathcal{O}\left( \sqrt{\frac{d \log N}{mN}} \right). \tag{106}$$

**(L1.2)** *An upper bound on the gradient of $\hat{\mathcal{L}}$ with respect to $\boldsymbol{W}_{O(i,\cdot)}$ at iteration $t$, in the direction of $\boldsymbol{o}_+$, is given by*

$$\left\langle -\frac{\partial \hat{\mathcal{L}}}{\partial \boldsymbol{W}_{O(i,\cdot)}^{(t)}}, \boldsymbol{o}_+ \right\rangle \leq \frac{1}{\sqrt{m}L} \cdot \sigma(q_1) \cdot (1 - \sigma(p_1))^2 \left[ (1 - \sigma(p_2))^{\Delta L_{\boldsymbol{o}_+}^+ - 2} - (1 - \sigma(q_2))^{\Delta L_{\boldsymbol{o}_+}^- - 2} \right]$$
$$+ \mathcal{O}\left( \sqrt{\frac{d \log N}{mN}} \right). \tag{107}$$

**(L1.3)** *A lower bound on the gradient of $\hat{\mathcal{L}}$ with respect to $\boldsymbol{W}_{O(i,\cdot)}$ at iteration $t$, in the direction of $\boldsymbol{o}_-$, is given by*

$$\left\langle -\frac{\partial \hat{\mathcal{L}}}{\partial \boldsymbol{W}_{O(i,\cdot)}^{(t)}}, \boldsymbol{o}_- \right\rangle \geq -\frac{1}{\sqrt{m}L} \cdot \sigma(q_1) \cdot (1 - \sigma(p_1))^2 \left[ (1 - \sigma(p_2))^{\Delta L_{\boldsymbol{o}_-}^- - 2} - (1 - \sigma(q_2))^{\Delta L_{\boldsymbol{o}_-}^+ - 2} \right]$$
$$- \mathcal{O}\left( \sqrt{\frac{d \log N}{mN}} \right). \tag{108}$$

**(L1.4)** *An upper bound on the gradient of $\hat{\mathcal{L}}$ with respect to $\boldsymbol{W}_{O(i,\cdot)}$ at iteration $t$, in the direction of $\boldsymbol{o}_-$, is given by*

$$\left\langle -\frac{\partial \hat{\mathcal{L}}}{\partial \boldsymbol{W}_{O(i,\cdot)}^{(t)}}, \boldsymbol{o}_- \right\rangle \leq \mathcal{O}\left( \sqrt{\frac{d \log N}{mN}} \right). \tag{109}$$

**(L1.5)** *An upper bound on the gradient of $\hat{\mathcal{L}}$ with respect to $\boldsymbol{W}_{O(i,\cdot)}$ at iteration $t$, in the direction of $\boldsymbol{o}_j$, is given by*

$$\left\langle -\frac{\partial \hat{\mathcal{L}}}{\partial \boldsymbol{W}_{O(i,\cdot)}^{(t)}}, \boldsymbol{o}_j \right\rangle \leq \mathcal{O}\left( \sqrt{\frac{d \log N}{mN}} \right), \quad \text{for } j \neq 1, 2. \tag{110}$$

Lemma C.2 shows that, for unlucky neurons associated with the positive class, the gradients in the directions of both class-relevant and irrelevant features are small.

**Lemma C.2.** *For any unlucky neuron $i \in \mathcal{K}_+ \setminus \mathcal{W}(t)$ at iteration $t$, the following bounds hold:*

**(L2.1)** *An upper bound on the gradient of $\hat{\mathcal{L}}$ with respect to $\boldsymbol{W}_{O(i,\cdot)}$ at iteration $t$, in the direction of $\boldsymbol{o}_+$, is given by*

$$\left\langle -\frac{\partial \hat{\mathcal{L}}}{\partial \boldsymbol{W}_{O(i,\cdot)}^{(t)}}, \boldsymbol{o}_+ \right\rangle \leq \mathcal{O}\left( \sqrt{\frac{d \log N}{mN}} \right). \tag{111}$$

**(L2.2)** *An upper bound on the gradient of $\hat{\mathcal{L}}$ with respect to $\boldsymbol{W}_{O(i,\cdot)}$ at iteration $t$, in the direction of $\boldsymbol{o}_-$, is given by*

$$\left\langle -\frac{\partial \hat{\mathcal{L}}}{\partial \boldsymbol{W}_{O(i,\cdot)}^{(t)}}, \boldsymbol{o}_- \right\rangle \leq \mathcal{O}\left( \sqrt{\frac{d \log N}{mN}} \right). \tag{112}$$

**(L2.3)** *An upper bound on the gradient of $\hat{\mathcal{L}}$ with respect to $\boldsymbol{W}_{O(i,\cdot)}$ at iteration $t$, in the direction of $\boldsymbol{o}_j$, is given by*

$$\left\langle -\frac{\partial \hat{\mathcal{L}}}{\partial \boldsymbol{W}_{O(i,\cdot)}^{(t)}}, \boldsymbol{o}_j \right\rangle \leq \mathcal{O}\left( \sqrt{\frac{d \log N}{mN}} \right), \quad \text{for } j \neq 1, 2. \tag{113}$$

Lemmas C.3 and C.4, by symmetry, state the analogous results for lucky and unlucky neurons associated with the negative class.

**Lemma C.3.** *Suppose $p_1 \leq \langle \boldsymbol{w}_\Delta^{(t)}, \boldsymbol{o}_- \rangle \leq q_1$, $p_1 \leq \langle \boldsymbol{w}_\Delta^{(t)}, \boldsymbol{o}_+ \rangle \leq q_1$, and $p_2 \leq \langle \boldsymbol{w}_\Delta^{(t)}, \boldsymbol{o}_j \rangle \leq q_2$ for $j \neq 1, 2$. Then, for any lucky neuron $i \in \mathcal{U}(t)$ at iteration $t$, the following bounds hold:*

**(L3.1)** *A lower bound on the gradient of $\hat{\mathcal{L}}$ with respect to $\boldsymbol{W}_{O(i,\cdot)}$ at iteration $t$, in the direction of $\boldsymbol{o}_-$, is given by*

$$\left\langle -\frac{\partial \hat{\mathcal{L}}}{\partial \boldsymbol{W}_{O(i,\cdot)}^{(t)}}, \boldsymbol{o}_- \right\rangle \geq \frac{1}{\sqrt{mL}} \cdot \sigma(p_1) \cdot (1 - \sigma(q_1))^2 \left[ (1 - \sigma(q_2))^{\Delta L_{\boldsymbol{o}_-}^- - 2} - (1 - \sigma(p_2))^{\Delta L_{\boldsymbol{o}_-}^+ - 2} \right]$$
$$- \mathcal{O}\left( \sqrt{\frac{d \log N}{mN}} \right). \tag{114}$$

**(L3.2)** *An upper bound on the gradient of $\hat{\mathcal{L}}$ with respect to $\boldsymbol{W}_{O(i,\cdot)}$ at iteration $t$, in the direction of $\boldsymbol{o}_-$, is given by*

$$\left\langle -\frac{\partial \hat{\mathcal{L}}}{\partial \boldsymbol{W}_{O(i,\cdot)}^{(t)}}, \boldsymbol{o}_- \right\rangle \leq \frac{1}{\sqrt{mL}} \cdot \sigma(q_1) \cdot (1 - \sigma(p_1))^2 \left[ (1 - \sigma(p_2))^{\Delta L_{\boldsymbol{o}_-}^- - 2} - (1 - \sigma(q_2))^{\Delta L_{\boldsymbol{o}_-}^+ - 2} \right]$$
$$+ \mathcal{O}\left( \sqrt{\frac{d \log N}{mN}} \right). \tag{115}$$

**(L3.3)** *A lower bound on the gradient of $\hat{\mathcal{L}}$ with respect to $\boldsymbol{W}_{O(i,\cdot)}$ at iteration $t$, in the direction of $\boldsymbol{o}_+$, is given by*

$$\left\langle -\frac{\partial \hat{\mathcal{L}}}{\partial \boldsymbol{W}_{O(i,\cdot)}^{(t)}}, \boldsymbol{o}_+ \right\rangle \geq -\frac{1}{\sqrt{mL}} \cdot \sigma(q_1) \cdot (1 - \sigma(p_1))^2 \left[ (1 - \sigma(p_2))^{\Delta L_{\boldsymbol{o}_+}^+ - 2} - (1 - \sigma(q_2))^{\Delta L_{\boldsymbol{o}_+}^- - 2} \right]$$
$$- \mathcal{O}\left( \sqrt{\frac{d \log N}{mN}} \right). \tag{116}$$

**(L3.4)** *An upper bound on the gradient of $\hat{\mathcal{L}}$ with respect to $\boldsymbol{W}_{O(i,\cdot)}$ at iteration $t$, in the direction of $\boldsymbol{o}_+$, is given by*

$$\left\langle -\frac{\partial \hat{\mathcal{L}}}{\partial \boldsymbol{W}_{O(i,\cdot)}^{(t)}}, \boldsymbol{o}_+ \right\rangle \leq \mathcal{O}\left( \sqrt{\frac{d \log N}{mN}} \right). \tag{117}$$

**(L3.5)** *An upper bound on the gradient of $\hat{\mathcal{L}}$ with respect to $\boldsymbol{W}_{O(i,\cdot)}$ at iteration $t$, in the direction of $\boldsymbol{o}_j$, is given by*

$$\left\langle -\frac{\partial \hat{\mathcal{L}}}{\partial \boldsymbol{W}_{O(i,\cdot)}^{(t)}}, \boldsymbol{o}_j \right\rangle \leq \mathcal{O}\left( \sqrt{\frac{d \log N}{mN}} \right), \quad \text{for } j \neq 1, 2. \tag{118}$$

**Lemma C.4.** *For any unlucky neuron $i \in \mathcal{K}_- \setminus \mathcal{U}(t)$ at iteration $t$, the following bounds hold:*

*(L4.1) An upper bound on the gradient of $\hat{\mathcal{L}}$ with respect to $\boldsymbol{W}_{O(i,\cdot)}$ at iteration $t$, in the direction of $\boldsymbol{o}_-$, is given by*

$$\left\langle -\frac{\partial \hat{\mathcal{L}}}{\partial \boldsymbol{W}_{O(i,\cdot)}^{(t)}}, \boldsymbol{o}_- \right\rangle \leq \mathcal{O}\left(\sqrt{\frac{d \log N}{mN}}\right). \tag{119}$$

*(L4.2) An upper bound on the gradient of $\hat{\mathcal{L}}$ with respect to $\boldsymbol{W}_{O(i,\cdot)}$ at iteration $t$, in the direction of $\boldsymbol{o}_+$, is given by*

$$\left\langle -\frac{\partial \hat{\mathcal{L}}}{\partial \boldsymbol{W}_{O(i,\cdot)}^{(t)}}, \boldsymbol{o}_+ \right\rangle \leq \mathcal{O}\left(\sqrt{\frac{d \log N}{mN}}\right). \tag{120}$$

*(L4.3) An upper bound on the gradient of $\hat{\mathcal{L}}$ with respect to $\boldsymbol{W}_{O(i,\cdot)}$ at iteration $t$, in the direction of $\boldsymbol{o}_j$, is given by*

$$\left\langle -\frac{\partial \hat{\mathcal{L}}}{\partial \boldsymbol{W}_{O(i,\cdot)}^{(t)}}, \boldsymbol{o}_j \right\rangle \leq \mathcal{O}\left(\sqrt{\frac{d \log N}{mN}}\right), \quad \text{for } j \neq 1, 2. \tag{121}$$

Lemma C.5 establishes bounds for the gradient updates of $\boldsymbol{w}_\Delta$ in the class-relevant feature directions.

**Lemma C.5.** *Suppose $p_1 \leq \langle \boldsymbol{w}_\Delta^{(t)}, \boldsymbol{o}_+ \rangle \leq q_1$ and $r_1^* \leq \langle \boldsymbol{W}_{O(i,\cdot)}^{(t+1)\top}, \boldsymbol{o}_+ \rangle \leq s_1^*$. Let $|\mathcal{W}(t)| = \rho_t^+$ and $|\mathcal{U}(t)| = \rho_t^-$. Then, we have:*

$$\left\langle -\frac{\partial \hat{\mathcal{L}}}{\partial \boldsymbol{w}_\Delta^{(t)}}, \boldsymbol{o}_+ \right\rangle \geq \frac{\sigma(p_1)(1 - \sigma(q_1)) r_1^* \cdot \rho_t^+}{\sqrt{m}} - \frac{\sigma(q_1)(1 - \sigma(p_1)) s_1^* \cdot \sqrt{m}}{2} - \mathcal{O}\left(\sqrt{\frac{d \log N}{mN}}\right). \tag{122}$$

*Suppose $p_1 \leq \langle \boldsymbol{w}_\Delta^{(t)}, \boldsymbol{o}_- \rangle \leq q_1$ and $r_1^* \leq \langle \boldsymbol{W}_{O(i,\cdot)}^{(t+1)\top}, \boldsymbol{o}_- \rangle \leq s_1^*$. Let $|\mathcal{W}(t)| = \rho_t^+$ and $|\mathcal{U}(t)| = \rho_t^-$. Then, we have:*

$$\left\langle -\frac{\partial \hat{\mathcal{L}}}{\partial \boldsymbol{w}_\Delta^{(t)}}, \boldsymbol{o}_- \right\rangle \geq \frac{\sigma(p_1)(1 - \sigma(q_1)) r_1^* \cdot \rho_t^-}{\sqrt{m}} - \frac{\sigma(q_1)(1 - \sigma(p_1)) s_1^* \cdot \sqrt{m}}{2} - \mathcal{O}\left(\sqrt{\frac{d \log N}{mN}}\right). \tag{123}$$

Lemma C.6 establishes bounds for the gradient updates of $\boldsymbol{w}_\Delta$ in the directions of irrelevant features.

**Lemma C.6.** *Suppose $p_1 \leq \langle \boldsymbol{w}_\Delta^{(t)}, \boldsymbol{o}_+ \rangle \leq q_1$, $p_1 \leq \langle \boldsymbol{w}_\Delta^{(t)}, \boldsymbol{o}_- \rangle \leq q_1$, $\langle \boldsymbol{w}_\Delta^{(t)}, \boldsymbol{o}_j \rangle \leq q_2$ for $j \neq 1, 2$, and $r_1^* \leq \langle \boldsymbol{W}_{O(i,\cdot)}^{(t)\top}, \boldsymbol{o}_+ \rangle$. Let $\rho_t^+ = |\mathcal{W}(t)|$ and $\rho_t^- = |\mathcal{U}(t)|$. Then we have:*

$$\begin{aligned}
\left\langle -\frac{\partial \hat{\mathcal{L}}}{\partial \boldsymbol{w}_\Delta^{(t)}}, \boldsymbol{o}_j \right\rangle &\leq -\frac{r_1^*}{2\sqrt{m}} \cdot \sigma(p_1)(1 - \sigma(q_1)) \left[(1 - \sigma(q_2))^{\Delta L_{o_+}^+} \rho_t^+ + (1 - \sigma(q_2))^{\Delta L_{o_-}^-} \rho_t^-\right] \\
&\quad + \mathcal{O}\left((1 - \sigma(p_2))^{\Delta L_{o_+}^-}\right) + \mathcal{O}\left((1 - \sigma(p_2))^{\Delta L_{o_-}^+}\right) + \mathcal{O}\left(\sqrt{\frac{d \log N}{mN}}\right).
\end{aligned} \tag{124}$$

## C.2 PROOF OF CONVERGENCE

*Proof of Theorem 2.* Similar to the proof of Theorem 1, the proof starts with the base case at $t = 0$ and proceeds to analyze the training dynamics in a deductive manner, providing additional details in deriving the corresponding convergence and sample complexity bounds.

(**S1**) Warm-up (Base case): Training dynamics at the first iteration $t = 0$.

Recall that we set $\boldsymbol{w}_\Delta^{(0)} = \boldsymbol{0}$. Then, we have

$$\langle \boldsymbol{w}_\Delta^{(0)}, \boldsymbol{o}_+ \rangle = 0, \quad \langle \boldsymbol{w}_\Delta^{(0)}, \boldsymbol{o}_- \rangle = 0, \quad \text{and} \quad \langle \boldsymbol{w}_\Delta^{(0)}, \boldsymbol{o}_j \rangle = 0 \quad \forall j.$$

(**S1.1**) Training dynamics of $\boldsymbol{W}_{O(i,\cdot)}$ at the first iteration $t = 0$.

From Lemma C.1, identify $p_1 = 0$, $q_1 = 0$, $p_2 = 0$ and $q_2 = 0$. Let $\Delta L_{\boldsymbol{o}_+}^+$ and $\Delta L_{\boldsymbol{o}_-}^+$ be the distance between two $\boldsymbol{o}_+$ and $\boldsymbol{o}_-$ features respectively in the positive sample. Similarly, in a negative sample, let the distance between the two $\boldsymbol{o}_+$ tokens as $\Delta L_{\boldsymbol{o}_+}^-$, and the distance between the two $\boldsymbol{o}_-$ tokens as $\Delta L_{\boldsymbol{o}_-}^-$. Then, for any lucky neuron $i \in \mathcal{W}(0)$, we obtain

$$
\begin{aligned}
\frac{c'^2}{2\sqrt{m}L} & \left[ \left(\frac{1}{2}\right)^{\Delta L_{\boldsymbol{o}_+}^+ - 2} - \left(\frac{1}{2}\right)^{\Delta L_{\boldsymbol{o}_+}^- - 2} \right] - \tilde{\mathcal{O}}\left(\frac{1}{\text{poly}(d)}\right) \\
& \leq \left\langle -\frac{\partial \hat{\mathcal{L}}}{\partial \boldsymbol{W}_{O(i,\cdot)}^{(0)}}, \boldsymbol{o}_+ \right\rangle \\
& \leq \frac{1}{2\sqrt{m}L} \left[ \left(\frac{1}{2}\right)^{\Delta L_{\boldsymbol{o}_+}^+} - \left(\frac{1}{2}\right)^{\Delta L_{\boldsymbol{o}_+}^-} \right] + \tilde{\mathcal{O}}\left(\frac{1}{\text{poly}(d)}\right)
\end{aligned}
\tag{125}
$$

$$
\text{and} \quad \left\langle -\frac{\partial \hat{\mathcal{L}}}{\partial \boldsymbol{W}_{O(i,\cdot)}^{(0)}}, \boldsymbol{o}_j \right\rangle \leq \tilde{\mathcal{O}}\left(\frac{1}{\text{poly}(d)}\right) \quad \text{for } j \neq 1.
\tag{126}
$$

Recall that we set the number of samples in a batch $N = \text{poly}(d)$.

Suppose the initialization is

$$
\boldsymbol{W}_{O(i,\cdot)}(0) = \delta_1 \boldsymbol{o}_+ + \delta_2 \boldsymbol{o}_- + \cdots + \delta_d \boldsymbol{o}_d, \quad \delta_j \overset{\text{i.i.d.}}{\sim} \mathcal{N}(0, \xi^2) \quad j = 1, 2, \cdots, d.
\tag{127}
$$

Then, after one gradient descent step, we have

$$
\begin{aligned}
\delta_1 + \frac{\eta c'^2}{2\sqrt{m}L} & \left[ \left(\frac{1}{2}\right)^{\Delta L_{\boldsymbol{o}_+}^+ - 2} - \left(\frac{1}{2}\right)^{\Delta L_{\boldsymbol{o}_+}^- - 2} \right] - \tilde{\mathcal{O}}\left(\frac{1}{\text{poly}(d)}\right) \\
& \leq \left\langle \boldsymbol{W}_{O(i,\cdot)}^{\top}{}^{(1)}, \boldsymbol{o}_+ \right\rangle \\
& \leq \delta_1 + \frac{\eta}{2\sqrt{m}L} \left[ \left(\frac{1}{2}\right)^{\Delta L_{\boldsymbol{o}_+}^+} - \left(\frac{1}{2}\right)^{\Delta L_{\boldsymbol{o}_+}^-} \right] + \tilde{\mathcal{O}}\left(\frac{1}{\text{poly}(d)}\right)
\end{aligned}
\tag{128}
$$

$$
\text{and} \quad \left\langle \boldsymbol{W}_{O(i,\cdot)}^{\top}{}^{(1)}, \boldsymbol{o}_j \right\rangle \leq \tilde{\mathcal{O}}\left(\frac{1}{\text{poly}(d)}\right) \quad \text{for } j \neq 1.
\tag{129}
$$

By applying Lemma C.3, for any lucky neuron $i \in \mathcal{U}(0)$, we obtain

$$
\begin{aligned}
\delta_2 + \frac{\eta c'^2}{2\sqrt{m}L} & \left[ \left(\frac{1}{2}\right)^{\Delta L_{\boldsymbol{o}_-}^- - 2} - \left(\frac{1}{2}\right)^{\Delta L_{\boldsymbol{o}_-}^+ - 2} \right] - \tilde{\mathcal{O}}\left(\frac{1}{\text{poly}(d)}\right) \\
& \leq \left\langle \boldsymbol{W}_{O(i,\cdot)}^{\top}{}^{(1)}, \boldsymbol{o}_+ \right\rangle \\
& \leq \delta_2 + \frac{\eta}{2\sqrt{m}L} \left[ \left(\frac{1}{2}\right)^{\Delta L_{\boldsymbol{o}_-}^-} - \left(\frac{1}{2}\right)^{\Delta L_{\boldsymbol{o}_-}^+} \right] + \tilde{\mathcal{O}}\left(\frac{1}{\text{poly}(d)}\right)
\end{aligned}
\tag{130}
$$

$$
\text{and} \quad \left\langle \boldsymbol{W}_{O(i,\cdot)}^{\top}{}^{(1)}, \boldsymbol{o}_j \right\rangle \leq \tilde{\mathcal{O}}\left(\frac{1}{\text{poly}(d)}\right) \quad \text{for } j \neq 2.
\tag{131}
$$

For any unlucky neuron $i \in \mathcal{K}_- \setminus \mathcal{U}(0)$, Lemma C.4 gives

$$\left\langle \boldsymbol{W}_{O(i,\cdot)}^{\top}{}^{(1)}, \boldsymbol{o}_j \right\rangle \leq \widetilde{\mathcal{O}}\left(\frac{1}{\text{poly}(d)}\right) \quad \text{for } \forall j. \tag{132}$$

(**S1.2**) Training dynamics of $\boldsymbol{w}_\Delta$ at the first iteration $t = 0$.

Now consider the gradient update for $\boldsymbol{w}_\Delta$. Define:

$$a = \delta_1 + \frac{\eta c'^2}{2\sqrt{m}L}\left[\left(\frac{1}{2}\right)^{\Delta L_{o_+}^+ - 2} - \left(\frac{1}{2}\right)^{\Delta L_{o_+}^- - 2}\right] - \widetilde{\mathcal{O}}\left(\frac{1}{\text{poly}(d)}\right)$$

$$b = \delta_1 + \frac{\eta}{2\sqrt{m}L}\left[\left(\frac{1}{2}\right)^{\Delta L_{o_+}^+} - \left(\frac{1}{2}\right)^{\Delta L_{o_+}^-}\right] + \widetilde{\mathcal{O}}\left(\frac{1}{\text{poly}(d)}\right)$$

Applying Lemma C.5 with $p_1 = 0$, $q_1 = 0$, $r_1^* = a$, $s_1^* = b$, and $\rho_0^+ = |\mathcal{W}(0)|$, we get

$$\left\langle -\frac{\partial \hat{\mathcal{L}}}{\partial \boldsymbol{w}_\Delta^{(0)}}, \boldsymbol{o}_+ \right\rangle \geq \frac{a}{4\sqrt{m}} \cdot \rho_0^+ - \frac{b\sqrt{m}}{8} - \widetilde{\mathcal{O}}\left(\frac{1}{\text{poly}(d)}\right) \tag{133}$$

We can relax this lower bound and obtain

$$\left\langle -\frac{\partial \hat{\mathcal{L}}}{\partial \boldsymbol{w}_\Delta^{(0)}}, \boldsymbol{o}_+ \right\rangle \geq \frac{c'}{4}\left[\frac{2a}{\sqrt{m}} \cdot \rho_0^+ - \sqrt{m}b\right] - \widetilde{\mathcal{O}}\left(\frac{1}{\text{poly}(d)}\right) =: \alpha \tag{134}$$

Recall that $\delta_1 = \frac{1}{\text{poly}(d)}$. Since $a - b = \widetilde{\mathcal{O}}\left(\frac{1}{\text{poly}(d)}\right)$ that is sufficiently small,

$$\begin{aligned}
\alpha &= \frac{c'}{4}\left[\frac{2a}{\sqrt{m}} \cdot \frac{m}{2} - \sqrt{m}b\right] - \widetilde{\mathcal{O}}\left(\frac{1}{\text{poly}(d)}\right) \\
&= \frac{c'}{4}\left[\sqrt{m}a - \sqrt{m}a\right] - \widetilde{\mathcal{O}}\left(\frac{1}{\text{poly}(d)}\right) \\
&= 0 - \widetilde{\mathcal{O}}\left(\frac{1}{\text{poly}(d)}\right) \\
&= -\widetilde{\mathcal{O}}\left(\frac{1}{\text{poly}(d)}\right) \approx 0
\end{aligned} \tag{135}$$

From Lemma C.6, we also obtain

$$\left\langle -\frac{\partial \hat{\mathcal{L}}}{\partial \boldsymbol{w}_\Delta^{(0)}}, \boldsymbol{o}_j \right\rangle \leq \frac{-a}{8\sqrt{m}}\left[\left(\frac{1}{2}\right)^{\Delta L_{o_+}^+} \cdot \rho_0^+ + \left(\frac{1}{2}\right)^{\Delta L_{o_-}^-} \cdot \rho_0^-\right] \tag{136}$$

where we apply the lemma with the values

$$p_1 = 0, \quad q_1 = 0, \quad q_2 = 0, \quad \text{and} \quad r_1^* = a.$$

We can relax this upper bound and obtain

$$\left\langle -\frac{\partial \hat{\mathcal{L}}}{\partial \boldsymbol{w}_\Delta^{(0)}}, \boldsymbol{o}_j \right\rangle \leq \frac{-ac'}{4\sqrt{m}}\left[\left(\frac{1}{2}\right)^{\Delta L_{o_+}^+} \cdot \rho_0^+ + \left(\frac{1}{2}\right)^{\Delta L_{o_-}^-} \cdot \rho_0^-\right] =: \gamma \tag{137}$$

Taking $\rho_0^+ = \rho_0^- = \frac{m}{2} + \widetilde{\mathcal{O}}\left(\frac{1}{\text{poly}(d)}\right)$, we can simplify and write

$$
\begin{aligned}
\gamma &= \frac{-ac'}{4\sqrt{m}} \cdot \frac{m}{2} \left[ \left(\frac{1}{2}\right)^{\Delta L_{o_+}^+} + \left(\frac{1}{2}\right)^{\Delta L_{o_-}^-} \right] \\
&= -\sqrt{m}a \cdot \frac{c'}{8} \left[ \left(\frac{1}{2}\right)^{\Delta L_{o_+}^+} + \left(\frac{1}{2}\right)^{\Delta L_{o_-}^-} \right] \\
&= \frac{-\eta c'^3}{16L} \left[ \left(\frac{1}{2}\right)^{\Delta L_{o_+}^+ - 2} - \left(\frac{1}{2}\right)^{\Delta L_{o_+}^- - 2} \right] \left[ \left(\frac{1}{2}\right)^{\Delta L_{o_+}^+} + \left(\frac{1}{2}\right)^{\Delta L_{o_-}^-} \right] \\
&\quad - \widetilde{\mathcal{O}}\left(\frac{1}{\text{poly}(d)}\right).
\end{aligned}
\tag{138}
$$

(**S2**) Induction Step: Training dynamics at a general iteration $t$.

Let $\langle \boldsymbol{w}_\Delta^{(t)}, \boldsymbol{o}_+ \rangle = \alpha^* \geq \alpha \cdot t$, $\langle \boldsymbol{w}_\Delta^{(t)}, \boldsymbol{o}_- \rangle = \beta^* \geq \beta \cdot t$, and $\langle \boldsymbol{w}_\Delta^{(t)}, \boldsymbol{o}_j \rangle = \gamma^* \leq \gamma \cdot t$, where

$$
\beta = \frac{c'}{4} \left[ \frac{2a'}{\sqrt{m}} \cdot \rho_0^- - \sqrt{m}b' \right] - \widetilde{\mathcal{O}}\left(\frac{1}{\text{poly}(d)}\right) > 0
\tag{139}
$$

$$
a' = \delta_2 + \frac{\eta c'^2}{2\sqrt{m}L} \left[ \left(\frac{1}{2}\right)^{\Delta L_{o_-}^- - 2} - \left(\frac{1}{2}\right)^{\Delta L_{o_-}^+ - 2} \right] - \widetilde{\mathcal{O}}\left(\frac{1}{\text{poly}(d)}\right)
$$

$$
b' = \delta_2 + \frac{\eta}{2\sqrt{m}L} \left[ \left(\frac{1}{2}\right)^{\Delta L_{o_-}^-} - \left(\frac{1}{2}\right)^{\Delta L_{o_-}^+} \right] + \widetilde{\mathcal{O}}\left(\frac{1}{\text{poly}(d)}\right)
$$

Following the same approach as in (135), we can simplify and obtain

$$
\beta = -\widetilde{\mathcal{O}}\left(\frac{1}{\text{poly}(d)}\right).
\tag{140}
$$

For any lucky neuron $i \in \mathcal{W}(t)$ at the $(t+1)$-th iteration, we have

$$
\begin{aligned}
&\frac{c'^2}{\sqrt{m}L} \cdot \sigma(\alpha^*) \left[ (1 - \sigma(\gamma^*))^{\Delta L_{o_+}^+ - 2} - (1 - \sigma(\gamma^*))^{\Delta L_{o_+}^- - 2} \right] - \widetilde{\mathcal{O}}\left(\frac{1}{\text{poly}(d)}\right) \\
&\leq \left\langle -\frac{\partial \hat{\mathcal{L}}}{\partial \boldsymbol{W}_{O(i,\cdot)}^{(t)}}, \boldsymbol{o}_+ \right\rangle \\
&\leq \frac{1}{\sqrt{m}L} \cdot \sigma(\alpha^*) \cdot (1 - \sigma(\alpha^*))^2 \left[ (1 - \sigma(\gamma^*))^{\Delta L_{o_+}^+ - 2} - (1 - \sigma(\gamma^*))^{\Delta L_{o_+}^- - 2} \right] \\
&\quad + \widetilde{\mathcal{O}}\left(\frac{1}{\text{poly}(d)}\right)
\end{aligned}
\tag{141}
$$

$$
\left\langle -\frac{\partial \hat{\mathcal{L}}}{\partial \boldsymbol{W}_{O(i,\cdot)}^{(1)}}, \boldsymbol{o}_j \right\rangle \leq \widetilde{\mathcal{O}}\left(\frac{1}{\text{poly}(d)}\right) \quad \text{for } j \neq 1
\tag{142}
$$

Note that $\sigma(\alpha^*) > \frac{1}{2}$ and $\sigma(\gamma^*) < \frac{1}{2}$ when $t = 1$. By a simple induction, this further ensures

$$
\left\langle -\frac{\partial \hat{\mathcal{L}}}{\partial \boldsymbol{W}_{O(i,\cdot)}^{(0)}}, \boldsymbol{o}_+ \right\rangle \leq \left\langle -\frac{\partial \hat{\mathcal{L}}}{\partial \boldsymbol{W}_{O(i,\cdot)}^{(1)}}, \boldsymbol{o}_+ \right\rangle \leq \cdots \left\langle -\frac{\partial \hat{\mathcal{L}}}{\partial \boldsymbol{W}_{O(i,\cdot)}^{(t)}}, \boldsymbol{o}_+ \right\rangle \leq \left\langle -\frac{\partial \hat{\mathcal{L}}}{\partial \boldsymbol{W}_{O(i,\cdot)}^{(t+1)}}, \boldsymbol{o}_+ \right\rangle.
\tag{143}
$$

Thus, we obtain the following bound after the second gradient descent step.

$$
\begin{aligned}
&\left\langle (\boldsymbol{W}_{O(i,\cdot)}^{(2)})^\top, \boldsymbol{o}_+ \right\rangle \\
&\geq \delta_1 + \frac{\eta c'^2}{\sqrt{mL}} \left[ \left(\frac{1}{2}\right)^{\Delta L_{o_+}^+ - 1} - \left(\frac{1}{2}\right)^{\Delta L_{o_+}^- - 1} \right. \\
&\quad \left. + \sigma(\alpha^*) \left( (1 - \sigma(\gamma^*))^{\Delta L_{o_+}^+ - 2} - (1 - \sigma(\gamma^*))^{\Delta L_{o_+}^- - 2} \right) \right] - \tilde{\mathcal{O}}\left(\frac{1}{\mathrm{poly}(d)}\right).
\end{aligned}
\tag{144}
$$

and

$$
\begin{aligned}
&\left\langle (\boldsymbol{W}_{O(i,\cdot)}^{(2)})^\top, \boldsymbol{o}_+ \right\rangle \\
&\leq \delta_1 + \frac{\eta}{2\sqrt{mL}} \left[ \left(\frac{1}{2}\right)^{\Delta L_{o_-}^-} - \left(\frac{1}{2}\right)^{\Delta L_{o_-}^+} \right. \\
&\quad \left. + 2\sigma(\alpha^*) \cdot (1 - \sigma(\alpha^*))^2 \left( (1 - \sigma(\gamma^*))^{\Delta L_{o_+}^+ - 2} - (1 - \sigma(\gamma^*))^{\Delta L_{o_+}^- - 2} \right) \right] + \tilde{\mathcal{O}}\left(\frac{1}{\mathrm{poly}(d)}\right).
\end{aligned}
\tag{145}
$$

For the convenience of presentation, we use $u$ to denote the lower bound in (144), and $v$ to denote the upper bound in (145).

Similarly, applying Lemma C.3 to any lucky neuron $i \in \mathcal{U}(1)$ at iteration 2, we get

$$
\begin{aligned}
\left\langle -\frac{\partial \hat{\mathcal{L}}}{\partial \boldsymbol{W}_{O(i,\cdot)}^{(1)}}, \boldsymbol{o}_- \right\rangle &\geq \frac{c'^2}{\sqrt{mL}} \cdot \sigma(\beta^*) \left[ (1 - \sigma(\gamma^*))^{\Delta L_{o_-}^- - 2} - (1 - \sigma(\gamma^*))^{\Delta L_{o_-}^+ - 2} \right] \\
&\quad - \tilde{\mathcal{O}}\left(\frac{1}{\mathrm{poly}(d)}\right),
\end{aligned}
\tag{146}
$$

$$
\begin{aligned}
\left\langle -\frac{\partial \hat{\mathcal{L}}}{\partial \boldsymbol{W}_{O(i,\cdot)}^{(1)}}, \boldsymbol{o}_- \right\rangle &\leq \frac{1}{\sqrt{mL}} \cdot \sigma(\beta^*) \cdot (1 - \sigma(\beta^*))^2 \left[ (1 - \sigma(\gamma^*))^{\Delta L_{o_-}^- - 2} - (1 - \sigma(\gamma^*))^{\Delta L_{o_-}^+ - 2} \right] \\
&\quad + \tilde{\mathcal{O}}\left(\frac{1}{\mathrm{poly}(d)}\right),
\end{aligned}
\tag{147}
$$

and

$$
\left\langle -\frac{\partial \hat{\mathcal{L}}}{\partial \boldsymbol{W}_{O(i,\cdot)}^{(1)}}, \boldsymbol{o}_j \right\rangle \leq \tilde{\mathcal{O}}\left(\frac{1}{\mathrm{poly}(d)}\right) \quad \text{for } j \neq 1
\tag{148}
$$

Applying Lemma C.5 with $p_1 = \alpha^*$, $q_1 = \alpha^*$, $r_1^* = u$, and $s_1^* = v$, we obtain

$$
\left\langle -\frac{\partial \hat{\mathcal{L}}}{\partial \boldsymbol{w}_\Delta^{(1)}}, \boldsymbol{o}_+ \right\rangle \geq \frac{\sigma(\alpha^*)c'}{2} \left[ \frac{2u}{\sqrt{m}} \cdot \rho_1^+ - \sqrt{m}v \right] - \tilde{\mathcal{O}}\left(\frac{1}{\mathrm{poly}(d)}\right) =: \chi
\tag{149}
$$

Since $\rho_1^+ = \frac{m}{2}$ and $u - v = \tilde{\mathcal{O}}\left(\frac{1}{\mathrm{poly}(d)}\right)$ that is sufficiently small, we have $\chi \approx 0$.

By applying Lemma C.6 with

$$
p_1 = \alpha^*(= \beta^*), \quad q_1 = \alpha^*(= \beta^*), \quad q_2 = \gamma^*, \text{and} \quad r_1^* = u, \text{we have}
$$

$$
\left\langle -\frac{\partial \hat{\mathcal{L}}}{\partial \boldsymbol{w}_\Delta^{(t)}}, \boldsymbol{o}_j \right\rangle \leq -\frac{c'u}{2\sqrt{m}} \sigma(\alpha^*) \left[ (1 - \sigma(\gamma^*))^{\Delta L_{o_+}^+} \rho_t^+ + (1 - \sigma(\gamma^*))^{\Delta L_{o_-}^-} \rho_t^- \right] =: \iota
\tag{150}
$$

Note that here we assumed the distribution of $\Delta L^+$ is identical to $\Delta L^-$ to have $\alpha^* = \beta^*$.

(**S3**) Induction conclusion: Training dynamics when the algorithm ends.

We proceed by induction on $t$: the base case $t = 0$ is established in (S1), and the induction step for general $t$ is shown in (S2). For, any lucky neuron $i \in \mathcal{W}(T)$, we obtain

$$\left\langle \boldsymbol{W}_{O(i,\cdot)}^{\top}{}^{(T)}, \boldsymbol{o}_+ \right\rangle \geq aT \tag{151}$$

$$\left\langle \boldsymbol{W}_{O(i,\cdot)}^{\top}{}^{(T)}, \boldsymbol{o}_j \right\rangle \leq \widetilde{\mathcal{O}}\left(\frac{1}{\text{poly}(d)}\right) \quad \text{for } j \neq 1 \tag{152}$$

For any lucky neuron $i \in \mathcal{U}(T)$, we obtain

$$\left\langle \boldsymbol{W}_{O(i,\cdot)}^{\top}{}^{(T)}, \boldsymbol{o}_- \right\rangle \geq aT \tag{153}$$

$$\left\langle \boldsymbol{W}_{O(i,\cdot)}^{\top}{}^{(T)}, \boldsymbol{o}_j \right\rangle \leq \widetilde{\mathcal{O}}\left(\frac{1}{\text{poly}(d)}\right) \quad \text{for } j \neq 2 \tag{154}$$

Also, we obtain

$$\left\langle \boldsymbol{w}_{\Delta}^{(T)}, \boldsymbol{o}_+ \right\rangle \geq \alpha T, \tag{155}$$

$$\left\langle \boldsymbol{w}_{\Delta}^{(T)}, \boldsymbol{o}_- \right\rangle \geq \beta T, \tag{156}$$

$$\text{and} \quad \left\langle \boldsymbol{w}_{\Delta}^{(T)}, \boldsymbol{o}_j \right\rangle \leq \gamma T. \tag{157}$$

(**S4**) Derivation for the generalization bound.

We will demonstrate that once the weights have converged at iteration $T$, the model accurately captures the underlying data distribution, which leads to zero generalization error, as shown in (181).

Consider $z^{(n)} = +1$ as an example. The sequence $\boldsymbol{X}^{(n)} = \begin{bmatrix} \boldsymbol{x}_1^{(n)} & \boldsymbol{x}_2^{(n)} & \cdots & \boldsymbol{x}_L^{(n)} \end{bmatrix}$ contains two $\boldsymbol{o}_+$ at $L_1^+$ and $L_2^+$ and two $\boldsymbol{o}_-$ at $L_1^-$ and $L_2^-$.

$$
\begin{aligned}
F(\boldsymbol{X}^{(n)}) &= \frac{1}{L} \sum_{l=1}^{L} \sum_{i=1}^{m} v_i \, \phi\left(\boldsymbol{W}_{O(i,\cdot)} \boldsymbol{y}_l^{(n)}\right) \\
&= \frac{1}{\sqrt{mL}} \sum_{i \in \mathcal{K}^+} \sum_{l=1}^{L} \phi\left(\boldsymbol{W}_{O(i,\cdot)} \boldsymbol{y}_l^{(n)}\right) - \frac{1}{\sqrt{mL}} \sum_{i \in \mathcal{K}^-} \sum_{l=1}^{L} \phi\left(\boldsymbol{W}_{O(i,\cdot)} \boldsymbol{y}_l^{(n)}\right) \\
&\geq \frac{1}{\sqrt{mL}} \sum_{i \in \mathcal{W}(0)} \sum_{l=1}^{L} \phi\left(\boldsymbol{W}_{O(i,\cdot)} \boldsymbol{y}_l^{(n)}\right) - \frac{1}{\sqrt{mL}} \sum_{i \in \mathcal{U}(0)} \sum_{l=1}^{L} \phi\left(\boldsymbol{W}_{O(i,\cdot)} \boldsymbol{y}_l^{(n)}\right) \\
&\quad - \frac{1}{\sqrt{mL}} \sum_{i \in \mathcal{K}^- \setminus \mathcal{U}(0)} \sum_{l=1}^{L} \phi\left(\boldsymbol{W}_{O(i,\cdot)} \boldsymbol{y}_l^{(n)}\right)
\end{aligned} \tag{158}
$$

The Mamba output $\boldsymbol{y}_l^{(n)}$ is defined as

$$\boldsymbol{y}_l^{(n)} = \sum_{s=1}^{l} \left( \prod_{j=s+1}^{l} \left(1 - \sigma(\boldsymbol{w}_{\Delta}^{\top} \boldsymbol{x}_j^{(n)})\right) \right) \cdot \sigma(\boldsymbol{w}_{\Delta}^{\top} \boldsymbol{x}_s^{(n)}) \cdot (\boldsymbol{x}_s^{(n)\top} \boldsymbol{x}_l^{(n)}) \boldsymbol{x}_s^{(n)}. \tag{159}$$

We now derive a lower bound for

$$\sum_{i \in \mathcal{W}(0)} \sum_{l=1}^{L} \phi(\boldsymbol{W}_{O(i,\cdot)} \boldsymbol{y}_l).$$

To that end, consider the aggregated projection

$$\sum_{i \in \mathcal{W}(0)} \sum_{l=1}^{L} \boldsymbol{W}_{O(i,\cdot)} \boldsymbol{y}_l = \sum_{i \in \mathcal{W}(0)} \sum_{l=1}^{L} \sum_{j=1}^{d} \langle \boldsymbol{W}_{O(i,\cdot)}^{\top}, \boldsymbol{o}_j \rangle \cdot \langle \boldsymbol{y}_l, \boldsymbol{o}_j \rangle. \tag{160}$$

For any $i \in \mathcal{W}(0)$, we know that

$$\langle \boldsymbol{W}_{O(i,\cdot)}^{\top}, \boldsymbol{o}_+ \rangle \geq aT. \tag{161}$$

Hence, let's obtain a lower bound for $\langle \boldsymbol{y}_l, \boldsymbol{o}_+ \rangle$

We only need to consider the cases where $\boldsymbol{x}_s = \boldsymbol{o}_+$ for some $s$ in the range $1 \leq s \leq l$. In particular, we will focus on the following instances:

$$s = L_1^+ \text{ and } l \in \{L_1^+, L_2^+\}, \qquad s = L_2^+ \text{ and } l = L_2^+.$$

After $T$ iterations, we know

$$\langle \boldsymbol{w}_\Delta, \boldsymbol{o}_+ \rangle \geq \alpha T, \quad \langle \boldsymbol{w}_\Delta, \boldsymbol{o}_- \rangle \geq \beta T, \quad \langle \boldsymbol{w}_\Delta, \boldsymbol{o}_j \rangle \leq \gamma T \quad \text{for } j \neq 1, 2. \tag{162}$$

Therefore,

$$\left\langle \boldsymbol{y}_{L_1^+}, \boldsymbol{o}_+ \right\rangle = \sigma\left(\langle \boldsymbol{w}_\Delta, \boldsymbol{o}_+ \rangle\right) \geq \sigma(\alpha T). \tag{163}$$

We have,

$$\langle \boldsymbol{w}_\Delta, \boldsymbol{o}_+ \rangle \leq W_1 T, \qquad \langle \boldsymbol{w}_\Delta, \boldsymbol{o}_- \rangle \leq W_2 T,$$

where

$$W_1 = \tilde{\mathcal{O}}\left(\frac{1}{\mathrm{poly}(d)}\right), \tag{164}$$

$$W_2 = \tilde{\mathcal{O}}\left(\frac{1}{\mathrm{poly}(d)}\right). \tag{165}$$

Then we obtain the following:

$$\left\langle \boldsymbol{y}_{L_2^+}, \boldsymbol{o}_+ \right\rangle \geq \sigma(\alpha T) + (1 - \sigma(W_1 T))(1 - \sigma(W_2 T))(1 - \sigma(\langle \boldsymbol{w}_\Delta, \boldsymbol{o}_j \rangle))^{\Delta L_{o_+}^+ - 2} \cdot \sigma(\alpha T)$$

$$= \sigma(\alpha T)\left[1 + (1 - \sigma(W_1 T))(1 - \sigma(W_2 T))(1 - \sigma(\langle \boldsymbol{w}_\Delta, \boldsymbol{o}_j \rangle))^{\Delta L_{o_+}^+ - 2}\right]. \tag{166}$$

We now lower bound the objective

$$\sum_{i \in \mathcal{W}(0)} \sum_{l=1}^{L} \phi(\boldsymbol{W}_{O(i,\cdot)} \boldsymbol{y}_l).$$

We begin with

$$\sum_{i \in \mathcal{W}(0)} \sum_{l=1}^{L} \phi(\boldsymbol{W}_{O(i,\cdot)} \boldsymbol{y}_l) \geq \sum_{i \in \mathcal{W}(0)} \left[\phi(\boldsymbol{W}_{O(i,\cdot)} \boldsymbol{y}_{L_1^+}) + \phi(\boldsymbol{W}_{O(i,\cdot)} \boldsymbol{y}_{L_2^+})\right].$$

Note that

$$\boldsymbol{W}_{O(i,\cdot)} \boldsymbol{y}_{L_1^+} = \sum_{j=1}^{d} \left\langle \boldsymbol{W}_{O(i,\cdot)}^{\top}, \boldsymbol{o}_j \right\rangle \left\langle \boldsymbol{y}_{L_1^+}, \boldsymbol{o}_j \right\rangle,$$

and $\boldsymbol{y}_{L_1^+}$ has only $\boldsymbol{o}_+$ component.

Therefore,

$$\boldsymbol{W}_{O(i,\cdot)}\boldsymbol{y}_{L_1^+} = \left\langle \boldsymbol{W}_{O(i,\cdot)}^{\top}, \boldsymbol{o}_+ \right\rangle \left\langle \boldsymbol{y}_{L_1^+}, \boldsymbol{o}_+ \right\rangle \ge aT \cdot \sigma(\alpha T) > 0.$$

Similarly, we can write

$$\boldsymbol{W}_{O(i,\cdot)}\boldsymbol{y}_{L_2^+} \ge aT \cdot \sigma(\alpha T) \left[1 + (1 - \sigma(W_1 T))(1 - \sigma(W_2 T))(1 - \sigma(\langle \boldsymbol{w}_\Delta, \boldsymbol{o}_j \rangle))^{\Delta L_{o_+}^+ - 2}\right] > 0.$$

Applying $\phi(z) = z$ for positive $z$, we obtain

$$\phi(\boldsymbol{W}_{O(i,\cdot)}\boldsymbol{y}_{L_1^+}) \ge aT \cdot \sigma(\alpha T),$$

$$\phi(\boldsymbol{W}_{O(i,\cdot)}\boldsymbol{y}_{L_2^+}) \ge aT \cdot \sigma(\alpha T) \left[1 + (1 - \sigma(W_1 T))(1 - \sigma(W_2 T))(1 - \sigma(\langle \boldsymbol{w}_\Delta, \boldsymbol{o}_j \rangle))^{\Delta L_{o_+}^+ - 2}\right].$$

Hence,

$$\sum_{i \in \mathcal{W}(0)} \sum_{l=1}^{L} \phi(\boldsymbol{W}_{O(i,\cdot)}\boldsymbol{y}_l) \ge \sum_{i \in \mathcal{W}(0)} aT \cdot \sigma(\alpha T) \cdot \tag{167}$$
$$\left[2 + (1 - \sigma(W_1 T))(1 - \sigma(W_2 T))(1 - \sigma(\langle \boldsymbol{w}_\Delta, \boldsymbol{o}_j \rangle))^{\Delta L_{o_+}^+ - 2}\right].$$

Next, we derive an upper bound for

$$\sum_{i \in \mathcal{U}(0)} \sum_{l=1}^{L} \phi\left(\boldsymbol{W}_{O(i,\cdot)}\boldsymbol{y}_l^{(n)}\right).$$

For any $i \in \mathcal{U}(0)$, we know that

$$0 < \langle \boldsymbol{W}_{O(i,\cdot)}^{\top}, \boldsymbol{o}_- \rangle \le bT. \tag{168}$$

We now derive an upper bound for $\langle \boldsymbol{y}_l, \boldsymbol{o}_- \rangle$. We need to focus on the following instances:

$$s = L_1^- \text{ and } l \in \{L_1^-, L_2^-\}, \qquad s = L_2^- \text{ and } l = L_2^-.$$

$$\left\langle \boldsymbol{y}_{L_1^-}, \boldsymbol{o}_- \right\rangle = \sigma\left(\langle \boldsymbol{w}_\Delta, \boldsymbol{o}_- \rangle\right) \le \sigma(W_2 T). \tag{169}$$

$$\left\langle \boldsymbol{y}_{L_2^-}, \boldsymbol{o}_- \right\rangle \le \sigma(W_2 T) + (1 - \sigma(\alpha T))(1 - \sigma(\beta T))(1 - \sigma(\langle \boldsymbol{w}_\Delta, \boldsymbol{o}_j \rangle))^{\Delta L_{o_-}^+ - 2} \cdot \sigma(W_2 T)$$
$$= \sigma(W_2 T) \left[1 + (1 - \sigma(\alpha T))(1 - \sigma(\beta T))(1 - \sigma(\langle \boldsymbol{w}_\Delta, \boldsymbol{o}_j \rangle))^{\Delta L_{o_-}^+ - 2}\right]. \tag{170}$$

Hence,

$$\sum_{i \in \mathcal{U}(0)} \sum_{l=1}^{L} \phi(\boldsymbol{W}_{O(i,\cdot)}\boldsymbol{y}_l) \le \sum_{i \in \mathcal{U}(0)} bT \cdot \sigma(W_2 T) \cdot$$
$$\left[2 + (1 - \sigma(\alpha T))(1 - \sigma(\beta T))(1 - \sigma(\langle \boldsymbol{w}_\Delta, \boldsymbol{o}_j \rangle))^{\Delta L_{o_-}^+ - 2}\right].$$

In addition, we have

$$\sum_{i \in \mathcal{K}^- \setminus \mathcal{U}(0)} \sum_{l=1}^{L} \phi\left(\boldsymbol{W}_{O(i,\cdot)}\boldsymbol{y}_l^{(n)}\right) \le \tilde{\mathcal{O}}\left(\frac{1}{\text{poly}(d)}\right). \tag{171}$$

By (158), we can write

$$F(\boldsymbol{X}^{(n)}) \geq \frac{1}{\sqrt{mL}} \left\{ \frac{m}{2} \cdot aT \cdot \sigma(\alpha T) \left[ 2 + (1 - \sigma(W_1 T))(1 - \sigma(W_2 T))(1 - \sigma(\langle \boldsymbol{w}_\Delta, \boldsymbol{o}_j \rangle))^{\Delta L_{o_+}^+ - 2} \right] \right.$$

$$- \frac{m}{2} \cdot bT \cdot \sigma(W_2 T) \left[ 2 + (1 - \sigma(\alpha T)) (1 - \sigma(\beta T)) (1 - \sigma(\langle \boldsymbol{w}_\Delta, \boldsymbol{o}_j \rangle))^{\Delta L_{o_-}^+ - 2} \right]$$

$$\left. - \tilde{\mathcal{O}} \left( \frac{1}{\text{poly}(d)} \right) \right\},$$

$$(172)$$

with

$$a = \frac{\eta c'^2}{2\sqrt{mL}} \left[ \left( \frac{1}{2} \right)^{\Delta L_{o_+}^+ - 2} - \left( \frac{1}{2} \right)^{\Delta L_{o_+}^- - 2} \right] - \tilde{\mathcal{O}} \left( \frac{1}{\text{poly}(d)} \right), \qquad (173)$$

$$\text{and} \quad b = \frac{\eta}{2\sqrt{mL}} \left[ \left( \frac{1}{2} \right)^{\Delta L_{o_+}^+} - \left( \frac{1}{2} \right)^{\Delta L_{o_+}^-} \right] + \tilde{\mathcal{O}} \left( \frac{1}{\text{poly}(d)} \right). \qquad (174)$$

$$\alpha = \frac{c'}{4} \left[ \frac{2a}{\sqrt{m}} \cdot \rho_0^+ - \sqrt{m} b \right] - \tilde{\mathcal{O}} \left( \frac{1}{\text{poly}(d)} \right)$$

$$= -\tilde{\mathcal{O}} \left( \frac{1}{\text{poly}(d)} \right) \qquad (175)$$

Therefore, we conclude that

$$F(\boldsymbol{X}^{(n)}) \geq \frac{1}{\sqrt{mL}} \left\{ \frac{m}{2} \cdot aT \cdot \sigma(\alpha T)(1 - \sigma(W_1 T))(1 - \sigma(W_2 T)) \right.$$

$$\left. \left[ (1 - \sigma(\langle \boldsymbol{w}_\Delta, \boldsymbol{o}_j \rangle))^{\Delta L_{o_+}^+ - 2} - (1 - \sigma(\langle \boldsymbol{w}_\Delta, \boldsymbol{o}_j \rangle))^{\Delta L_{o_-}^+ - 2} \right] \right\} - \tilde{\mathcal{O}} \left( \frac{1}{\text{poly}(d)} \right) \quad (176)$$

If we can show $\left[ (1 - \sigma(\langle \boldsymbol{w}_\Delta, \boldsymbol{o}_j \rangle))^{\Delta L_{o_+}^+ - 2} - (1 - \sigma(\langle \boldsymbol{w}_\Delta, \boldsymbol{o}_j \rangle))^{\Delta L_{o_-}^+ - 2} \right] > 0$, then we can prove $F(\boldsymbol{X}^{(n)}) \geq C$ for some positive constant $C$.

First define a random variable $\psi_1 = \langle \boldsymbol{w}_\Delta, \boldsymbol{o}_j \rangle$. Then, we have from the definition of our locality-structured data type

$$\mathbb{E}_n \left[ (1 - \sigma(\psi_1))^{\Delta L_{o_+}^+ - 2} - (1 - \sigma(\psi_1))^{\Delta L_{o_-}^+ - 2} \right] = k' > 0 \qquad (177)$$

for some positive constant $k'$.

The random variable $\psi_2 = (1 - \sigma(\psi_1))^{\Delta L_{o_+}^+ - 2} - (1 - \sigma(\psi_1))^{\Delta L_{o_-}^+ - 2}$ is bounded above by 1.

Applying Hoeffding's bound, for any $q > 0$,

$$\mathbb{P} \left( |\psi_2 - \mathbb{E}\psi_2| \gtrsim \sqrt{\frac{q \log N}{N}} \right) \leq N^{-q}. \qquad (178)$$

From this we can conclude that,

$$\psi_2 = \left[ (1 - \sigma(\langle \boldsymbol{w}_\Delta, \boldsymbol{o}_j \rangle))^{\Delta L_{o_+}^+ - 2} - (1 - \sigma(\langle \boldsymbol{w}_\Delta, \boldsymbol{o}_j \rangle))^{\Delta L_{o_-}^+ - 2} \right] \geq k' - \mathcal{O} \left( \sqrt{\frac{q \log N}{N}} \right),$$

$$(179)$$

with probability at most $N^{-q}$.

Hence, for sufficiently large N, we have from (177)

$$\left[(1 - \sigma(\langle \boldsymbol{w}_\Delta, \boldsymbol{o}_j \rangle))^{\Delta L_{o_+}^+ - 2} - (1 - \sigma(\langle \boldsymbol{w}_\Delta, \boldsymbol{o}_j \rangle))^{\Delta L_{o_-}^+ - 2}\right] > 0 \tag{180}$$

Therefore,

$$F(\boldsymbol{X}^{(n)}) \geq C, \text{ where } C \text{ is some positive constant.} \tag{181}$$

Similarly, for a negative sample, one can show by symmetry that the model output satisfies $F(\boldsymbol{X}^{(n)}) \leq -C$.

(**S4.1**) Derivation for the convergence rate.

Let's find the number of iterations $T$ required such that $F(\boldsymbol{X}^{(n)}) \geq 1$, since the label is $+1$. We require

$$\frac{1}{\sqrt{mL}} \cdot \frac{m}{2} \cdot aT \cdot \sigma(\alpha T) \geq 1 + \epsilon. \tag{182}$$

Substituting the value of $a = \frac{\eta}{2\sqrt{mL}}\left[\left(\frac{1}{2}\right)^{\Delta L_{o_+}^+} - \left(\frac{1}{2}\right)^{\Delta L_{o_+}^-}\right]$ and $\sigma(\alpha T) \approx \frac{1}{2}$ since $\alpha \approx 0$, the condition becomes

$$\frac{\sqrt{m}aT}{4L} = \frac{\sqrt{m}}{4L} \cdot \frac{\eta}{2\sqrt{mL}}\left[\left(\frac{1}{2}\right)^{\Delta L_{o_+}^+} - \left(\frac{1}{2}\right)^{\Delta L_{o_+}^-}\right]T$$

$$= \frac{\eta T}{8L^2}\left[\left(\frac{1}{2}\right)^{\Delta L_{o_+}^+} - \left(\frac{1}{2}\right)^{\Delta L_{o_+}^-}\right] \geq 1 + \epsilon. \tag{183}$$

Solving for $T$, we obtain

$$T \geq \frac{8L^2(1 + \epsilon)}{\eta\left[\left(\frac{1}{2}\right)^{\Delta L_{o_+}^+} - \left(\frac{1}{2}\right)^{\Delta L_{o_+}^-}\right]} \geq \frac{8L^2}{\eta\left[\left(\frac{1}{2}\right)^{\Delta L_{o_+}^+} - \left(\frac{1}{2}\right)^{\Delta L_{o_+}^-}\right]}. \tag{184}$$

By combining (182) with the expression for the model output $F(\boldsymbol{X}^{(n)})$ in (176), we obtain

$$F(\boldsymbol{X}^{(n)}) \geq (1 + \epsilon) \tag{185}$$

Hence, the model output satisfies $F(\boldsymbol{X}^{(n)}) \geq 1$.

(**S4.2**) Derivation for the sample complexity.

Now we derive a sample-complexity bound that guarantees zero generalization error.

Assuming enough samples, we can write for sufficiently small $\lambda \ll 1$

$$\mathcal{O}\left(\sqrt{\frac{d \log N}{mN}}\right) \leq \lambda \cdot \frac{\eta}{2\sqrt{mL}}\left[\left(\frac{1}{2}\right)^{\Delta L_{o_+}^+} - \left(\frac{1}{2}\right)^{\Delta L_{o_+}^-}\right]. \tag{186}$$

From this we can derive a lower bound on the required sample size,

$$N \geq \Omega\left(\lambda^{-2} \cdot \frac{4L^2 d}{\eta^2\left[\left(\frac{1}{2}\right)^{\Delta L_{o_+}^+} - \left(\frac{1}{2}\right)^{\Delta L_{o_+}^-}\right]^2}\right)$$

$$\geq \Omega\left(\frac{L^2 d}{\eta^2\left[\left(\frac{1}{2}\right)^{\Delta L_{o_+}^+} - \left(\frac{1}{2}\right)^{\Delta L_{o_+}^-}\right]^2}\right), \tag{187}$$

which will be (19) in Theorem 2.

$\square$

# D   PROOF OF LEMMAS IN APPENDIX B

Please refer to the supplementary material or a preliminary work at (Shandirasegaran et al., 2025) for this section. We defer all proofs to the supplementary material, as the high-level ideas underlying the lemmas overlap with those presented in Appendix C for locality data. However, the case of locality-structured data presents additional challenges. Appendix E provides the complete proofs for the locality-structured data, which contain the main technical ideas.

# E   PROOF OF LEMMAS IN APPENDIX C

## E.1   PROOF OF LEMMA C.1

*Proof.* We know that the gradient of the loss function for the $n^{\text{th}}$ sample is

$$
\frac{\partial \ell}{\partial \boldsymbol{W}_{O(i,\cdot)}} = \frac{\partial \ell}{\partial F(\boldsymbol{X}^{(n)})} \cdot \frac{\partial F(\boldsymbol{X}^{(n)})}{\partial \boldsymbol{W}_{O(i,\cdot)}}
$$

$$
= -\frac{z^{(n)}}{L} \sum_{l=1}^{L} v_i \cdot \phi'\left(\boldsymbol{W}_{O(i,\cdot)} \boldsymbol{y}_l^{(n)}\right) \cdot \boldsymbol{y}_l^{(n)}. \tag{188}
$$

If we consider the gradient for the population loss,

$$
\frac{\partial \mathcal{L}}{\partial \boldsymbol{W}_{O(i,\cdot)}} = -\mathbb{E}\left[\frac{z^{(n)}}{L} \sum_{l=1}^{L} v_i \cdot \phi'\left(\boldsymbol{W}_{O(i,\cdot)} \boldsymbol{y}_l^{(n)}\right) \cdot \boldsymbol{y}_l^{(n)}\right] \tag{189}
$$

$$
= -\mathbb{E}_{z=+1}\left[\sum_{l=1}^{L} \frac{1}{L} v_i \cdot \phi'\left(\boldsymbol{W}_{O(i,\cdot)} \boldsymbol{y}_l^{(n)}\right) \cdot \boldsymbol{y}_l^{(n)}\right]
$$

$$
+ \mathbb{E}_{z=-1}\left[\sum_{l=1}^{L} \frac{1}{L} v_i \cdot \phi'\left(\boldsymbol{W}_{O(i,\cdot)} \boldsymbol{y}_l^{(n)}\right) \cdot \boldsymbol{y}_l^{(n)}\right]. \tag{190}
$$

We are given that

$$
p_1 \le \left\langle \boldsymbol{w}_\Delta^{(t)}, \boldsymbol{o}_+ \right\rangle \le q_1, \quad p_2 \le \left\langle \boldsymbol{w}_\Delta^{(t)}, \boldsymbol{o}_- \right\rangle \le q_2, \quad \text{and} \quad p_3 \le \left\langle \boldsymbol{w}_\Delta^{(t)}, \boldsymbol{o}_j \right\rangle \le q_3 \quad \text{for } j \ne 1, 2. \tag{191}
$$

The Mamba output can be written as

$$
\boldsymbol{y}_l(t) = \sum_{s=1}^{l} \left(\prod_{j=s+1}^{l} \left(1 - \sigma(\boldsymbol{w}_\Delta^{(t)\top} \boldsymbol{x}_j)\right)\right) \cdot \sigma(\boldsymbol{w}_\Delta^{(t)\top} \boldsymbol{x}_s) \cdot (\boldsymbol{x}_s^\top \boldsymbol{x}_l) \boldsymbol{x}_s \tag{192}
$$

We have to consider FOUR cases.

**Case I:** $l = s = L_1^+$

$$
\boldsymbol{x}_s = \boldsymbol{x}_l = \boldsymbol{o}_+ \tag{193}
$$

$$
\left\langle \mathbb{E} \boldsymbol{y}_{L_1^+}, \boldsymbol{o}_+ \right\rangle = \sigma(\boldsymbol{w}_\Delta^{(t)\top} \boldsymbol{o}_+) \ge \sigma(p_1) \pm \mathcal{O}(\tau) = \frac{1}{1 + e^{-p_1}} \pm \mathcal{O}(\tau). \tag{194}
$$

**Case II:** $l = s = L_2^+$

$$
\left\langle \mathbb{E} \boldsymbol{y}_{L_2^+, L_2^+}, \boldsymbol{o}_+ \right\rangle = \sigma(\boldsymbol{w}_\Delta^{(t)\top} \boldsymbol{o}_+) \pm \mathcal{O}(\tau). \tag{195}
$$

**Case III:** $l = L_2^+, s = L_1^+$

$$
\left\langle \mathbb{E} \boldsymbol{y}_{L_2^+, L_1^+}, \boldsymbol{o}_+ \right\rangle = \left(1 - \sigma(\boldsymbol{w}_\Delta^{(t)\top} \boldsymbol{o}_+)\right) \left(1 - \sigma(\boldsymbol{w}_\Delta^{(t)\top} \boldsymbol{o}_-)\right)
$$

$$
\cdot \left(1 - \sigma(\boldsymbol{w}_\Delta^{(t)\top} \boldsymbol{o}_j)\right)^{\Delta L_{o_+}^+ - 2} \cdot \sigma(\boldsymbol{w}_\Delta^{(t)\top} \boldsymbol{o}_+) \pm \mathcal{O}(\tau). \tag{196}
$$

Combining (195) and (196), we obtain

$$
\left\langle \mathbb{E}\boldsymbol{y}_{L_2^+}, \boldsymbol{o}_+ \right\rangle = \sigma({\boldsymbol{w}_\Delta^{(t)}}^\top \boldsymbol{o}_+)
$$
$$
+ \left(1 - \sigma({\boldsymbol{w}_\Delta^{(t)}}^\top \boldsymbol{o}_+)\right)\left(1 - \sigma({\boldsymbol{w}_\Delta^{(t)}}^\top \boldsymbol{o}_-)\right)
$$
$$
\cdot \left(1 - \sigma({\boldsymbol{w}_\Delta^{(t)}}^\top \boldsymbol{o}_j)\right)^{\Delta L_{o_+}^+ - 2} \cdot \sigma({\boldsymbol{w}_\Delta^{(t)}}^\top \boldsymbol{o}_+) \pm \mathcal{O}(\tau).
\tag{197}
$$

**Case IV: Others**
For the other token positions, $\boldsymbol{x}_l \neq \boldsymbol{o}_+$. Since we assume orthogonality among the features, $\boldsymbol{y}_l = 0$.

From our initialization, for the lucky neuron $i \in \mathcal{W}(0)$, $v_i = +\frac{1}{\sqrt{m}}$. For $i \in \mathcal{W}(0)$, and $z^{(n)} = +1$, we have

$$
\left\langle \mathbb{E}_{z=+1}\left[\sum_{l=1}^L \frac{1}{L} v_i \cdot \phi'\left(\boldsymbol{W}_{O(i,\cdot)}\boldsymbol{y}_l^{(n)}\right)\cdot \boldsymbol{y}_l^{(n)}\right], \boldsymbol{o}_+ \right\rangle
$$
$$
= \frac{1}{\sqrt{m}L}\cdot \sigma({\boldsymbol{w}_\Delta^{(t)}}^\top \boldsymbol{o}_+)\left[2 + \left(1 - \sigma({\boldsymbol{w}_\Delta^{(t)}}^\top \boldsymbol{o}_+)\right)\left(1 - \sigma({\boldsymbol{w}_\Delta^{(t)}}^\top \boldsymbol{o}_-)\right)\right.
$$
$$
\left.\cdot \left(1 - \sigma({\boldsymbol{w}_\Delta^{(t)}}^\top \boldsymbol{o}_j)\right)^{\Delta L_{o_+}^+ - 2}\right] \pm \mathcal{O}(\tau).
\tag{198}
$$

Similarly for $z = -1$, we can obtain

$$
\left\langle \mathbb{E}_{z=-1}\left[\sum_{l=1}^L \frac{1}{L} v_i \cdot \phi'\left(\boldsymbol{W}_{O(i,\cdot)}\boldsymbol{y}_l^{(n)}\right)\cdot \boldsymbol{y}_l^{(n)}\right], \boldsymbol{o}_+ \right\rangle
$$
$$
= \frac{1}{\sqrt{m}L}\cdot \sigma({\boldsymbol{w}_\Delta^{(t)}}^\top \boldsymbol{o}_+)\left[2 + \left(1 - \sigma({\boldsymbol{w}_\Delta^{(t)}}^\top \boldsymbol{o}_+)\right)\left(1 - \sigma({\boldsymbol{w}_\Delta^{(t)}}^\top \boldsymbol{o}_-)\right)\right.
$$
$$
\left.\cdot \left(1 - \sigma({\boldsymbol{w}_\Delta^{(t)}}^\top \boldsymbol{o}_j)\right)^{\Delta L_{o_+}^- - 2}\right] \pm \mathcal{O}(\tau).
\tag{199}
$$

Therfore, combining (198) and (199),

$$
\left\langle -\frac{\partial \mathcal{L}}{\partial \boldsymbol{W}_{O(i,\cdot)}^{(t)}}, \boldsymbol{o}_+ \right\rangle = \left\langle \mathbb{E}_{z=+1}\left[\sum_{l=1}^L \frac{1}{L} v_i \cdot \phi'\left(\boldsymbol{W}_{O(i,\cdot)}\boldsymbol{y}_l^{(n)}\right)\cdot \boldsymbol{y}_l^{(n)}\right], \boldsymbol{o}_+ \right\rangle
$$
$$
- \left\langle \mathbb{E}_{z=-1}\left[\sum_{l=1}^L \frac{1}{L} v_i \cdot \phi'\left(\boldsymbol{W}_{O(i,\cdot)}\boldsymbol{y}_l^{(n)}\right)\cdot \boldsymbol{y}_l^{(n)}\right], \boldsymbol{o}_+ \right\rangle
$$
$$
= \frac{1}{\sqrt{m}L}\cdot \sigma({\boldsymbol{w}_\Delta^{(t)}}^\top \boldsymbol{o}_+)\cdot \left(1 - \sigma({\boldsymbol{w}_\Delta^{(t)}}^\top \boldsymbol{o}_+)\right)\cdot \left(1 - \sigma({\boldsymbol{w}_\Delta^{(t)}}^\top \boldsymbol{o}_-)\right)\cdot
$$
$$
\left[\left(1 - \sigma({\boldsymbol{w}_\Delta^{(t)}}^\top \boldsymbol{o}_j)\right)^{\Delta L_{o_+}^+ - 2} - \left(1 - \sigma({\boldsymbol{w}_\Delta^{(t)}}^\top \boldsymbol{o}_j)\right)^{\Delta L_{o_+}^- - 2}\right] \pm \mathcal{O}(\tau).
\tag{200}
$$

We aim to bound the deviation between the gradient of the population loss and that of the empirical loss. Specifically, $\left\| \frac{\partial \mathcal{L}}{\partial \boldsymbol{W}_{O(i,\cdot)}^{(t)}} - \frac{\partial \hat{\mathcal{L}}}{\partial \boldsymbol{W}_{O(i,\cdot)}^{(t)}} \right\|_2 = \left\| \frac{1}{N}\sum_{n=1}^N \boldsymbol{\gamma}_n - \mathbb{E}\boldsymbol{\gamma}_n \right\|_2$, where

$$\boldsymbol{\gamma}_n = \frac{z^{(n)}}{L} \sum_{l=1}^{L} v_i \, \phi' \left( \boldsymbol{W}_{O(i,\cdot)} \boldsymbol{y}_l^{(n)} \right) \boldsymbol{y}_l^{(n)}. \tag{201}$$

Consider a fixed vector $\boldsymbol{\alpha}$ with $\|\boldsymbol{\alpha}\|_2 = 1$. We will show that $\boldsymbol{\alpha}^\top \boldsymbol{\gamma}_n$ is a sub-Gaussian random variable.

$$\left| \boldsymbol{\alpha}^\top \boldsymbol{\gamma}_n \right| \leq \|\boldsymbol{\alpha}\|_2 \cdot \|\boldsymbol{\gamma}_n\|_2 = \|\boldsymbol{\gamma}_n\|_2. \tag{202}$$

By the problem setup, we know that

$$|v_i| = \frac{1}{\sqrt{m}}, \quad |z^{(n)}| = 1, \quad \left| \phi' \left( \boldsymbol{W}_{O(i,\cdot)} \boldsymbol{y}_l^{(n)} \right) \right| \leq 1. \tag{203}$$

Recall the Mamba output,

$$\boldsymbol{y}_l^{(n)}(t) = \sum_{s=1}^{l} \left( \prod_{j=s+1}^{l} \left( 1 - \sigma(\boldsymbol{w}_\Delta^{(t)\top} \boldsymbol{x}_j) \right) \right) \cdot \sigma(\boldsymbol{w}_\Delta^{(t)\top} \boldsymbol{x}_s) \cdot (\boldsymbol{x}_s^\top \boldsymbol{x}_l) \boldsymbol{x}_s. \tag{204}$$

Since $\|\boldsymbol{x}_s\|_2 = 1$, we get

$$\left\| \boldsymbol{y}_l^{(n)} \right\|_2 \leq \sum_{s=1}^{l} \left| a^{l-s+1} \cdot \left( \boldsymbol{x}_s^\top \boldsymbol{x}_l \right) \right| \cdot \|\boldsymbol{x}_s\|_2$$

$$\leq \sum_{s=1}^{l} \frac{a}{1-a} \cdot 1 \cdot 1 = a' \quad \text{(where } a' \text{ denotes a constant).} \tag{205}$$

Therefore, the norm of $\boldsymbol{\gamma}_n$ satisfies

$$\|\boldsymbol{\gamma}_n\|_2 \leq \frac{1}{L} \sum_{l=1}^{L} |v_i| \cdot \left| \phi' \left( \boldsymbol{W}_{O(i,\cdot)} \boldsymbol{y}_l^{(n)} \right) \right| \cdot \left\| \boldsymbol{y}_l^{(n)} \right\|_2$$

$$\leq \frac{1}{L} \cdot \frac{1}{\sqrt{m}} \sum_{l=1}^{L} \left\| \boldsymbol{y}_l^{(n)} \right\|_2$$

$$\leq \frac{1}{L} \cdot \frac{1}{\sqrt{m}} \cdot \sum_{l=1}^{L} a' = \frac{a'}{\sqrt{m}}. \tag{206}$$

Hence,

$$\left| \boldsymbol{\alpha}^\top \boldsymbol{\gamma}_n \right| \leq \frac{a'}{\sqrt{m}} \quad \text{(bounded).} \tag{207}$$

This implies that $\boldsymbol{\alpha}^\top \boldsymbol{\gamma}_n$ is sub-Gaussian with variance proxy

$$\sigma^2 = \mathcal{O} \left( \frac{1}{m} \right). \tag{208}$$

Now consider the independent sub-Gaussian variables $\boldsymbol{\alpha}^\top \boldsymbol{\gamma}_1, \ldots, \boldsymbol{\alpha}^\top \boldsymbol{\gamma}_N$, each bounded as

$$-\frac{1}{\sqrt{m}} \leq \boldsymbol{\alpha}^\top \boldsymbol{\gamma}_n \leq \frac{1}{\sqrt{m}}. \tag{209}$$

Applying Hoeffding's inequality, for any $q > 0$,

$$\mathbb{P} \left( \left| \frac{1}{N} \sum_{n=1}^{N} \boldsymbol{\alpha}^\top \boldsymbol{\gamma}_n - \mathbb{E} \boldsymbol{\alpha}^\top \boldsymbol{\gamma}_n \right| \gtrsim \sqrt{\frac{q \log N}{mN}} \right) \leq N^{-q}. \tag{210}$$

Observe that this can be written as

$$\frac{1}{N}\sum_{n=1}^{N}\boldsymbol{\alpha}^{\top}\boldsymbol{\gamma}_n - \mathbb{E}\boldsymbol{\alpha}^{\top}\boldsymbol{\gamma}_n = \boldsymbol{\alpha}^{\top}\left(\frac{1}{N}\sum_{n=1}^{N}\boldsymbol{\gamma}_n - \mathbb{E}\boldsymbol{\gamma}_n\right) := \boldsymbol{\alpha}^{\top}\boldsymbol{\zeta}. \tag{211}$$

Therefore, by Hoeffding's inequality (cf. (210)),

$$\mathbb{P}\left(|\boldsymbol{\alpha}^{\top}\boldsymbol{\zeta}| \gtrsim \sqrt{\frac{q\log N}{mN}}\right) \le N^{-q}. \tag{212}$$

To bound $\|\boldsymbol{\zeta}\|_2$, we use the dual norm identity

$$\|\boldsymbol{\zeta}\|_2 = \sup_{\|\boldsymbol{\alpha}\|_2=1}\boldsymbol{\alpha}^{\top}\boldsymbol{\zeta}. \tag{213}$$

We apply an $\varepsilon$-cover argument to obtain

$$\sup_{\|\boldsymbol{\alpha}\|_2=1}\boldsymbol{\alpha}^{\top}\boldsymbol{\zeta} \le \frac{1}{1-\varepsilon}\max_{\boldsymbol{\alpha}\in\mathcal{C}_\varepsilon}\boldsymbol{\alpha}^{\top}\boldsymbol{\zeta}$$
$$\le 2\max_{\boldsymbol{\alpha}\in\mathcal{C}_{1/2}}\boldsymbol{\alpha}^{\top}\boldsymbol{\zeta}. \tag{214}$$

We have shown that for any fixed $\boldsymbol{\alpha}$,

$$\mathbb{P}\left(|\boldsymbol{\alpha}^{\top}\boldsymbol{\zeta}| \gtrsim \sqrt{\frac{q\log N}{mN}}\right) \le N^{-q}. \tag{215}$$

Therefore, for all fixed $\boldsymbol{\alpha}\in\mathcal{C}_{1/2}$,

$$|\boldsymbol{\alpha}^{\top}\boldsymbol{\zeta}| \gtrsim \sqrt{\frac{q\log N}{mN}} \quad \text{with probability at most } N^{-q}. \tag{216}$$

Then,

$$\max_{\boldsymbol{\alpha}\in\mathcal{C}_{1/2}}|\boldsymbol{\alpha}^{\top}\boldsymbol{\zeta}| \gtrsim \sqrt{\frac{q\log N}{mN}} \quad \text{with probability at most } |\mathcal{C}_{1/2}|N^{-q}. \tag{217}$$

Recall that the covering number satisfies

$$|\mathcal{C}_\varepsilon| \le \left(\frac{3B}{\varepsilon}\right)^d. \tag{218}$$

For $B=1$ and $\varepsilon=\frac{1}{2}$, we have

$$|\mathcal{C}_{1/2}| \le 6^d. \tag{219}$$

We can therefore write

$$\mathbb{P}\left(\|\boldsymbol{\zeta}\|_2 \gtrsim \sqrt{\frac{q\log N}{mN}}\right) \le 6^d \cdot N^{-q}. \tag{220}$$

We want this probability to be sufficiently small. Set $q=d$, so that

$$\mathbb{P}\left(\|\boldsymbol{\zeta}\|_2 \gtrsim 2\sqrt{\frac{d\log N}{mN}}\right) \le \left(\frac{N}{6}\right)^{-d}. \tag{221}$$

Hence, the deviation is bounded with high probability:

$$\|\boldsymbol{\zeta}\|_2 > \mathcal{O}\left(\sqrt{\frac{d\log N}{mN}}\right) \quad \text{with probability at most } \mathcal{O}(N^{-d}). \tag{222}$$

Or equivalently, with probability at most $\mathcal{O}(N^{-d})$,

$$\left\| \frac{1}{N} \sum_{n=1}^{N} \boldsymbol{\gamma}_n - \mathbb{E}\boldsymbol{\gamma}_n \right\|_2 > \mathcal{O}\left( \sqrt{\frac{d \log N}{mN}} \right). \tag{223}$$

That is, with high probability $1 - \mathcal{O}(N^{-d})$, we have

$$\left\| \frac{1}{N} \sum_{n=1}^{N} \boldsymbol{\gamma}_n - \mathbb{E}\boldsymbol{\gamma}_n \right\|_2 \leq \mathcal{O}\left( \sqrt{\frac{d \log N}{mN}} \right). \tag{224}$$

Using the identities

$$-\frac{\partial \hat{\mathcal{L}}}{\partial \boldsymbol{W}_{O(i,\cdot)}^{(t)}} = \frac{1}{N} \sum_{n=1}^{N} \boldsymbol{\gamma}_n, \quad -\frac{\partial \mathcal{L}}{\partial \boldsymbol{W}_{O(i,\cdot)}^{(t)}} = \mathbb{E}\boldsymbol{\gamma}_n, \tag{225}$$

we conclude that, with high probability,

$$\left\| \left( -\frac{\partial \hat{\mathcal{L}}}{\partial \boldsymbol{W}_{O(i,\cdot)}^{(t)}} \right) - \left( -\frac{\partial \mathcal{L}}{\partial \boldsymbol{W}_{O(i,\cdot)}^{(t)}} \right) \right\|_2 = \left\| \frac{\partial \mathcal{L}}{\partial \boldsymbol{W}_{O(i,\cdot)}^{(t)}} - \frac{\partial \hat{\mathcal{L}}}{\partial \boldsymbol{W}_{O(i,\cdot)}^{(t)}} \right\|_2 \leq \mathcal{O}\left( \sqrt{\frac{d \log N}{mN}} \right). \tag{226}$$

Using the Cauchy–Schwarz inequality, we have

$$\left| \left\langle \frac{\partial \mathcal{L}}{\partial \boldsymbol{W}_{O(i,\cdot)}^{(t)}} - \frac{\partial \hat{\mathcal{L}}}{\partial \boldsymbol{W}_{O(i,\cdot)}^{(t)}}, \boldsymbol{o}_+ \right\rangle \right| \leq \left\| \frac{\partial \mathcal{L}}{\partial \boldsymbol{W}_{O(i,\cdot)}^{(t)}} - \frac{\partial \hat{\mathcal{L}}}{\partial \boldsymbol{W}_{O(i,\cdot)}^{(t)}} \right\|_2 \cdot \|\boldsymbol{o}_+\|_2$$

$$= \left\| \frac{\partial \mathcal{L}}{\partial \boldsymbol{W}_{O(i,\cdot)}^{(t)}} - \frac{\partial \hat{\mathcal{L}}}{\partial \boldsymbol{W}_{O(i,\cdot)}^{(t)}} \right\|_2 \quad \text{(since } \|\boldsymbol{o}_+\|_2 = 1\text{)}$$

$$\leq \mathcal{O}\left( \sqrt{\frac{d \log N}{mN}} \right). \tag{227}$$

Therefore, we obtain

$$\left\langle -\frac{\partial \mathcal{L}}{\partial \boldsymbol{W}_{O(i,\cdot)}^{(t)}}, \boldsymbol{o}_+ \right\rangle - \mathcal{O}\left( \sqrt{\frac{d \log N}{mN}} \right) \leq \left\langle -\frac{\partial \hat{\mathcal{L}}}{\partial \boldsymbol{W}_{O(i,\cdot)}^{(t)}}, \boldsymbol{o}_+ \right\rangle$$

$$\leq \left\langle -\frac{\partial \mathcal{L}}{\partial \boldsymbol{W}_{O(i,\cdot)}^{(t)}}, \boldsymbol{o}_+ \right\rangle + \mathcal{O}\left( \sqrt{\frac{d \log N}{mN}} \right). \tag{228}$$

By pairing (200) with the given the conditions on $\boldsymbol{w}_\Delta$ in (191), we can write

$$\left\langle -\frac{\partial \mathcal{L}}{\partial \boldsymbol{W}_{O(i,\cdot)}^{(t)}}, \boldsymbol{o}_+ \right\rangle$$

$$\geq \frac{1}{\sqrt{mL}} \cdot \sigma(p_1) \cdot (1 - \sigma(q_1)) \cdot (1 - \sigma(q_2)) \left[ (1 - \sigma(q_3))^{\Delta L_{o_+}^+ - 2} - (1 - \sigma(p_3))^{\Delta L_{o_+}^- - 2} \right] - \mathcal{O}(\tau) \tag{229}$$

and

$$\left\langle -\frac{\partial \mathcal{L}}{\partial \boldsymbol{W}_{O(i,\cdot)}^{(t)}}, \boldsymbol{o}_+ \right\rangle$$

$$\leq \frac{1}{\sqrt{mL}} \cdot \sigma(q_1) \cdot (1 - \sigma(p_1)) \cdot (1 - \sigma(p_2)) \left[ (1 - \sigma(p_3))^{\Delta L_{o_+}^+ - 2} - (1 - \sigma(q_3))^{\Delta L_{o_+}^- - 2} \right] + \mathcal{O}(\tau) \tag{230}$$

Therefore, we can obtain the lower bound and the upper bound of $\left\langle -\frac{\partial \hat{\mathcal{L}}}{\partial \boldsymbol{W}_{O(i,\cdot)}^{(t)}}, \boldsymbol{o}_+ \right\rangle$ as

$$
\frac{1}{\sqrt{mL}} \cdot \sigma(p_1) \cdot (1 - \sigma(q_1)) \cdot (1 - \sigma(q_2)) \left[ (1 - \sigma(q_3))^{\Delta L_{o_+}^+ - 2} \right.
$$

$$
\left. - (1 - \sigma(p_3))^{\Delta L_{o_+}^- - 2} \right] - \mathcal{O}\left( \sqrt{\frac{d \log N}{mN}} \right) - \mathcal{O}(\tau)
$$

$$
\leq \left\langle -\frac{\partial \hat{\mathcal{L}}}{\partial \boldsymbol{W}_{O(i,\cdot)}^{(t)}}, \boldsymbol{o}_+ \right\rangle
$$

(231)

and

$$
\left\langle -\frac{\partial \hat{\mathcal{L}}}{\partial \boldsymbol{W}_{O(i,\cdot)}^{(t)}}, \boldsymbol{o}_+ \right\rangle
$$

$$
\leq \frac{1}{\sqrt{mL}} \cdot \sigma(q_1) \cdot (1 - \sigma(p_1)) \cdot (1 - \sigma(p_2)) \left[ (1 - \sigma(p_3))^{\Delta L_{o_+}^+ - 2} - (1 - \sigma(q_3))^{\Delta L_{o_+}^- - 2} \right] \quad (232)
$$

$$
+ \mathcal{O}\left( \sqrt{\frac{d \log N}{mN}} \right) + \mathcal{O}(\tau).
$$

This concludes the proof of (106) and (107) in Lemma C.1.

To obtain $\left\langle -\frac{\partial \hat{\mathcal{L}}}{\partial \boldsymbol{W}_{O(i,\cdot)}^{(t)}}, \boldsymbol{o}_- \right\rangle$, we have to consider $\mathbb{E}_{z=-1}\left[ \sum_{l=1}^{L} \frac{1}{L} v_i \cdot \phi'\left( \boldsymbol{W}_{O(i,\cdot)} \boldsymbol{y}_l^{(n)} \right) \cdot \boldsymbol{y}_l^{(n)} \right]$.

If $\boldsymbol{W}_{O(i,\cdot)}^{(t)} \boldsymbol{o}_- > 0$,

$$
\left\langle \mathbb{E}_{z=-1}\left[ \sum_{l=1}^{L} \frac{1}{L} v_i \cdot \phi'\left( \boldsymbol{W}_{O(i,\cdot)} \boldsymbol{y}_l^{(n)} \right) \cdot \boldsymbol{y}_l^{(n)} \right], \boldsymbol{o}_- \right\rangle
$$

$$
= \frac{1}{\sqrt{mL}} \cdot \sigma(\boldsymbol{w}_\Delta^{(t)\top} \boldsymbol{o}_-) \left[ 2 + \left( 1 - \sigma(\boldsymbol{w}_\Delta^{(t)\top} \boldsymbol{o}_-) \right) \left( 1 - \sigma(\boldsymbol{w}_\Delta^{(t)\top} \boldsymbol{o}_+) \right) \right.
$$

$$
\left. \cdot \left( 1 - \sigma(\boldsymbol{w}_\Delta^{(t)\top} \boldsymbol{o}_j) \right)^{\Delta L_{o_-}^- - 2} \right] \pm \mathcal{O}(\tau).
$$

(233)

If $\boldsymbol{W}_{O(i,\cdot)}^{(t)} \boldsymbol{o}_- \leq 0$,

$$
\left\langle \mathbb{E}_{z=-1}\left[ \sum_{l=1}^{L} \frac{1}{L} v_i \cdot \phi'\left( \boldsymbol{W}_{O(i,\cdot)} \boldsymbol{y}_l^{(n)} \right) \cdot \boldsymbol{y}_l^{(n)} \right], \boldsymbol{o}_- \right\rangle = 0 \pm \mathcal{O}(\tau).
$$

(234)

From (190), We know that

$$
\left\langle -\frac{\partial \mathcal{L}}{\partial \boldsymbol{W}_{O(i,\cdot)}^{(t)}}, \boldsymbol{o}_- \right\rangle = \left\langle \mathbb{E}_{z=+1}\left[ \sum_{l=1}^{L} \frac{1}{L} v_i \cdot \phi'\left( \boldsymbol{W}_{O(i,\cdot)} \boldsymbol{y}_l^{(n)} \right) \cdot \boldsymbol{y}_l^{(n)} \right], \boldsymbol{o}_- \right\rangle
$$

$$
- \left\langle \mathbb{E}_{z=-1}\left[ \sum_{l=1}^{L} \frac{1}{L} v_i \cdot \phi'\left( \boldsymbol{W}_{O(i,\cdot)} \boldsymbol{y}_l^{(n)} \right) \cdot \boldsymbol{y}_l^{(n)} \right], \boldsymbol{o}_- \right\rangle.
$$

(235)

Hence, combining both cases, we conclude

$$- \frac{1}{\sqrt{mL}} \cdot \sigma(\boldsymbol{w}_\Delta^{(t)\top}\boldsymbol{o}_-) \left[ 2 + \left( 1 - \sigma(\boldsymbol{w}_\Delta^{(t)\top}\boldsymbol{o}_-) \right) \left( 1 - \sigma(\boldsymbol{w}_\Delta^{(t)\top}\boldsymbol{o}_+) \right) \right.$$
$$\left. \cdot \left( 1 - \sigma(\boldsymbol{w}_\Delta^{(t)\top}\boldsymbol{o}_j) \right)^{\Delta L_{o_-}^- - 2} \right] - \mathcal{O}(\tau)$$
$$\leq \left\langle -\frac{\partial \mathcal{L}}{\partial \boldsymbol{W}_{O(i,\cdot)}^{(t)}}, \boldsymbol{o}_- \right\rangle \leq \mathcal{O}(\tau). \tag{236}$$

From (226), similar to (228), we can write

$$\left\langle -\frac{\partial \mathcal{L}}{\partial \boldsymbol{W}_{O(i,\cdot)}^{(t)}}, \boldsymbol{o}_- \right\rangle - \mathcal{O}\left( \sqrt{\frac{d \log N}{mN}} \right)$$
$$\leq \left\langle -\frac{\partial \hat{\mathcal{L}}}{\partial \boldsymbol{W}_{O(i,\cdot)}^{(t)}}, \boldsymbol{o}_- \right\rangle$$
$$\leq \left\langle -\frac{\partial \mathcal{L}}{\partial \boldsymbol{W}_{O(i,\cdot)}^{(t)}}, \boldsymbol{o}_- \right\rangle + \mathcal{O}\left( \sqrt{\frac{d \log N}{mN}} \right). \tag{237}$$

Hence, we have

$$- \frac{1}{\sqrt{mL}} \cdot \sigma(\boldsymbol{w}_\Delta^{(t)\top}\boldsymbol{o}_-) \left[ 2 + \left( 1 - \sigma(\boldsymbol{w}_\Delta^{(t)\top}\boldsymbol{o}_-) \right) \left( 1 - \sigma(\boldsymbol{w}_\Delta^{(t)\top}\boldsymbol{o}_+) \right) \right.$$
$$\left. \cdot \left( 1 - \sigma(\boldsymbol{w}_\Delta^{(t)\top}\boldsymbol{o}_j) \right)^{\Delta L_{o_-}^- - 2} \right] - \mathcal{O}\left( \sqrt{\frac{d \log N}{mN}} \right) - \mathcal{O}(\tau)$$
$$\leq \left\langle -\frac{\partial \hat{\mathcal{L}}}{\partial \boldsymbol{W}_{O(i,\cdot)}^{(t)}}, \boldsymbol{o}_- \right\rangle \leq \mathcal{O}\left( \sqrt{\frac{d \log N}{mN}} \right) + \mathcal{O}(\tau). \tag{238}$$

This concludes the proof of (108) and (109) in Lemma C.1.

Now consider $\left\langle -\frac{\partial \mathcal{L}}{\partial \boldsymbol{W}_{O(i,\cdot)}^{(t)}}, \boldsymbol{o}_j \right\rangle$ for $j \neq 1, 2$.

$$\left\langle -\frac{\partial \mathcal{L}}{\partial \boldsymbol{W}_{O(i,\cdot)}^{(t)}}, \boldsymbol{o}_j \right\rangle = \left\langle \mathbb{E}_{z=+1} \left[ \sum_{l=1}^{L} \frac{1}{L} v_i \cdot \phi'\left( \boldsymbol{W}_{O(i,\cdot)} \boldsymbol{y}_l^{(n)} \right) \cdot \boldsymbol{y}_l^{(n)} \right], \boldsymbol{o}_j \right\rangle$$
$$- \left\langle \mathbb{E}_{z=-1} \left[ \sum_{l=1}^{L} \frac{1}{L} v_i \cdot \phi'\left( \boldsymbol{W}_{O(i,\cdot)} \boldsymbol{y}_l^{(n)} \right) \cdot \boldsymbol{y}_l^{(n)} \right], \boldsymbol{o}_j \right\rangle$$
$$:= \langle I_1, \boldsymbol{o}_j \rangle - \langle I_2, \boldsymbol{o}_j \rangle. \tag{239}$$

Because $\boldsymbol{o}_j$ for $j \neq 1, 2$ is identical in both $I_1$ and $I_2$, $\langle I_1, \boldsymbol{o}_j \rangle - \langle I_2, \boldsymbol{o}_j \rangle = 0 \pm \mathcal{O}(\tau)$. Hence, $\left\langle -\frac{\partial \mathcal{L}}{\partial \boldsymbol{W}_{O(i,\cdot)}^{(t)}}, \boldsymbol{o}_j \right\rangle = 0 \pm \mathcal{O}(\tau)$. From (226), similar to (228), we can write

$$\left\langle -\frac{\partial \mathcal{L}}{\partial \boldsymbol{W}_{O(i,\cdot)}^{(t)}}, \boldsymbol{o}_j \right\rangle - \mathcal{O}\left( \sqrt{\frac{d \log N}{mN}} \right)$$
$$\leq \left\langle -\frac{\partial \hat{\mathcal{L}}}{\partial \boldsymbol{W}_{O(i,\cdot)}^{(t)}}, \boldsymbol{o}_j \right\rangle$$
$$\leq \left\langle -\frac{\partial \mathcal{L}}{\partial \boldsymbol{W}_{O(i,\cdot)}^{(t)}}, \boldsymbol{o}_j \right\rangle + \mathcal{O}\left( \sqrt{\frac{d \log N}{mN}} \right) + \mathcal{O}(\tau). \tag{240}$$

Therefore,

$$\left\langle -\frac{\partial \hat{\mathcal{L}}}{\partial \boldsymbol{W}_{O(i,\cdot)}^{(t)}}, \boldsymbol{o}_j \right\rangle \leq \mathcal{O}\left(\sqrt{\frac{d\log N}{mN}}\right) \text{ for } j \neq 1, 2. \tag{241}$$

This concludes the proof of (110) in Lemma C.1.

$\square$

### E.2 Proof of Lemma C.2

*Proof.* By definition, for any unlucky neuron $i \in \mathcal{K}_+ \setminus \mathcal{W}(0)$, we have

$$\boldsymbol{W}_{O(i,\cdot)}\boldsymbol{o}_+ \leq 0. \tag{242}$$

We first consider the alignment with $\boldsymbol{o}_+$. That is,

$$\left\langle -\frac{\partial \hat{\mathcal{L}}}{\partial \boldsymbol{W}_{O(i,\cdot)}^{(t)}}, \boldsymbol{o}_+ \right\rangle. \tag{243}$$

The gradient is given in (189). We only need to consider the cases where $\left\langle \boldsymbol{y}_l^{(n)}, \boldsymbol{o}_+ \right\rangle > 0$. However, since $\boldsymbol{W}_{O(i,\cdot)}\boldsymbol{o}_+ \leq 0$, we have

$$\phi'\left(\boldsymbol{W}_{O(i,\cdot)}\boldsymbol{y}_l^{(n)}\right) = 0. \tag{244}$$

$$\left\langle -\frac{\partial \mathcal{L}}{\partial \boldsymbol{W}_{O(i,\cdot)}^{(t)}}, \boldsymbol{o}_+ \right\rangle = \left\langle \mathbb{E}_{z=+1}\left[\sum_{l=1}^L \frac{1}{L}v_i \cdot \phi'\left(\boldsymbol{W}_{O(i,\cdot)}\boldsymbol{y}_l^{(n)}\right) \cdot \boldsymbol{y}_l^{(n)}\right], \boldsymbol{o}_+ \right\rangle$$

$$- \left\langle \mathbb{E}_{z=-1}\left[\sum_{l=1}^L \frac{1}{L}v_i \cdot \phi'\left(\boldsymbol{W}_{O(i,\cdot)}\boldsymbol{y}_l^{(n)}\right) \cdot \boldsymbol{y}_l^{(n)}\right], \boldsymbol{o}_+ \right\rangle$$

$$= 0 \pm \mathcal{O}(\tau). \tag{245}$$

We know by (228),

$$\left\langle -\frac{\partial \hat{\mathcal{L}}}{\partial \boldsymbol{W}_{O(i,\cdot)}^{(t)}}, \boldsymbol{o}_+ \right\rangle \leq \left\langle -\frac{\partial \mathcal{L}}{\partial \boldsymbol{W}_{O(i,\cdot)}^{(t)}}, \boldsymbol{o}_+ \right\rangle + \mathcal{O}\left(\sqrt{\frac{d\log N}{mN}}\right). \tag{246}$$

Hence,

$$\left\langle -\frac{\partial \hat{\mathcal{L}}}{\partial \boldsymbol{W}_{O(i,\cdot)}^{(t)}}, \boldsymbol{o}_+ \right\rangle \leq \mathcal{O}\left(\sqrt{\frac{d\log N}{mN}}\right) + \mathcal{O}(\tau). \tag{247}$$

We now analyze the alignment with $\boldsymbol{o}_-$. To obtain the bound on $\left\langle -\frac{\partial \hat{\mathcal{L}}}{\partial \boldsymbol{W}_{O(i,\cdot)}^{(t)}}, \boldsymbol{o}_- \right\rangle$, we consider the expectation $\mathbb{E}_{z=-1}\left[\sum_{l=1}^L \frac{1}{L}v_i \cdot \phi'\left(\boldsymbol{W}_{O(i,\cdot)}\boldsymbol{y}_l^{(n)}\right) \cdot \boldsymbol{y}_l^{(n)}\right]$.

If $\boldsymbol{W}_{O(i,\cdot)}^{(t)}\boldsymbol{o}_- > 0$, the inner product satisfies

$$\left\langle \mathbb{E}_{z=-1}\left[\sum_{l=1}^L \frac{1}{L}v_i \cdot \phi'\left(\boldsymbol{W}_{O(i,\cdot)}\boldsymbol{y}_l^{(n)}\right) \cdot \boldsymbol{y}_l^{(n)}\right], \boldsymbol{o}_- \right\rangle$$

$$= \frac{1}{\sqrt{mL}} \cdot \sigma(\boldsymbol{w}_\Delta^{(t)^\top}\boldsymbol{o}_-)\left[2 + \left(1 - \sigma(\boldsymbol{w}_\Delta^{(t)^\top}\boldsymbol{o}_-)\right)\left(1 - \sigma(\boldsymbol{w}_\Delta^{(t)^\top}\boldsymbol{o}_+)\right)\right.$$

$$\left. \cdot \left(1 - \sigma(\boldsymbol{w}_\Delta^{(t)^\top}\boldsymbol{o}_j)\right)^{\Delta L_{\boldsymbol{o}_-}^- - 2}\right] \pm \mathcal{O}(\tau). \tag{248}$$

If $\boldsymbol{W}_{O(i,\cdot)}^{(t)}\boldsymbol{o}_- \leq 0$, then

$$\left\langle \mathbb{E}_{z=-1}\left[\sum_{l=1}^{L}\frac{1}{L}v_i \cdot \phi'\left(\boldsymbol{W}_{O(i,\cdot)}\boldsymbol{y}_l^{(n)}\right)\cdot \boldsymbol{y}_l^{(n)}\right], \boldsymbol{o}_-\right\rangle = 0 \pm \mathcal{O}(\tau). \tag{249}$$

From (190), We know that

$$\left\langle -\frac{\partial \mathcal{L}}{\partial \boldsymbol{W}_{O(i,\cdot)}^{(t)}}, \boldsymbol{o}_-\right\rangle = \left\langle \mathbb{E}_{z=+1}\left[\sum_{l=1}^{L}\frac{1}{L}v_i \cdot \phi'\left(\boldsymbol{W}_{O(i,\cdot)}\boldsymbol{y}_l^{(n)}\right)\cdot \boldsymbol{y}_l^{(n)}\right], \boldsymbol{o}_-\right\rangle$$

$$- \left\langle \mathbb{E}_{z=-1}\left[\sum_{l=1}^{L}\frac{1}{L}v_i \cdot \phi'\left(\boldsymbol{W}_{O(i,\cdot)}\boldsymbol{y}_l^{(n)}\right)\cdot \boldsymbol{y}_l^{(n)}\right], \boldsymbol{o}_-\right\rangle. \tag{250}$$

Hence, combining both cases, we conclude

$$-\frac{1}{\sqrt{mL}} \cdot \sigma(\boldsymbol{w}_\Delta^{(t)^\top}\boldsymbol{o}_-)\left[2 + \left(1 - \sigma(\boldsymbol{w}_\Delta^{(t)^\top}\boldsymbol{o}_-)\right)\left(1 - \sigma(\boldsymbol{w}_\Delta^{(t)^\top}\boldsymbol{o}_+)\right)\right.$$

$$\left.\cdot \left(1 - \sigma(\boldsymbol{w}_\Delta^{(t)^\top}\boldsymbol{o}_j)\right)^{\Delta L_{\boldsymbol{o}_-}^- - 2}\right] - \mathcal{O}(\tau).$$

$$\leq \left\langle -\frac{\partial \mathcal{L}}{\partial \boldsymbol{W}_{O(i,\cdot)}^{(t)}}, \boldsymbol{o}_-\right\rangle \leq \mathcal{O}(\tau). \tag{251}$$

From (226), similar to (228), we can write

$$\left\langle -\frac{\partial \mathcal{L}}{\partial \boldsymbol{W}_{O(i,\cdot)}^{(t)}}, \boldsymbol{o}_-\right\rangle - \mathcal{O}\left(\sqrt{\frac{d\log N}{mN}}\right)$$

$$\leq \left\langle -\frac{\partial \hat{\mathcal{L}}}{\partial \boldsymbol{W}_{O(i,\cdot)}^{(t)}}, \boldsymbol{o}_-\right\rangle$$

$$\leq \left\langle -\frac{\partial \mathcal{L}}{\partial \boldsymbol{W}_{O(i,\cdot)}^{(t)}}, \boldsymbol{o}_-\right\rangle + \mathcal{O}\left(\sqrt{\frac{d\log N}{mN}}\right). \tag{252}$$

Hence,

$$-\frac{1}{\sqrt{mL}} \cdot \sigma(\boldsymbol{w}_\Delta^{(t)^\top}\boldsymbol{o}_-)\left[2 + \left(1 - \sigma(\boldsymbol{w}_\Delta^{(t)^\top}\boldsymbol{o}_-)\right)\left(1 - \sigma(\boldsymbol{w}_\Delta^{(t)^\top}\boldsymbol{o}_+)\right)\right.$$

$$\left.\cdot \left(1 - \sigma(\boldsymbol{w}_\Delta^{(t)^\top}\boldsymbol{o}_j)\right)^{\Delta L_{\boldsymbol{o}_-}^- - 2}\right] - \mathcal{O}\left(\sqrt{\frac{d\log N}{mN}}\right) - \mathcal{O}(\tau)$$

$$\leq \left\langle -\frac{\partial \hat{\mathcal{L}}}{\partial \boldsymbol{W}_{O(i,\cdot)}^{(t)}}, \boldsymbol{o}_-\right\rangle \leq \mathcal{O}\left(\sqrt{\frac{d\log N}{mN}}\right) + \mathcal{O}(\tau). \tag{253}$$

Now consider $\left\langle -\frac{\partial \mathcal{L}}{\partial \boldsymbol{W}_{O(i,\cdot)}^{(t)}}, \boldsymbol{o}_j\right\rangle$ for $j \neq 1, 2$.

$$\left\langle -\frac{\partial \mathcal{L}}{\partial \boldsymbol{W}_{O(i,\cdot)}^{(t)}}, \boldsymbol{o}_j\right\rangle = \left\langle \mathbb{E}_{z=+1}\left[\sum_{l=1}^{L}\frac{1}{L}v_i \cdot \phi'\left(\boldsymbol{W}_{O(i,\cdot)}\boldsymbol{y}_l^{(n)}\right)\cdot \boldsymbol{y}_l^{(n)}\right], \boldsymbol{o}_j\right\rangle$$

$$- \left\langle \mathbb{E}_{z=-1}\left[\sum_{l=1}^{L}\frac{1}{L}v_i \cdot \phi'\left(\boldsymbol{W}_{O(i,\cdot)}\boldsymbol{y}_l^{(n)}\right)\cdot \boldsymbol{y}_l^{(n)}\right], \boldsymbol{o}_j\right\rangle$$

$$:= \langle I_1, \boldsymbol{o}_j\rangle - \langle I_2, \boldsymbol{o}_j\rangle. \tag{254}$$

Because $\boldsymbol{o}_j$ for $j \neq 1, 2$ is identical in both $I_1$ and $I_2$, $\langle I_1, \boldsymbol{o}_j \rangle - \langle I_2, \boldsymbol{o}_j \rangle = 0 \pm \mathcal{O}(\tau)$. Hence, $\left\langle -\frac{\partial \mathcal{L}}{\partial \boldsymbol{W}_{O(i,\cdot)}^{(t)}}, \boldsymbol{o}_j \right\rangle = 0 \pm \mathcal{O}(\tau)$. From (226), similar to (228), we can write

$$
\left\langle -\frac{\partial \mathcal{L}}{\partial \boldsymbol{W}_{O(i,\cdot)}^{(t)}}, \boldsymbol{o}_j \right\rangle - \mathcal{O}\left( \sqrt{\frac{d \log N}{mN}} \right)
$$

$$
\leq \left\langle -\frac{\partial \hat{\mathcal{L}}}{\partial \boldsymbol{W}_{O(i,\cdot)}^{(t)}}, \boldsymbol{o}_j \right\rangle
$$

$$
\leq \left\langle -\frac{\partial \mathcal{L}}{\partial \boldsymbol{W}_{O(i,\cdot)}^{(t)}}, \boldsymbol{o}_j \right\rangle + \mathcal{O}\left( \sqrt{\frac{d \log N}{mN}} \right) + \mathcal{O}(\tau). \tag{255}
$$

Therefore,

$$
\left\langle -\frac{\partial \hat{\mathcal{L}}}{\partial \boldsymbol{W}_{O(i,\cdot)}^{(t)}}, \boldsymbol{o}_j \right\rangle \leq \mathcal{O}\left( \sqrt{\frac{d \log N}{mN}} \right) \text{ for } j \neq 1, 2. \tag{256}
$$

$\square$

### E.3 PROOF OF LEMMA C.3

By symmetry, the proof is analogous to that of Lemma C.1; Please see Appendix E.1.

### E.4 PROOF OF LEMMA C.4

By symmetry, the proof is analogous to that of Lemma C.2; Please see Appendix E.2.

### E.5 PROOF OF LEMMA C.5

*Proof.* The gradient of the loss with respect to $\boldsymbol{w}_\Delta$ for the $n^{\text{th}}$ sample is given by

$$
\frac{\partial \ell}{\partial \boldsymbol{w}_\Delta} = -\frac{z^{(n)}}{L} \cdot \sum_{i=1}^{m} \sum_{l=1}^{L} v_i\, \phi'\left( \boldsymbol{W}_{O(i,\cdot)} \boldsymbol{y}_l^{(n)} \right) \cdot \sum_{s=1}^{l} \left( \boldsymbol{W}_B^\top \boldsymbol{x}_s^{(n)} \right)^\top \left( \boldsymbol{W}_C^\top \boldsymbol{x}_l^{(n)} \right) \left( \boldsymbol{W}_{O(i,\cdot)} \boldsymbol{x}_s^{(n)} \right)
$$

$$
\cdot\, \sigma\left( \boldsymbol{w}_\Delta^\top \boldsymbol{x}_s^{(n)} \right) \cdot \prod_{r=s+1}^{l} \left( 1 - \sigma\left( \boldsymbol{w}_\Delta^\top \boldsymbol{x}_r^{(n)} \right) \right)
$$

$$
\cdot \left[ \left( 1 - \sigma\left( \boldsymbol{w}_\Delta^\top \boldsymbol{x}_s^{(n)} \right) \right) \boldsymbol{x}_s^{(n)} - \sum_{j=s+1}^{l} \left( 1 - \sigma\left( \boldsymbol{w}_\Delta^\top \boldsymbol{x}_j^{(n)} \right) \right) \boldsymbol{x}_j^{(n)} \right]
$$

$$
:= -\frac{z^{(n)}}{L} \cdot \sum_{i=1}^{m} \sum_{l=1}^{L} v_i\, \phi'\left( \boldsymbol{W}_{O(i,\cdot)} \boldsymbol{y}_l^{(n)} \right) \cdot \sum_{s=1}^{l} \boldsymbol{I}_{l,s}^{(n)}. \tag{257}
$$

We define the gradient summand $\boldsymbol{I}_{l,s}^{(n)}$ as

$$
\boldsymbol{I}_{l,s}^{(n)} = \beta_{s,s} \cdot \boldsymbol{x}_s^{(n)} - \sum_{j=s+1}^{l} \beta_{s,j} \boldsymbol{x}_j^{(n)}, \tag{258}
$$

where the coefficients $\beta_{s,s}$ and $\beta_{s,j}$ are given by

$$
\beta_{s,s} = (\boldsymbol{W}_B^\top \boldsymbol{x}_s^{(n)})^\top (\boldsymbol{W}_C^\top \boldsymbol{x}_l^{(n)})(\boldsymbol{W}_{O(i,\cdot)} \boldsymbol{x}_s^{(n)}) \sigma(\boldsymbol{w}_\Delta^\top \boldsymbol{x}_s^{(n)})
$$

$$
\times \left[ \prod_{r=s+1}^{l} \left( 1 - \sigma(\boldsymbol{w}_\Delta^\top \boldsymbol{x}_r^{(n)}) \right) \right] (1 - \sigma(\boldsymbol{w}_\Delta^\top \boldsymbol{x}_s^{(n)})). \tag{259}
$$

and

$$\beta_{s,j} = (\boldsymbol{W}_B^\top \boldsymbol{x}_s^{(n)})^\top (\boldsymbol{W}_C^\top \boldsymbol{x}_l^{(n)})(\boldsymbol{W}_{O(i,\cdot)} \boldsymbol{x}_s^{(n)}) \sigma(\boldsymbol{w}_\Delta^\top \boldsymbol{x}_s^{(n)})$$
$$\times \left[ \prod_{r=s+1}^{l} \left( 1 - \sigma(\boldsymbol{w}_\Delta^\top \boldsymbol{x}_r^{(n)}) \right) \right] (1 - \sigma(\boldsymbol{w}_\Delta^\top \boldsymbol{x}_j^{(n)})). \tag{260}$$

If we consider the gradient of the empirical loss,

$$\frac{\partial \hat{\mathcal{L}}}{\partial \boldsymbol{w}_\Delta} = -\frac{1}{N} \sum_{n=1}^{N} \frac{z^{(n)}}{L} \cdot \sum_{i=1}^{m} \sum_{l=1}^{L} v_i \, \phi' \left( \boldsymbol{W}_{O(i,\cdot)} \boldsymbol{y}_l^{(n)} \right) \cdot \sum_{s=1}^{l} \boldsymbol{I}_{l,s}^{(n)}. \tag{261}$$

We are given that

$$p_1 \leq \langle \boldsymbol{w}_\Delta^{(t)}, \boldsymbol{o}_+ \rangle \leq q_1, \quad \text{and} \quad r_1^* \leq \langle \boldsymbol{W}_{O(i,\cdot)}^{(t+1)^\top}, \boldsymbol{o}_+ \rangle \leq s_1^*. \tag{262}$$

From our initialization, for all $i \in \mathcal{K}^+$, we have $v_i = \frac{1}{\sqrt{m}}$. This gives

$$\left\langle -\frac{\partial \ell}{\partial \boldsymbol{w}_\Delta}, \boldsymbol{o}_+ \right\rangle = \frac{z^{(n)}}{L} \sum_{i=1}^{m} \sum_{l=1}^{L} \frac{1}{\sqrt{m}} \cdot \phi'(\boldsymbol{W}_{O(i,\cdot)} \boldsymbol{y}_l^{(n)}) \sum_{s=1}^{l} \left\langle \boldsymbol{I}_{l,s}^{(n)}, \boldsymbol{o}_+ \right\rangle. \tag{263}$$

Averaging over the training samples, the inner product of the empirical gradient becomes

$$\left\langle -\frac{\partial \hat{\mathcal{L}}}{\partial \boldsymbol{w}_\Delta}, \boldsymbol{o}_+ \right\rangle = \frac{1}{N} \sum_{n=1}^{N} \frac{z^{(n)}}{L} \cdot \sum_{i=1}^{m} \sum_{l=1}^{L} v_i \phi'(\boldsymbol{W}_{O(i,\cdot)} \boldsymbol{y}_l^{(n)}) \cdot \sum_{s=1}^{l} \left\langle \boldsymbol{I}_{l,s}^{(n)}, \boldsymbol{o}_+ \right\rangle$$

$$= \frac{1}{N} \sum_{n:z^{(n)}=+1} \frac{1}{L} \left[ \sum_{i \in \mathcal{K}_+} \sum_{l=1}^{L} \frac{1}{\sqrt{m}} \phi'(\boldsymbol{W}_{O(i,\cdot)} \boldsymbol{y}_l^{(n)}) \sum_{s=1}^{l} \left\langle \boldsymbol{I}_{l,s}^{(n)}, \boldsymbol{o}_+ \right\rangle \right.$$

$$\left. + \sum_{i \in \mathcal{K}_-} \sum_{l=1}^{L} \left( -\frac{1}{\sqrt{m}} \right) \phi'(\boldsymbol{W}_{O(i,\cdot)} \boldsymbol{y}_l^{(n)}) \sum_{s=1}^{l} \left\langle \boldsymbol{I}_{l,s}^{(n)}, \boldsymbol{o}_+ \right\rangle \right]$$

$$+ \frac{1}{N} \sum_{n:z^{(n)}=-1} \frac{-1}{L} \left[ \sum_{i \in \mathcal{K}_+} \sum_{l=1}^{L} \frac{1}{\sqrt{m}} \phi'(\boldsymbol{W}_{O(i,\cdot)} \boldsymbol{y}_l^{(n)}) \sum_{s=1}^{l} \left\langle \boldsymbol{I}_{l,s}^{(n)}, \boldsymbol{o}_+ \right\rangle \right.$$

$$\left. + \sum_{i \in \mathcal{K}_-} \sum_{l=1}^{L} \left( -\frac{1}{\sqrt{m}} \right) \phi'(\boldsymbol{W}_{O(i,\cdot)} \boldsymbol{y}_l^{(n)}) \sum_{s=1}^{l} \left\langle \boldsymbol{I}_{l,s}^{(n)}, \boldsymbol{o}_+ \right\rangle \right]. \tag{264}$$

First, we focus on the contribution from the samples where $z^{(n)} = +1$, for which we seek a lower bound. We analyze the inner terms by considering four cases.

**Case I:** $l = L_1^+, \; s = L_1^+$

Since $l = s$ and $\boldsymbol{x}_s = \boldsymbol{o}_+$, it follows from (258) that

$$\left\langle \boldsymbol{I}_{l,s}^{(n)}, \boldsymbol{o}_+ \right\rangle = \beta_{s,s}. \tag{265}$$

Using (259), with $\boldsymbol{W}_B = \boldsymbol{W}_C = I$ and $\boldsymbol{x}_l = \boldsymbol{x}_s = \boldsymbol{o}_+$, we obtain

$$\left\langle \boldsymbol{I}_{l,s}^{(n)}, \boldsymbol{o}_+ \right\rangle = \beta_{s,s} = \langle \boldsymbol{W}_{O(i,\cdot)}^{(t+1)^\top}, \boldsymbol{o}_+ \rangle \cdot \sigma(\langle \boldsymbol{w}_\Delta^{(t)}, \boldsymbol{o}_+ \rangle) \cdot \left( 1 - \sigma(\langle \boldsymbol{w}_\Delta^{(t)}, \boldsymbol{o}_+ \rangle) \right). \tag{266}$$

Given the conditions in (262), we can write

$$\left\langle \boldsymbol{I}_{l,s}^{(n)}, \boldsymbol{o}_+ \right\rangle \geq (r_1^* - \mathcal{O}(\tau)) \cdot \sigma(p_1 - \mathcal{O}(\tau)) \cdot (1 - \sigma(q_1 + \mathcal{O}(\tau))). \tag{267}$$

We can approximate $\sigma(p_1 - \mathcal{O}(\tau)) \approx \sigma(p_1) - \mathcal{O}(\tau)$ and $1 - \sigma(q_1 + \mathcal{O}(\tau)) \approx 1 - \sigma(q_1) - \mathcal{O}(\tau)$, since $\mathcal{O}(\tau) < \mathcal{O}(\frac{1}{d})$.

Therefore, we obtain

$$
\begin{aligned}
\left\langle \boldsymbol{I}_{l,s}^{(n)}, \boldsymbol{o}_+ \right\rangle &\geq (r_1^* - \mathcal{O}(\tau)) \cdot (\sigma(p_1) - \mathcal{O}(\tau)) \cdot (1 - \sigma(q_1) - \mathcal{O}(\tau)) \\
&\geq r_1^* \cdot \sigma(p_1) \cdot (1 - \sigma(q_1)) - \mathcal{O}(\tau).
\end{aligned}
\tag{268}
$$

**Case II:** $l = L_2^+,\ s = L_2^+$

This configuration yields the same result as in Case I. We again obtain

$$
\left\langle \boldsymbol{I}_{l,s}^{(n)}, \boldsymbol{o}_+ \right\rangle \geq r_1^* \cdot \sigma(p_1) \cdot (1 - \sigma(q_1)) - \mathcal{O}(\tau).
\tag{269}
$$

**Case III:** $l = L_2^+,\ s = L_1^+$ Comparing (259) with (260), we see that the two expressions differ only in their last term. In this setting, $\boldsymbol{x}_j$ equals $\boldsymbol{o}_+$ only when $j = L_2^+$. Consequently, $\boldsymbol{x}_s = \boldsymbol{x}_j = \boldsymbol{o}_+$, which implies $\beta_{s,s} = \beta_{s,j}$. Hence,

$$
\left\langle \boldsymbol{I}_{l,s}^{(n)}, \boldsymbol{o}_+ \right\rangle = \beta_{s,s} - \beta_{s,j} = 0 \pm \mathcal{O}(\tau).
\tag{270}
$$

**Case IV:** Others

For the other token positions, $\left\langle \boldsymbol{I}_{l,s}^{(n)}, \boldsymbol{o}_+ \right\rangle = 0$ due to orthogonality among the features.

Combining the above, the total contribution becomes

$$
\sum_{s=1}^{l} \left\langle \boldsymbol{I}_{l,s}^{(n)}, \boldsymbol{o}_+ \right\rangle \geq 2r_1^* \cdot \sigma(p_1) \cdot (1 - \sigma(q_1)) - \mathcal{O}(\tau).
\tag{271}
$$

We now bound the entire sum over all tokens:

$$
\frac{1}{L} \sum_{l=1}^{L} \frac{1}{\sqrt{m}} \phi' \left( \boldsymbol{W}_{O(i,\cdot)} \boldsymbol{y}_l \right) \sum_{s=1}^{l} \left\langle \boldsymbol{I}_{l,s}^{(n)}, \boldsymbol{o}_+ \right\rangle \geq \frac{1}{L} \sum_{l=1}^{L} \frac{1}{\sqrt{m}} \cdot 1 \cdot 2r_1^* \cdot \sigma(p_1) \cdot (1 - \sigma(q_1)) - \mathcal{O}(\tau). \tag{272}
$$

Let $\rho_t^+ = |\mathcal{W}(t)|$ be the number of contributing neurons. Then the total contribution from the active neurons is lower bounded as

$$
\frac{1}{L} \sum_{i \in \mathcal{K}_+} \sum_{l=1}^{L} v_i \phi' \left( \boldsymbol{W}_{O(i,\cdot)} \boldsymbol{y}_l \right) \sum_{s=1}^{l} \left\langle \boldsymbol{I}_{l,s}^{(n)}, \boldsymbol{o}_+ \right\rangle \geq \frac{2r_1^* \cdot \sigma(p_1) \cdot (1 - \sigma(q_1))}{\sqrt{m}} \cdot \rho_t^+ - \mathcal{O}(\tau).
\tag{273}
$$

Next, we consider $z^{(n)} = -1$ for $i \in \mathcal{K}_+$. For $z^{(n)} = -1$, the negative sample also contains two $\boldsymbol{o}_+$ features.

Similar to the above, we have to consider 4 cases.

**Case I:** $l = L_1^+,\ s = L_1^+$

Since $l = s$, it follows from (258) that

$$
\boldsymbol{I}_{l,s}^{(n)} = \beta_{s,s} \cdot \boldsymbol{x}_l.
\tag{274}
$$

Since $\boldsymbol{x}_l = \boldsymbol{o}_+$, we have

$$
\left\langle \boldsymbol{I}_{l,s}^{(n)}, \boldsymbol{o}_+ \right\rangle = \beta_{s,s}.
\tag{275}
$$

We now seek an upper bound for this contribution. From the initial conditions in (262), we know

$$
\left\langle \boldsymbol{W}_{O(i,\cdot)}^{(t+1)^\top}, \boldsymbol{o}_+ \right\rangle \leq s_1^* + \mathcal{O}(\tau).
\tag{276}
$$

Hence, we obtain

$$\left\langle \boldsymbol{I}_{l,s}^{(n)}, \boldsymbol{o}_+ \right\rangle \le (s_1^* + \mathcal{O}(\tau)) \cdot \sigma(q_1 + \mathcal{O}(\tau)) \cdot (1 - \sigma(p_1 - \mathcal{O}(\tau))). \tag{277}$$

We can approximate $\sigma(q_1 + \mathcal{O}(\tau)) \approx \sigma(q_1) + \mathcal{O}(\tau)$ and $1 - \sigma(p_1 - \mathcal{O}(\tau)) \approx 1 - \sigma(p_1) + \mathcal{O}(\tau)$, since $\mathcal{O}(\tau) < \mathcal{O}(\frac{1}{d})$.

Therefore, we obtain

$$\begin{aligned} \left\langle \boldsymbol{I}_{l,s}^{(n)}, \boldsymbol{o}_+ \right\rangle &\le (s_1^* + \mathcal{O}(\tau)) \cdot (\sigma(q_1) + \mathcal{O}(\tau)) \cdot (1 - \sigma(p_1) + \mathcal{O}(\tau)) \\ &\le s_1^* \cdot \sigma(q_1) \cdot (1 - \sigma(p_1)) + \mathcal{O}(\tau). \end{aligned} \tag{278}$$

**Case II:** $l = L_2^+$, $s = L_2^+$

This configuration yields the same result as in Case I. We again obtain

$$\left\langle \boldsymbol{I}_{l,s}^{(n)}, \boldsymbol{o}_+ \right\rangle \le s_1^* \cdot \sigma(q_1) \cdot (1 - \sigma(p_1)) + \mathcal{O}(\tau). \tag{279}$$

**Case III:** $l = L_2^+$, $s = L_1^+$

In this case, the contribution vanishes:

$$\left\langle \boldsymbol{I}_{l,s}^{(n)}, \boldsymbol{o}_+ \right\rangle = 0 \pm \mathcal{O}(\tau). \tag{280}$$

**Case IV:** Others

For the other token positions, $\left\langle \boldsymbol{I}_{l,s}^{(n)}, \boldsymbol{o}_+ \right\rangle = 0$ due to orthogonality among the features.

The maximum number of such contributing neurons is $\frac{m}{2}$. Therefore, the total contribution is bounded above by

$$\begin{aligned} \frac{1}{L} \sum_{i \in \mathcal{K}_+} \sum_{l=1}^{L} v_i \, \phi' \left( \boldsymbol{W}_{O(i,\cdot)} \boldsymbol{y}_l \right) \sum_{s=1}^{l} \left\langle \boldsymbol{I}_{l,s}^{(n)}, \boldsymbol{o}_+ \right\rangle &\le \frac{2 s_1^* \cdot \sigma(q_1) \cdot (1 - \sigma(p_1))}{\sqrt{m}} \cdot \frac{m}{2} + \mathcal{O}(\tau) \\ &= \sqrt{m} \cdot s_1^* \cdot \sigma(q_1) \cdot (1 - \sigma(p_1)) + \mathcal{O}(\tau). \end{aligned} \tag{281}$$

Thirdly, let us consider the contribution for $z^{(n)} = +1$ from $i \in \mathcal{K}_-$. From our initialization, for $i \in \mathcal{K}^-$, $v_i = -\frac{1}{\sqrt{m}}$. For $z^{(n)} = +1$, we seek an upper bound on the contribution from such neurons.

Let $z^{(n)} = +1$. To maximize the term $\boldsymbol{W}_{O(i,\cdot)} \boldsymbol{x}_s^{(n)}$ in (259), we can consider the token locations which contain $\boldsymbol{o}_-$ features since $\boldsymbol{W}_{O(i,\cdot)}$ has a large component in the $\boldsymbol{o}_-$ direction. Then $\boldsymbol{x}_l = \boldsymbol{o}_- \Rightarrow \boldsymbol{y}_l$ contains the $\boldsymbol{o}_-$ feature.

However, in this case, $\boldsymbol{x}_s = \boldsymbol{o}_- = \boldsymbol{x}_l$, and due to orthogonality,

$$\left\langle \boldsymbol{I}_{l,s}^{(n)}, \boldsymbol{o}_+ \right\rangle = 0. \tag{282}$$

Hence, we only need to consider time steps $l = L_1^+, L_2^+$, where $\boldsymbol{o}_+$ features appear.

Recall that

$$\left\langle -\frac{\partial \ell}{\partial \boldsymbol{w}_\Delta}, \boldsymbol{o}_+ \right\rangle = \frac{1}{L} \sum_{i=1}^{m} \sum_{l=1}^{L} -\frac{1}{\sqrt{m}} \cdot \phi'(\boldsymbol{W}_{O(i,\cdot)} \boldsymbol{y}_l) \sum_{s=1}^{l} \left\langle \boldsymbol{I}_{l,s}^{(n)}, \boldsymbol{o}_+ \right\rangle. \tag{283}$$

We analyze the inner contributions case by case.

**Case I:** $l = L_1^+$, $s = L_1^+$

Given that

$$\boldsymbol{W}_{O(i,\cdot)}\boldsymbol{o}_+ \le \delta_1 + \mathcal{O}\left(\sqrt{\frac{d\log N}{mN}}\right) =: c, \tag{284}$$

we obtain

$$\left\langle \boldsymbol{I}_{l,s}^{(n)}, \boldsymbol{o}_+ \right\rangle \le c \cdot \sigma(q_1) \cdot (1 - \sigma(p_1)) + \mathcal{O}(\tau). \tag{285}$$

**Case II:** $l = L_2^+$, $s = L_2^+$

This configuration yields the same bound:

$$\left\langle \boldsymbol{I}_{l,s}^{(n)}, \boldsymbol{o}_+ \right\rangle \le c \cdot \sigma(q_1) \cdot (1 - \sigma(p_1)) + \mathcal{O}(\tau). \tag{286}$$

**Case III:** $l = L_2^+$, $s = L_1^+$

In this case, the contribution vanishes:

$$\left\langle \boldsymbol{I}_{l,s}^{(n)}, \boldsymbol{o}_+ \right\rangle = 0 \pm \mathcal{O}(\tau). \tag{287}$$

**Case IV:** Others

For the other token positions, $\left\langle \boldsymbol{I}_{l,s}^{(n)}, \boldsymbol{o}_+ \right\rangle = 0$ due to orthogonality among the features.

Thus, the total contribution from each $i \in \mathcal{K}^-$ satisfies

$$\sum_{s=1}^{l} \left\langle \boldsymbol{I}_{l,s}^{(n)}, \boldsymbol{o}_+ \right\rangle \le 2c \cdot \sigma(q_1) \cdot (1 - \sigma(p_1)) + \mathcal{O}(\tau). \tag{288}$$

The maximum number of such contributing neurons is $\frac{m}{2}$, so the full contribution is bounded by

$$\frac{1}{\sqrt{mL}} \sum_{i \in \mathcal{K}_-} \sum_{l=1}^{L} \phi'\left(\boldsymbol{W}_{O(i,\cdot)}\boldsymbol{y}_l\right) \sum_{s=1}^{l} \left\langle \boldsymbol{I}_{l,s}^{(n)}, \boldsymbol{o}_+ \right\rangle \le \frac{2c \cdot \sigma(q_1) \cdot (1 - \sigma(p_1))}{\sqrt{m}} \cdot \frac{m}{2} + \mathcal{O}(\tau)$$
$$= \sqrt{m}c \cdot \sigma(q_1) \cdot (1 - \sigma(p_1)) + \mathcal{O}(\tau). \tag{289}$$

Therefore, the overall contribution is

$$-\frac{1}{\sqrt{mL}} \sum_{i \in \mathcal{K}_-} \sum_{l=1}^{L} \phi'(\boldsymbol{W}_{O(i,\cdot)}\boldsymbol{y}_l) \sum_{s=1}^{l} \left\langle \boldsymbol{I}_{l,s}^{(n)}, \boldsymbol{o}_+ \right\rangle \ge -\sqrt{m}c \cdot \sigma(q_1) \cdot (1 - \sigma(p_1)) - \mathcal{O}(\tau). \tag{290}$$

Finally, we consider $z^{(n)} = -1$ for $i \in \mathcal{K}_-$. For $z^{(n)} = -1$, we want a lower bound since $v_i = -\frac{1}{\sqrt{m}}$.

We could consider $l = L^+ \Rightarrow \boldsymbol{x}_l = \boldsymbol{o}_+$, and write

$$\left\langle \boldsymbol{W}_{O(i,\cdot)}, \boldsymbol{o}_+ \right\rangle \ge \delta_1 - \mathcal{O}\left(\sqrt{\frac{d\log N}{mN}}\right). \tag{291}$$

However, the minimum number of such contributing neurons is not tractable. Thus, if we consider the worst case where $\boldsymbol{W}_{O(i,\cdot)}$ for $i \in \mathcal{K}^-$ does not learn the $\boldsymbol{o}_+$ feature, the obvious lower bound is zero:

$$\frac{1}{L} \sum_{i \in \mathcal{K}_-} \sum_{l=1}^{L} \frac{1}{\sqrt{m}} \phi'\left(\boldsymbol{W}_{O(i,\cdot)}\boldsymbol{y}_l\right) \sum_{s=1}^{l} \left\langle \boldsymbol{I}_{l,s}^{(n)}, \boldsymbol{o}_+ \right\rangle \ge 0. \tag{292}$$

We now combine the bounds for the four terms identified in (264), corresponding to the contributions from: (i) $\mathcal{K}_+$ with $z^{(n)} = +1$ as shown in (273), (ii) $\mathcal{K}_+$ with $z^{(n)} = -1$ as shown in (281), (iii) $\mathcal{K}_-$ with $z^{(n)} = +1$ as shown in (290), and (iv) $\mathcal{K}_-$ with $z^{(n)} = -1$ as shown in (292). We assume

the batch is balanced, so the number of positive and negative samples is equal, with each class contributing $\frac{N}{2}$ samples. Then we have

$$\left\langle -\frac{\partial\hat{\mathcal{L}}}{\partial\boldsymbol{w}_{\Delta}}, \boldsymbol{o}_{+}\right\rangle \geq \frac{1}{2}\left[\frac{2r_1^* \cdot \sigma(p_1)\left(1-\sigma(q_1)\right)}{\sqrt{m}}\cdot\rho_t^+ - \sqrt{m}\cdot c\cdot\sigma(q_1)\left(1-\sigma(p_1)\right)\right.$$

$$\left. -\sqrt{m}\cdot s_1^*\cdot\sigma(q_1)\cdot(1-\sigma(p_1))+0\right] - \mathcal{O}(\tau)$$

$$= \frac{\sigma(p_1)\left(1-\sigma(q_1)\right)r_1^*\cdot\rho_t^+}{\sqrt{m}} - \frac{\sigma(q_1)\left(1-\sigma(p_1)\right)s_1^*\cdot\sqrt{m}}{2} \tag{293}$$

$$-\mathcal{O}\left(\sqrt{\frac{d\log N}{mN}}\right) - \mathcal{O}(\tau). \tag{294}$$

where we have used the fact $\frac{\sqrt{m}}{2}\cdot\sigma(q_1)\left(1-\sigma(p_1)\right)\cdot c = \mathcal{O}\left(\sqrt{\frac{d\log N}{mN}}\right)$ since $c = \mathcal{O}\left(\sqrt{\frac{d\log N}{mN}}\right)$.

$\square$

### E.6 PROOF OF LEMMA C.6

*Proof.* The gradient is given in (257).

Let's consider the alignment with $\boldsymbol{o}_k$ for $k \neq 1, 2$.

$$\left\langle -\frac{\partial\ell}{\partial\boldsymbol{w}_{\Delta}}, \boldsymbol{o}_k\right\rangle = \frac{z^{(n)}}{L}\sum_{i=1}^m\sum_{l=1}^L v_i\cdot\phi'(\boldsymbol{W}_{O(i,\cdot)}\boldsymbol{y}_l^{(n)})\sum_{s=1}^l\left\langle\boldsymbol{I}_{l,s}^{(n)}, \boldsymbol{o}_k\right\rangle \tag{295}$$

From our initialization, for all $i \in \mathcal{K}^+$, we have $v_i = \frac{1}{\sqrt{m}}$.

We first consider the case $z^{(n)} = +1$ for $i \in \mathcal{K}^+$. Since $\boldsymbol{W}_{O(i,\cdot)}$, for $i \in \mathcal{K}^+$ has a large $\boldsymbol{o}_+$ component, we have to consider the token features with $\boldsymbol{o}_+$. For $z^{(n)} = +1$, only when $l = L_2^+$, $s = L_1^+$ we have $\boldsymbol{x}_l = \boldsymbol{x}_s = \boldsymbol{o}_+$. Therefore, $\boldsymbol{W}_{O(i,\cdot)}\boldsymbol{x}_s$ is significant. Hence, we have

$$\left\langle\boldsymbol{I}_{l,s}^{(n)}, \boldsymbol{o}_k\right\rangle$$

$$= -\sum_{j=s+1}^l \beta_{s,j}\langle\boldsymbol{x}_j^{(n)}, \boldsymbol{o}_k\rangle$$

$$\leq -\beta_{s,s+1} \qquad (\text{Assuming W.L.O.G. } \boldsymbol{x}_{s+1}^{(n)} = \boldsymbol{o}_k)$$

$$\leq -\langle\boldsymbol{W}_{O(i,\cdot)}^{(t+1)\top}, \boldsymbol{o}_+\rangle\cdot\sigma(\langle\boldsymbol{w}_{\Delta}^{(t)}, \boldsymbol{o}_+\rangle)\cdot\left(1-\sigma(\langle\boldsymbol{w}_{\Delta}^{(t)}, \boldsymbol{o}_+\rangle)\right)\cdot\left(1-\sigma(\langle\boldsymbol{w}_{\Delta}^{(t)}, \boldsymbol{o}_k\rangle)\right)^{\Delta L_{o_+}^+}. \tag{296}$$

Using the the conditions in (262), we can write

$$\left\langle\boldsymbol{I}_{l,s}^{(n)}, \boldsymbol{o}_k\right\rangle \leq (-r_1^* + \mathcal{O}(\tau))\cdot\sigma(p_1 + \mathcal{O}(\tau))\cdot(1-\sigma(q_1 - \mathcal{O}(\tau)))\cdot(1-\sigma(q_2 - \mathcal{O}(\tau)))^{\Delta L_{o_+}^+}. \tag{297}$$

We can approximate $\sigma(p_1 + \mathcal{O}(\tau)) \approx \sigma(p_1) + \mathcal{O}(\tau)$, $1 - \sigma(q_1 - \mathcal{O}(\tau)) \approx 1 - \sigma(q_1) + \mathcal{O}(\tau)$ and $1 - \sigma(q_2 - \mathcal{O}(\tau)) \approx 1 - \sigma(q_2) + \mathcal{O}(\tau)$, since $\mathcal{O}(\tau) < \mathcal{O}(\frac{1}{d})$.

Hence, we obtain

$$\frac{1}{L}\sum_{l=1}^L\frac{1}{\sqrt{m}}\cdot\phi'\left(\boldsymbol{W}_{O(i,\cdot)}\boldsymbol{y}_l^{(n)}\right)\sum_{s=1}^l\left\langle\boldsymbol{I}_{l,s}^{(n)}, \boldsymbol{o}_k\right\rangle$$

$$\leq\frac{1}{L}\sum_{l=1}^L\frac{1}{\sqrt{m}}\cdot 1\cdot\left[-r_1^*\cdot\sigma(p_1)\cdot(1-\sigma(q_1))\cdot(1-\sigma(q_2))^{\Delta L_{o_+}^+}\right] + \mathcal{O}(\tau). \tag{298}$$

Let $\rho_t^+ = |\mathcal{W}(t)|$ be the number of contributing neurons. Then the total contribution from $\mathcal{K}_+$ neurons is bounded as

$$\frac{1}{L} \sum_{i \in \mathcal{K}_+} \sum_{l=1}^{L} v_i \cdot \phi'\left(\boldsymbol{W}_{O(i,\cdot)}\boldsymbol{y}_l^{(n)}\right) \sum_{s=1}^{l} \left\langle \boldsymbol{I}_{l,s}^{(n)}, \boldsymbol{o}_k \right\rangle \leq -\frac{r_1^*}{\sqrt{m}} \cdot \sigma(p_1)\left(1 - \sigma(q_1)\right) \tag{299}$$
$$\cdot \left(1 - \sigma(q_2)\right)^{\Delta L_{o_+}^+} \cdot \rho_t^+ + \mathcal{O}(\tau).$$

Next, we consider $z^{(n)} = -1$ for $i \in \mathcal{K}_+$. Since $\left\langle \boldsymbol{I}_{l,s}^{(n)}, \boldsymbol{o}_k \right\rangle < 0$, we require a lower bound for this.

$$\left\langle \boldsymbol{I}_{l,s}^{(n)}, \boldsymbol{o}_k \right\rangle$$
$$= -\sum_{j=s+1}^{l} \beta_{s,j} \langle \boldsymbol{x}_j^{(n)}, \boldsymbol{o}_k \rangle$$
$$\gtrsim -\langle \boldsymbol{W}_{O(i,\cdot)}^{(t+1)\top}, \boldsymbol{o}_+ \rangle \cdot \sigma(\langle \boldsymbol{w}_\Delta^{(t)}, \boldsymbol{o}_+ \rangle) \cdot \left(1 - \sigma(\langle \boldsymbol{w}_\Delta^{(t)}, \boldsymbol{o}_+ \rangle)\right) \cdot \left(1 - \sigma(\langle \boldsymbol{w}_\Delta^{(t)}, \boldsymbol{o}_k \rangle)\right)^{\Delta L_{o_+}^-}. \tag{300}$$

Using the the conditions in (262), we can write

$$\left\langle \boldsymbol{I}_{l,s}^{(n)}, \boldsymbol{o}_k \right\rangle \gtrsim -s_1^* \cdot \sigma(q_1) \cdot \left(1 - \sigma(p_1)\right) \cdot \left(1 - \sigma(p_2)\right)^{\Delta L_{o_+}^-}. \tag{301}$$

$$\frac{1}{L} \sum_{i \in \mathcal{K}_+} \sum_{l=1}^{L} v_i \cdot \phi'\left(\boldsymbol{W}_{O(i,\cdot)}\boldsymbol{y}_l^{(n)}\right) \sum_{s=1}^{l} \left\langle \boldsymbol{I}_{l,s}^{(n)}, \boldsymbol{o}_k \right\rangle \tag{302}$$
$$\gtrsim -\frac{s_1^*}{\sqrt{m}} \cdot \sigma(q_1)\left(1 - \sigma(p_1)\right)\left(1 - \sigma(p_2)\right)^{\Delta L_{o_+}^-} \cdot \rho_t^+.$$

Since $\Delta L_{o_+}^- \gg \Delta L_{o_+}^+$, this term is negligible which leads to

$$\frac{1}{L} \sum_{i \in \mathcal{K}_+} \sum_{l=1}^{L} v_i \cdot \phi'\left(\boldsymbol{W}_{O(i,\cdot)}\boldsymbol{y}_l^{(n)}\right) \sum_{s=1}^{l} \left\langle \boldsymbol{I}_{l,s}^{(n)}, \boldsymbol{o}_k \right\rangle \geq -\mathcal{O}\left(\left(1 - \sigma(p_2)\right)^{\Delta L_{o_+}^-}\right) \approx 0. \tag{303}$$

Thirdly, we consider the case $i \in \mathcal{K}_-$, for $z^{(n)} = -1$. Similar to (296) and (297), when $l = L_2^-$, $s = L_1^-$ the contribution is significant.

$$\left\langle \boldsymbol{I}_{l,s}^{(n)}, \boldsymbol{o}_k \right\rangle \leq -r_1^* \cdot \sigma(p_1) \cdot \left(1 - \sigma(q_1)\right) \cdot \left(1 - \sigma(q_2)\right)^{\Delta L_{o_-}^-} + \mathcal{O}(\tau). \tag{304}$$

Hence, we obtain

$$\frac{1}{L} \sum_{l=1}^{L} \frac{1}{\sqrt{m}} \cdot \phi'(\boldsymbol{W}_{O(i,\cdot)}\boldsymbol{y}_l^{(n)}) \sum_{s=1}^{l} \left\langle \boldsymbol{I}_{l,s}^{(n)}, \boldsymbol{o}_k \right\rangle \tag{305}$$
$$\leq \frac{1}{L} \sum_{l=1}^{L} \frac{1}{\sqrt{m}} \cdot 1 \cdot \left[-r_1^* \cdot \sigma(p_1) \cdot \left(1 - \sigma(q_1)\right) \cdot \left(1 - \sigma(q_2)\right)^{\Delta L_{o_-}^-}\right] + \mathcal{O}(\tau))$$

Let $\rho_t^- = |\mathcal{U}(t)|$ be the number of contributing neurons. Then the total contribution from $\mathcal{K}_-$ neurons is bounded as

$$\frac{1}{L} \sum_{i \in \mathcal{K}_-} \sum_{l=1}^{L} v_i \, \phi'\left(\boldsymbol{W}_{O(i,\cdot)}\boldsymbol{y}_l^{(n)}\right) \sum_{s=1}^{l} \left\langle \boldsymbol{I}_{l,s}^{(n)}, \boldsymbol{o}_k \right\rangle \leq -\frac{r_1^*}{\sqrt{m}} \sigma(p_1)\left(1 - \sigma(q_1)\right)\left(1 - \sigma(q_2)\right)^{\Delta L_{o_-}^-} \rho_t^-$$
$$+ \mathcal{O}(\tau). \tag{306}$$

Finally, we consider $i \in \mathcal{K}_-$ for $z^{(n)} = +1$. Following the same approach as in (300) to (302), we can write

$$\frac{1}{L} \sum_{i \in \mathcal{K}_-} \sum_{l=1}^{L} v_i \cdot \phi'\left(\boldsymbol{W}_{O(i,\cdot)} \boldsymbol{y}_l^{(n)}\right) \sum_{s=1}^{l} \left\langle \boldsymbol{I}_{l,s}^{(n)}, \boldsymbol{o}_k \right\rangle$$

$$\gtrsim -\frac{s_1^*}{\sqrt{m}} \cdot \sigma(q_1) \left(1 - \sigma(p_1)\right) \left(1 - \sigma(p_2)\right)^{\Delta L_{o_-}^+} \cdot \rho_t^-. \tag{307}$$

Since $\Delta L_{o_-}^+ \gg \Delta L_{o_-}^-$, this term is negligible which leads to

$$\frac{1}{L} \sum_{i \in \mathcal{K}_-} \sum_{l=1}^{L} v_i \cdot \phi'\left(\boldsymbol{W}_{O(i,\cdot)} \boldsymbol{y}_l^{(n)}\right) \sum_{s=1}^{l} \left\langle \boldsymbol{I}_{l,s}^{(n)}, \boldsymbol{o}_k \right\rangle \geq -\mathcal{O}\left((1 - \sigma(p_2))^{\Delta L_{o_-}^+}\right) \approx 0. \tag{308}$$

Putting it together, We know

$$\left\langle -\frac{\partial \hat{\mathcal{L}}}{\partial \boldsymbol{w}_\Delta}, \boldsymbol{o}_k \right\rangle = \frac{1}{N} \sum_{n=1}^{N} \frac{z^{(n)}}{L} \cdot \sum_{i=1}^{m} \sum_{l=1}^{L} v_i \phi'(\boldsymbol{W}_{O(i,\cdot)} \boldsymbol{y}_l^{(n)}) \cdot \sum_{s=1}^{l} \left\langle \boldsymbol{I}_{l,s}^{(n)}, \boldsymbol{o}_k \right\rangle$$

$$= \frac{1}{N} \sum_{n:z^{(n)}=+1} \frac{1}{L} \left[ \sum_{i \in \mathcal{K}_+} \sum_{l=1}^{L} \frac{1}{\sqrt{m}} \phi'(\boldsymbol{W}_{O(i,\cdot)} \boldsymbol{y}_l^{(n)}) \sum_{s=1}^{l} \left\langle \boldsymbol{I}_{l,s}^{(n)}, \boldsymbol{o}_k \right\rangle \right.$$

$$\left. + \sum_{i \in \mathcal{K}_-} \sum_{l=1}^{L} \left(-\frac{1}{\sqrt{m}}\right) \phi'(\boldsymbol{W}_{O(i,\cdot)} \boldsymbol{y}_l^{(n)}) \sum_{s=1}^{l} \left\langle \boldsymbol{I}_{l,s}^{(n)}, \boldsymbol{o}_k \right\rangle \right]$$

$$+ \frac{1}{N} \sum_{n:z^{(n)}=-1} \frac{-1}{L} \left[ \sum_{i \in \mathcal{K}_+} \sum_{l=1}^{L} \frac{1}{\sqrt{m}} \phi'(\boldsymbol{W}_{O(i,\cdot)} \boldsymbol{y}_l^{(n)}) \sum_{s=1}^{l} \left\langle \boldsymbol{I}_{l,s}^{(n)}, \boldsymbol{o}_k \right\rangle \right.$$

$$\left. + \sum_{i \in \mathcal{K}_-} \sum_{l=1}^{L} \left(-\frac{1}{\sqrt{m}}\right) \phi'(\boldsymbol{W}_{O(i,\cdot)} \boldsymbol{y}_l^{(n)}) \sum_{s=1}^{l} \left\langle \boldsymbol{I}_{l,s}^{(n)}, \boldsymbol{o}_k \right\rangle \right]. \tag{309}$$

We now combine the bounds for the two terms identified in equation (309), corresponding to the contributions from: (i) $\mathcal{K}_+$ with $z^{(n)} = +1$ (299), (ii) $\mathcal{K}_+$ with $z^{(n)} = -1$ (307), (iii) $\mathcal{K}_-$ with $z^{(n)} = +1$ (302), and (iv) $\mathcal{K}_-$ with $z^{(n)} = -1$ (306). We assume the batch is balanced, so the number of positive and negative samples is equal, with each class contributing $\frac{N}{2}$ samples. Then we have

$$\left\langle -\frac{\partial \hat{\mathcal{L}}}{\partial \boldsymbol{w}_\Delta^{(t)}}, \boldsymbol{o}_k \right\rangle$$

$$\leq -\frac{r_1^*}{2\sqrt{m}} \sigma(p_1) \left(1 - \sigma(q_1)\right) \left[ (1 - \sigma(q_2))^{\Delta L_{o_+}^+} \rho_t^+ + (1 - \sigma(q_2))^{\Delta L_{o_-}^-} \rho_t^- \right]$$

$$+ \frac{s_1^*}{\sqrt{m}} \cdot \sigma(q_1) \left(1 - \sigma(p_1)\right) \left[ \mathcal{O}\left((1 - \sigma(p_2))^{\Delta L_{o_-}^+}\right) \cdot \rho_t^- + \mathcal{O}\left((1 - \sigma(p_2))^{\Delta L_{o_+}^-}\right) \cdot \rho_t^+ \right] \tag{310}$$

$$+ \mathcal{O}(\tau) \tag{311}$$

$$\left\langle -\frac{\partial \hat{\mathcal{L}}}{\partial \boldsymbol{w}_\Delta^{(t)}}, \boldsymbol{o}_k \right\rangle$$

$$\leq -\frac{r_1^*}{2\sqrt{m}} \sigma(p_1) \left(1 - \sigma(q_1)\right) \left[ (1 - \sigma(q_2))^{\Delta L_{o_+}^+} \rho_t^+ + (1 - \sigma(q_2))^{\Delta L_{o_-}^-} \rho_t^- \right]$$

$$+ \mathcal{O}\left((1 - \sigma(p_2))^{\Delta L_{o_+}^-}\right) + \mathcal{O}\left((1 - \sigma(p_2))^{\Delta L_{o_-}^+}\right) + \mathcal{O}(\tau) \tag{312}$$

From (303) and (308), we can conclude

$$\left\langle -\frac{\partial \hat{\mathcal{L}}}{\partial \boldsymbol{w}_\Delta^{(t)}}, \boldsymbol{o}_k \right\rangle \leq -\frac{r_1^*}{2\sqrt{m}} \sigma(p_1)\left(1 - \sigma(q_1)\right)\left[(1-\sigma(q_2))^{\Delta L_{o+}^+} \rho_t^+ + (1-\sigma(q_2))^{\Delta L_{o-}^-} \rho_t^-\right]$$
$$+ \mathcal{O}(\tau)). \tag{313}$$

$\square$

## F  EXTENSION TO MULTI-CLASS CLASSIFICATION

Consider the classification problem with four classes, where each example is assigned a label $\boldsymbol{z} = (z_1, z_2) \in \{+1, -1\}^2$ representing four distinct classes. Similarly to the binary setting, there exist four orthogonal discriminative patterns. In the output layer, the scalar coefficient $v_i$ associated with hidden neuron $i$ is replaced by a two-dimensional vector $\boldsymbol{v}_i \in \mathbb{R}^2$.

Hence, we define the model output as

$$\boldsymbol{F}(\boldsymbol{X}) = \frac{1}{L} \sum_{l=1}^{L} \sum_{i=1}^{m} \boldsymbol{v}_i \, \phi\big(\boldsymbol{W}_{O(i,\cdot)} \boldsymbol{y}_l(\boldsymbol{X})\big). \tag{314}$$

$$F_1(\boldsymbol{X}^{(n)}) = \frac{1}{L} \sum_{l=1}^{L} \sum_{i=1}^{m} (\boldsymbol{v}_i)_1 \, \phi\big(\boldsymbol{W}_{O(i,\cdot)} \boldsymbol{y}_l(\boldsymbol{X}^{(n)})\big), \tag{315}$$

$$F_2(\boldsymbol{X}^{(n)}) = \frac{1}{L} \sum_{l=1}^{L} \sum_{i=1}^{m} (\boldsymbol{v}_i)_2 \, \phi\big(\boldsymbol{W}_{O(i,\cdot)} \boldsymbol{y}_l(\boldsymbol{X}^{(n)})\big). \tag{316}$$

The dataset can be divided into four groups as

$$\begin{aligned} \mathcal{D}_1 &= \{(\boldsymbol{X}^{(n)}, \boldsymbol{z}^{(n)}) \mid \boldsymbol{z}^{(n)} = (1,1)\}, \\ \mathcal{D}_2 &= \{(\boldsymbol{X}^{(n)}, \boldsymbol{z}^{(n)}) \mid \boldsymbol{z}^{(n)} = (1,-1)\}, \\ \mathcal{D}_3 &= \{(\boldsymbol{X}^{(n)}, \boldsymbol{z}^{(n)}) \mid \boldsymbol{z}^{(n)} = (-1,1)\}, \\ \mathcal{D}_4 &= \{(\boldsymbol{X}^{(n)}, \boldsymbol{z}^{(n)}) \mid \boldsymbol{z}^{(n)} = (-1,-1)\}. \end{aligned} \tag{317}$$

The loss function for data $(\boldsymbol{X}^{(n)}, \boldsymbol{z}^{(n)})$ is

$$\text{Loss}(\boldsymbol{X}^{(n)}, \boldsymbol{z}^{(n)}) = \max\left\{1 - \boldsymbol{z}^{(n)\top} \boldsymbol{F}(\boldsymbol{X}^{(n)}), 0\right\}. \tag{318}$$

Since $\boldsymbol{v}_i \in \{\pm\frac{1}{\sqrt{m}}\}^2$, we divide neurons into four groups:

$$\begin{aligned} \mathcal{W}_1 &= \{i : \boldsymbol{v}_i = \tfrac{1}{\sqrt{m}}(1,1)\}, \\ \mathcal{W}_2 &= \{i : \boldsymbol{v}_i = \tfrac{1}{\sqrt{m}}(1,-1)\}, \\ \mathcal{W}_3 &= \{i : \boldsymbol{v}_i = \tfrac{1}{\sqrt{m}}(-1,1)\}, \\ \mathcal{W}_4 &= \{i : \boldsymbol{v}_i = \tfrac{1}{\sqrt{m}}(-1,-1)\}. \end{aligned} \tag{319}$$

For neuron $i$, the gradient decomposes as

$$\frac{\partial \text{Loss}}{\partial \boldsymbol{W}_{O(i,\cdot)}} = -z_1^{(n)} \frac{\partial F_1(\boldsymbol{X}^{(n)})}{\partial \boldsymbol{W}_{O(i,\cdot)}} - z_2^{(n)} \frac{\partial F_2(\boldsymbol{X}^{(n)})}{\partial \boldsymbol{W}_{O(i,\cdot)}}. \tag{320}$$

Let $\boldsymbol{o}_1, \boldsymbol{o}_2, \boldsymbol{o}_3, \boldsymbol{o}_4$ denote the four discriminative directions. Consider $i \in \mathcal{W}_2$, i.e. $\boldsymbol{v}_i = \frac{1}{\sqrt{m}}(1,-1)$. Projecting the gradient onto $\boldsymbol{o}_2$, for any $(\boldsymbol{X}^{(n)}, \boldsymbol{z}^{(n)}) \in \mathcal{D}_2$ we obtain

$$-\left\langle \frac{\partial \text{Loss}}{\partial \boldsymbol{W}_{O(i,\cdot)}}, \boldsymbol{o}_2 \right\rangle \approx \frac{2}{\sqrt{m}} \|\boldsymbol{o}_2\|^2 \; > \; 0, \tag{321}$$

showing GD moves $\boldsymbol{W}_{O(i,\cdot)}$ toward $\boldsymbol{o}_2$.

For samples from the other classes:

$$
\begin{aligned}
(\boldsymbol{X}^{(n)}, \boldsymbol{z}^{(n)}) \in \mathcal{D}_1: \quad &-\left\langle \frac{\partial \mathrm{Loss}}{\partial \boldsymbol{W}_{O(i,\cdot)}}, \boldsymbol{o}_1 \right\rangle \approx 0, \\
(\boldsymbol{X}^{(n)}, \boldsymbol{z}^{(n)}) \in \mathcal{D}_3: \quad &-\left\langle \frac{\partial \mathrm{Loss}}{\partial \boldsymbol{W}_{O(i,\cdot)}}, \boldsymbol{o}_3 \right\rangle \approx -\frac{2}{\sqrt{m}} \|\boldsymbol{o}_3\|^2, \\
(\boldsymbol{X}^{(n)}, \boldsymbol{z}^{(n)}) \in \mathcal{D}_4: \quad &-\left\langle \frac{\partial \mathrm{Loss}}{\partial \boldsymbol{W}_{O(i,\cdot)}}, \boldsymbol{o}_4 \right\rangle \approx 0.
\end{aligned}
\tag{322}
$$

Thus, for $i \in \mathcal{W}_2$, the update direction aligns with $\boldsymbol{o}_2$, and similarly neurons in $\mathcal{W}_1, \mathcal{W}_3, \mathcal{W}_4$ align with $\boldsymbol{o}_1, \boldsymbol{o}_3, \boldsymbol{o}_4$ respectively. Similarly, we can analyze the gradient dynamics of the gating vector $\boldsymbol{w}_\Delta$.

