# OpenReview forum: "A Theoretical Analysis of Mamba’s Training Dynamics: Filtering Relevant Features for Generalization in State Space Models"
_ICLR.cc/2026/Conference — ICLR 2026 Poster_

### Official Review · Reviewer_T7xH · 2025-10-31

**Soundness:** 3
**Presentation:** 3
**Contribution:** 2
**Rating:** 6
**Confidence:** 3

**Summary:**

The paper presents a formal theoretical study of the Mamba architecture. It analyzes a simplified single-layer Mamba followed by a two-layer MLP trained with gradient descent on two synthetic data regimes — a majority-voting model and a locality-structured model — where class labels depend on subsets of relevant features within the input sequence. It derives non-asymptotic convergence and generalization guarantees, proving that Mamba’s gating mechanism learns to amplify class-relevant features while suppressing irrelevant ones. The analysis links learning efficiency and generalization to interpretable data parameters such as signal-to-noise ratio and the fraction of informative tokens. Synthetic experiments confirm these trends qualitatively.

**Strengths:**

1. The paper is well-written, clearly structured, and theoretical results are supported by rigorous proofs with proof intuitions given in the main text.
2. The paper is well-situated in the literature. I would only suggest to add [1,2], when discussing the connection of SSMs and Transformers in L147-154, for additional reading on how Mamba relates to attention, altough not critical.
3. The theoretical analysis of the learning process of Mamba is novel and timely, as Mamba has become an alternative to Transformers.
4. The results are convincing, however their generality is not assessed yet.
5. The paper does a good job in linking the empirical findings to the theoretical results, supporting each statement with empirical evidence.

[1] Dao & Gu (2024), "Transformers are SSMs: Generalized Models and Efficient Algorithms Through Structured State Space Duality", https://arxiv.org/abs/2405.21060

[2] Sieber et al. (2024), "Understanding the differences in Foundation Models: Attention, State Space Models, and Recurrent Neural Networks", https://arxiv.org/abs/2405.15731

**Weaknesses:**

1. The main drawback of the paper is its limited scope, both in its model selection and the application setting. While chosing a single model (in this case Mamba) for such a study is reasonable, I would have expected a broader application setting, i.e., not limiting the study to binary classification.
2. The investigated data models are purely theoretical and are not representative of more realistic data models, e.g. the Gaussian distribution assumption is not realistic for practical applications. Having a discussion on this point in the paper would be appreciated.
3. The theoretical results are not novel in itself and are mainly extensions to the Mamba specifics. Altough I think this is not a major problem, but it reduces the originality of the contribution.
4. The paper could do a better job at clarifying its limitations, e.g. with a paragraph in the conclusion.


Minor points:
- The short paragraphs before lemmas/theorems already discuss the results of the lemmas/theorems before they are stated. It would be better for the flow to reserve this discussion for after the lemma/theorem and only give a brief introduction to which results is shown next.
- The remarks after the results are nice, but should not be wrapped in a proof environment, i.e., should not have a square at the end.
- the empirical results provide no quantitative comparison of actual to theoretical convergence rates and are purely qualitative. Including, such a comparison/result would strengthen the paper.

**Questions:**

1. Have you observed any empirical evidence suggesting these theoretical results hold beyond your simplified setup? Or how do you think these results fair in practice?
2. Can your analysis or conclusions be generalized to more complex or non-Gaussian data distributions (e.g., correlated, heavy-tailed, or structured inputs as in language or vision)? If not, what aspects of Mamba’s behavior might your theory fail to capture?

---

> ### Author Response · Authors · 2025-11-21
> **Point-to-point response to Reviewer T7xH - part 1**
>
> # Point-to-point response to Reviewer T7xH
>
> >(W1) "While chosing a single model (in this case Mamba) for such a study is reasonable, I would have expected a broader application setting, i.e., not limiting the study to binary classification."
> >
> **Response to W1 (Scope and application setting).**
> Thank you for raising this concern. We would like to clarify that it is indeed challenging to provide completely generic results that cover all types of learning problems. Following standard practice in training-dynamics analysis, we use the binary classification setting to illustrate the theoretical insights. Nevertheless, to address your concern, we have conducted additional studies, including (i) a brief illustration of how our framework can be extended to the multi-class classification setting, and (ii) new numerical experiment on multi-class classification using the full Mamba model, demonstrating that our theoretical insights generalize to broader learning problems.
>
> **First, binary classification is the standard practice in existing training-dynamics analysis.** Most existing theoretical studies (see, e.g., [R1, R2, R4]) on training dynamics adopt the binary case, as it allows clean derivations while still capturing the essential feature-learning behavior of deep models.
>
> **Second, we have included a brief discussion of the extension to a multi-class classification setting.** For instance, in the 4-class case, the scalar output of the model is replaced by a 2-dimensional vector, and the gradient dynamics continue to move neurons toward their corresponding class-discriminative directions, mirroring the behavior observed in the binary setting.
> Please see Appendix F for the full details of this extension.
>
> **Third, a numerical experiment on multi-class classification demonstrates that our theoretical insights can generalize to this setting as well.** Following the same 4-class formulation described in Appendix F, we generated a synthetic dataset with four discriminative feature directions and trained a full Mamba model on this setting. The learned hidden-layer weights progressively align with their corresponding class-relevant feature directions while remaining nearly orthogonal to the irrelevant directions, mirroring the behavior predicted by our theory and observed in the binary case.
>  We believe these results could further justify the applicability of our theoretical insights beyond the binary setting. Please see [Figure A9] (https://imgur.com/a/6hrZIYv).
>
> >(W2) "The investigated data models are purely theoretical and are not representative of more realistic data models, e.g. the Gaussian distribution assumption is not realistic for practical applications."
>
> **Response to W2 (Validity of the data model).**
> We kindly note that the Gaussian assumption applies only to the *weight initialization*, not to the data distribution itself, as clarified in [GR3](https://openreview.net/forum?id=hvpKqEYJjj&noteId=bclldrQMeS).
> Please see [GR3](https://openreview.net/forum?id=hvpKqEYJjj&noteId=bclldrQMeS) for our detailed response on the practicality and interpretation of the majority-voting and locality-structured data models.

---

> ### Author Response · Authors · 2025-11-21
> **Point-to-point response to Reviewer T7xH - part 2**
>
> >(W3) "The theoretical results are not novel in itself and are mainly extensions to the Mamba specifics. Although I think this is not a major problem, but it reduces the originality of the contribution."
>
> **Response to W3 (novelty).**
> We appreciate the reviewer’s comment that this is not a major issue. However, to avoid any potential misalignment among reviewers and readers, we would like to take this opportunity to clarify several points. We suspect that the reviewer may be suggesting that our setting appears similar to existing works. While our framework is indeed informed by prior studies under the feature learning framework, training dynamics are highly dependent on the underlying architecture. To the best of our knowledge, no prior work explicitly analyzes the role of gating vectors in Mamba (or in closely related models with similar gating mechanisms), nor how they interact with MLP weights to shape the learning process. If the reviewer has a specific existing work in mind whose proofs appear similar, we would be grateful if they could point it out, and we would be happy to clarify the differences.
>
> To the best of our knowledge, the closest work to ours is [R1], which analyzes the training dynamics of shallow Vision Transformers under a majority-voting data model. While we also adopt the majority-voting data model, we additionally introduce a new locality-based data model. More importantly, even under the same data model, the architectural differences prevent a simple extension of the proof from [R1] to our setting. Specifically, the key differences are:
> (i) Mamba introduces an input-dependent gating mechanism that dynamically selects and modulates token contributions, a behavior entirely absent in self-attention; and
> (ii) this gating enters multiplicatively through the recurrent update, generating token–interaction patterns and gradient structures that require analytical tools beyond those used in prior work.
>
> >(W4) "The paper could do a better job at clarifying its limitations, e.g. with a paragraph in the conclusion"
>
> **Response to W4 (limitations paragraph).**
> Thank you very much for the suggestion. We have added a dedicated paragraph in the conclusion explicitly outlining the main limitations of our work, including the simplified Mamba setting and data models, and clarifying directions for future extensions.
>
> *"Finally, we note some limitations of our work. First, our theoretical analysis focuses on a simplified Mamba setting that abstracts away practical components such as depth, multiple heads, residual connections, and layer normalization. Second, our data model, while standard in theoretical studies, also simplifies real-world sequence structures. Extending the analysis to more realistic multi-layer and multi-head Mamba architectures, richer data models, and alternative designs such as gated Transformers or hybrid Mamba–Transformer frameworks
> remains an important direction for future work."*
>
> >(Q1) "Have you observed any empirical evidence suggesting these theoretical results hold beyond your simplified setup? Or how do you think these results fair in practice?"
>
> **Response to Q1 (Empirical support beyond the simplified model).**
> Yes, we have observed consistent empirical evidence. Our additional experiments on practical multi-layer Mamba models closely align with the theoretical predictions. For a detailed summary, please see [GR2](https://openreview.net/forum?id=hvpKqEYJjj&noteId=rm0XF7CM4I).
>
> >(Q2) "Can your analysis or conclusions be generalized to more complex or non-Gaussian data distributions?"
>
> **Response to Q2 (Generality of data assumptions).**
> Thank you for raising this point. The question arises from a small misunderstanding: our analysis does *not* assume the data itself is Gaussian; only the initialization is, which is standard. We provide a clear explanation of this in [GR3](https://openreview.net/forum?id=hvpKqEYJjj&noteId=bclldrQMeS).
>
> >Response to Minor Points.
>
> (1) Thank you, we have revised the paper by moving the result discussions to after each lemma/theorem.
>
> (2) We have removed the proof-style formatting from the remarks as suggested.
>
> (3) Thank you for the suggestion. A quantitative result is provided in [Figure A8] (https://imgur.com/a/AGv1iI6).
>
> [R1] Li, Hongkang, et al. “A Theoretical Understanding of Shallow Vision Transformers: Learning,
> Generalization, and Sample Complexity,” International Conference on Learning Representations
> (ICLR), 2023.
>
> [R2] Zhang, Shuai, et al. “Joint Edge-Model Sparse Learning is Provably Efficient for Graph Neural
> Networks,” International Conference on Learning Representations (ICLR), 2023.
>
> [R4] Li, Hongkang, et al. “How Do Nonlinear Transformers Learn and Generalize in In-Context
> Learning?” International Conference on Machine Learning (ICML), 2024.

---

> > ### Author Response · Authors · 2025-11-26
> > **Gentle Reminder on Rebuttal Discussion**
> >
> > Dear Reviewer T7xH,
> >
> > Thank you again for your detailed review. As the discussion deadline is approaching, we believe that our rebuttal addressed your main concerns by (i) extending the scope beyond binary classification with a multi-class formulation and new full-model experiments, (ii) clarifying the misunderstanding regarding the Gaussian assumption (which applies only to initialization) and explaining the practicality of our structured data models, (iii) refining our discussion of novelty and adding explicit limitations in the conclusion, and (iv) addressing all minor comments with the suggested improvements.
> >
> > If there are any remaining concerns that you feel were not fully addressed, please let us know, we would be happy to clarify before the discussion concludes.
> >
> > Thank you for your time.

---

### Official Review · Reviewer_rcjx · 2025-11-01

**Soundness:** 3
**Presentation:** 2
**Contribution:** 3
**Rating:** 6
**Confidence:** 3

**Summary:**

This paper claims to provide the first theoretical study of the learning and generalization dynamics of the Mamba architecture, a selective state space model (SSM) that has recently emerged as an efficient alternative to Transformers. The authors analyze a simplified single-layer Mamba block with input-dependent gating followed by a two-layer MLP trained via gradient descent under two structured data regimes—majority-voting and locality-structured sequences containing both class-relevant and irrelevant tokens. They prove non-asymptotic sample complexity and convergence guarantees, showing that Mamba generalizes efficiently when class-relevant signals are strong or locally concentrated, and that its gating vector naturally aligns with informative features while suppressing noisy ones, formalizing its feature-selection role. Synthetic experiments align with theoretical insights.

**Strengths:**

The paper’s primary strength lies in its original theoretical treatment of Mamba, offering the first formal generalization and convergence analysis for selective state space models—an area that has so far been dominated by empirical studies. In terms of originality, it introduces a novel analytical framework for gated architectures trained with gradient descent, bridging ideas from feature-learning theory and state-space modeling, and extending prior analyses of Transformers to a fundamentally different recurrence-based mechanism. In terms of technical quality, the authors provide rigorous, non-asymptotic sample complexity and convergence bounds under well-defined data models, supported by clear mathematical reasoning and alignment with empirical trends. The paper could benefit from an improved exposition of the theoretical results. In general, the paper is well structured, with intuitive explanations preceding formal results. Regarding significance, the work fills a key theoretical gap by formalizing how Mamba’s gating acts as a feature selector that prioritizes class-relevant signals, thus deepening understanding of why such models generalize efficiently and under what data conditions they outperform attention-based architectures. While it is unclear whether this specific architecture will stand the test of time, this reviewer celebrates the introduction of rigorous analysis of architectural components.

**Weaknesses:**

While the paper makes a valuable theoretical contribution, several weaknesses limit its impact and clarity. First, the analysis is restricted to a highly simplified single-layer Mamba block and may not generalize to deeper or multi-head architectures used in practice; extending the framework to multi-layer dynamics or coupling between blocks would strengthen its relevance. While this is hard to do, authors could comment on whether these results extend to multi-layer settings. Second, the structured data assumptions (majority-voting and locality-structured) are somewhat idealized and may not capture the complexity of real-world sequences; providing evidence that these regimes approximate practical scenarios (e.g., via ablations on natural datasets) would improve the paper’s applicability. Third, the empirical validation is minimal—limited to synthetic data—and lacks comparisons to other theoretically analyzed architectures (e.g., gated linear attention), making it hard to assess the distinct benefits of Mamba’s gating mechanism. Fourth, while the paper claims to establish generalization guarantees, the proofs rely on strong assumptions such as balanced datasets, small token noise, and specific initialization schemes, which could limit robustness. Finally, the connection to prior theoretical work on Transformers and SSMs could be deepened by discussing how Mamba’s multiplicative gating fundamentally changes training dynamics beyond attention-style weighting. Strengthening these aspects would make the work more comprehensive and impactful.

**Questions:**

Please address weaknesses above.

---

> ### Author Response · Authors · 2025-11-21
> **Point-to-point response to Reviewer rcjx - part 1**
>
> # Point-to-point response to Reviewer rcjx
>
>
> >(W1) "The analysis is restricted to a highly simplified single-layer Mamba block and may not generalize to deeper or multi-head architectures used in practice. Authors could comment on whether these results extend to multi-layer settings."
>
> **Response to W1 (Simplified model and multi-layer settings).**
>
> Thank you for this question. We agree that understanding whether the theoretical insights extend beyond the simplified one-layer setting is essential. As detailed in [GR2](https://openreview.net/forum?id=hvpKqEYJjj&noteId=rm0XF7CM4I), we conducted new experiments on practical multi-layer, multi-head Mamba architectures. We believe these experiments directly address your concern, and the learned gating vectors and MLP weights in deeper Mamba models exhibit the same alignment behavior predicted by our theory. This provides empirical evidence that the theoretical mechanisms identified in our simplified model continue to hold in more realistic settings.
>
> Please see [GR2](https://openreview.net/forum?id=hvpKqEYJjj&noteId=rm0XF7CM4I) for our detailed response on the use of a simplified single-layer Mamba block and its connection to deeper, multi-head architectures.
>
> >(W2) "The structured data assumptions (majority-voting and locality-structured) are somewhat idealized and may not capture the complexity of real-world sequences."
>
> **Response to W2 (Structured data assumptions).**
> Thank you for raising this point. As we clarify in [GR3](https://openreview.net/forum?id=hvpKqEYJjj&noteId=bclldrQMeS), the majority voting and locality structured data models are standard abstractions in feature learning theory. Our analysis relies only on their structural properties, such as the way class-relevant features appear in the sequence, rather than on strict assumptions about the underlying data distribution.
>
>
> Please see [GR3](https://openreview.net/forum?id=hvpKqEYJjj&noteId=bclldrQMeS) for our detailed response on the practicality and interpretation of the majority-voting and locality-structured data models.
>
>
> >(W3.1) "The empirical validation is minimal."
>
> **Response to W3.1 (Empirical validation).**
> Thank you for the suggestion. We conducted additional experiments on practical multi-layer Mamba models, and the results, which are summarized in [GR1](https://openreview.net/forum?id=hvpKqEYJjj&noteId=8HXIkJ7rSq), consistently support the theoretical insights developed in our simplified setting.
> Please see [GR1](https://openreview.net/forum?id=hvpKqEYJjj&noteId=8HXIkJ7rSq) for more details.
>
>
> >(W3.2) "Lacks comparisons to other theoretically analyzed architectures (e.g., gated linear attention), making it hard to assess the distinct benefits of Mamba’s gating mechanism."
>
> **Response to W3.2 (Comparison with other architectures).**
> Compared with self-attention, which contains no gating mechanism, Mamba introduces an input-dependent gating function that modulates the recurrent update. This architectural difference leads to fundamentally different training dynamics, and our analysis is specifically aimed at characterizing the behavior of this gating mechanism.
> Regarding related work, we were unable to locate papers that directly study gated linear attention. The closest paper we identified is [R1], which investigates self-attention. We have included the comparison with this work in Remarks (See Remark 2 in Section 4.3.1).

---

> ### Author Response · Authors · 2025-11-21
> **Point-to-point response to Reviewer rcjx - part 2**
>
> >(W4) "While the paper claims to establish generalization guarantees, the proofs rely on strong assumptions such as balanced datasets, small token noise, and specific initialization schemes, which could limit robustness."
>
> **Response to W4 (Clarification on proof assumptions).**
>
> We believe this concern has arisen due to a misunderstanding regarding the current development and the theoretical analysis.
> We would like to clarify that such conditions are standard and appear widely in prior theoretical analyses of modern architectures (e.g., [R1, R2, R4, R6]), making our assumptions consistent with the prevailing practice in training-dynamics studies. These assumptions are introduced to keep the dynamics analytically tractable, especially in the presence of Mamba’s multiplicative gating, which introduces stronger token-interaction effects than in attention (please see the gradient expression in Appendix E.5).
> **Importantly, these conditions are required only for the proofs to hold. Empirically, we observe that the model remains stable and learns effectively even when these assumptions are relaxed.** For example, in our practical multi-layer Mamba experiments, we used a noise magnitude of $\tau = 0.5$, which is of the same order as the signal itself, and still observed clear convergence of both the gating vector and the MLP weights (please see [GR1](https://openreview.net/forum?id=hvpKqEYJjj&noteId=8HXIkJ7rSq)).
>
> >(W5) "The connection to prior theoretical work on Transformers and SSMs could be deepened by discussing how Mamba’s multiplicative gating fundamentally changes training dynamics beyond attention-style weighting."
> >
> **Response to W5 (Discussion of multiplicative gating).**
> Thank you for the suggestion. We agree that clarifying how multiplicative gating affects the training dynamics improves the exposition. In the revised manuscript, we expanded this discussion in the paragraph titled **"Connection and Difference with Transformer"** in **Section 2**, where we explicitly contrast Mamba’s input-dependent multiplicative gating with the additive weighting used in attention mechanisms.
>
> From a technical standpoint, the key challenge is that Mamba introduces input-dependent *multiplicative* interactions across tokens. These multiplicative recurrences accumulate over time and make the gradient dynamics explicitly sensitive to token order, which substantially increases the analytical complexity compared with attention-style models.  This changes how information propagates through the network and leads to training behavior that differs from attention-based models.
>
> [R1] Li, Hongkang, et al. “A Theoretical Understanding of Shallow Vision Transformers: Learning,
> Generalization, and Sample Complexity,” International Conference on Learning Representations
> (ICLR), 2023.
>
> [R2] Zhang, Shuai, et al. “Joint Edge-Model Sparse Learning is Provably Efficient for Graph Neural
> Networks,” International Conference on Learning Representations (ICLR), 2023.
>
> [R4] Li, Hongkang, et al. “How Do Nonlinear Transformers Learn and Generalize in In-Context
> Learning?” International Conference on Machine Learning (ICML), 2024.
>
> [R6] Brutzkus, Alon, et al. “An Optimization and Generalization Analysis for Max-Pooling Net-
> works,” Uncertainty in Artificial Intelligence (UAI), pp. 1650–1660, 2021.

---

> > ### Author Response · Authors · 2025-11-26
> > **Gentle Reminder on Rebuttal Discussion**
> >
> > Dear Reviewer rcjx,
> >
> > Thank you again for your detailed review. As the discussion deadline is approaching, we believe that our rebuttal addressed your main concerns by (i) adding full-model experiments with multi-layer, multi-head Mamba2 to assess whether the simplified analysis extends to practical architectures, (ii) clarifying the role and practicality of the structured data assumptions, and (iii) expanding the empirical section with new ablations, stronger baselines, and clearer connections to prior theoretical work.
> >
> > If there are any remaining concerns that you feel were not fully addressed, please let us know, we would be happy to clarify before the discussion concludes.
> >
> > Thank you for your time.

---

### Official Review · Reviewer_8FgG · 2025-11-01

**Soundness:** 3
**Presentation:** 3
**Contribution:** 3
**Rating:** 6
**Confidence:** 4

**Summary:**

The paper presents a theoretical analysis of learning and generalization for a simplified Mamba block (single-layer selective SSM with input-dependent gating followed by a 2-layer MLP) trained by gradient descent on structured sequence data. Two scenarios are studied: majority-voting and locality-structured sequences with token-level noise. The authors prove non-asymptotic sample complexity and convergence guarantees under width and noise conditions, show that the gating vector aligns with class-relevant features while suppressing irrelevant ones, and provide synthetic experiments supporting the theory. They also position Mamba’s dynamics relative to Transformers, arguing that Mamba can match the majority-voting setting and outperform attention baselines on locality-structured data due to selective recurrence.

**Strengths:**

- This paper contains a clear novel theoretical results. In particular, two formal and non-asymptotic bounds are presented for both sample complexity and iteration complexity in two scenarios (Theorems 1 and 2), with interpretable dependence on signal gap, locality separations, step size, sequence length, and noise.

- In addition, the authors provide a rigorous characterization that the gate prioritizes class-relevant features and suppresses confusing ones; connects to attention’s feature-selection role while highlighting differences arising from multiplicative recurrence.

- Synthetic experiments are closely tied to theory, including comparisons where Mamba outperforms Transformer/local attention for locality.

**Weaknesses:**

- My first main concern is the simplified assumptions (single layer, single head, simplified recurrence), which make the considered setting far from practical Mamba stacks (with depth, multi-head structure, residuals, normalization, and learned discretization). However, I understand that a simplified setting is commonly used in theoretical analysis, as the practical setting is usually too complicated to allow for a thorough analysis.

- My second concern is the strong assumptions required in Theorems 1 and 2: the width $m$ must be larger than or equal to the square of the dimension $d$, and the noise level must satisfy $\tau = O(1/d)$. These assumptions weaken the novelty of the theoretical results. A justification for these assumptions is needed.

- The experimental results are quite limited. The empirical comparison to Transformers and local attention is restricted to synthetic settings, with no controlled ablations for optimizer choice, learning rate scaling, or positional encoding. It is unclear whether attention could close the gap with appropriately tuned locality biases.

**Questions:**

- What would the theoretical results be if a more complex setting of the Mamba block were used? For example, what happens if a normalization layer is added or if the Mamba block is repeated?

- Are the assumptions in Theorems 1 and 2 regarding width and small noise required only for the proofs to work, or are they fundamentally unavoidable? What happens if these assumptions do not hold?

---

> ### Author Response · Authors · 2025-11-21
> **Point-to-point response to Reviewer 8FgG**
>
> # Point-to-point response to Reviewer 8FgG
> >(W1 \& Q1) "What would the theoretical results be if a more complex setting of the Mamba block were used? For example, what happens if a normalization layer is added or if the Mamba block is repeated? "
>
> **Response to W1 and Q1 (Effect of deeper and more complex Mamba blocks).**
> We thank the reviewer for the suggestion. All additional experiments addressing these points, including multi-layer Mamba evaluations and the requested ablations, are summarized in the general response section (please see [GR1](https://openreview.net/forum?id=hvpKqEYJjj&noteId=8HXIkJ7rSq)).
>
>
> >(W2 \& Q2) "Are the assumptions in Theorems 1 and 2 regarding width and small noise required only for the proofs to work, or are they fundamentally unavoidable? What happens if these assumptions do not hold? "
>
> **Response to W2 and Q2 (Justification of assumptions).**
> We appreciate your suggestions, and we agree that justifications for the assumptions can strengthen the paper. Here, we would like to clarify that these assumptions are standard within feature-learning frameworks and are not strictly required in the practical experiments. Since the theoretical proof analyzes worst-case scenarios, it is possible that noise and accumulated errors across tokens during Mamba’s sequential processing may enlarge the worst-case bound, which naturally leads to more conservative analytical requirements compared with the empirical setting. Nevertheless, our theoretical insights in Section 4.1 remain valid and generalizable, even when these conditions are only weakly satisfied in numerical experiments.
>
> **First, we emphasize the condition requiring the width \(m\) to be at least on the order of $d^2$ is not a strong assumption in the context of modern overparameterized neural networks.** Similar width requirements appear in several recent theoretical works on Transformers (e.g., [R1, R2, R4]), where overparameterization is essential for establishing non-asymptotic convergence and generalization guarantees.  Our condition is therefore consistent with the prevailing assumptions used in training-dynamics analyses of contemporary architectures. Moreover, numerical experiments also favor overparameterization, since practical models are generally large.
>
> **Second, the small-noise condition is introduced to make the analysis tractable.** Mamba’s gating mechanism creates multiplicative interactions across tokens, causing the gradient dynamics to couple features in a more complex way than in attention-based models (please see the gradient expression in Appendix E.5). These interactions can amplify noise as it propagates through the recurrent updates, and controlling this amplification during training requires restricting the noise magnitude. The small-noise assumption ensures that these interaction terms remain bounded so that we can derive meaningful generalization guarantees.
>
> **Importantly, this assumption is only required for the proofs to hold. Empirically, the model remains stable and successfully learns the structured data even when the noise level is substantially larger than assumed in the theory.** For example, in our practical multi-layer Mamba experiments, we used a noise magnitude of $\tau = 0.5$, which is of the same order as the signal itself, and still observed clear convergence of both the gating vector and the MLP weights (please see [GR1](https://openreview.net/forum?id=hvpKqEYJjj&noteId=8HXIkJ7rSq)).
>
> >(W3) "The empirical comparison to Transformers and local attention is restricted to synthetic settings, with no controlled ablations for optimizer choice, learning rate scaling, or positional encoding. It is unclear whether attention could close the gap with appropriately tuned locality biases. "
>
> **Response to W3 (Comparison with attention baselines).**
> Thank you for raising this point. As you suggested, we incorporated positional encodings and tuned the attention-based baselines to ensure they achieve their best performance under our setting. The updated results, along with full details, are provided in [GR1](https://openreview.net/forum?id=hvpKqEYJjj&noteId=8HXIkJ7rSq). Note that the insight that Mamba outperforms local attention and transformers on locality tasks remains unchanged.
>
> Moreover, we would like to clarify that the goal of this toy experiment is not to offer new numerical evidence that Mamba outperforms Transformers, particularly given that extensive empirical results already exist across a wide range of domains (e.g., [R8–R12]). Rather, the experiment is intended to illustrate that part of the performance gains observed with Mamba stem from its ability to effectively capture the locality structure inherent in the data. For this reason, we initially used the same hyperparameter settings across all models. However, we recognize that this choice may be confusing, and we have therefore conducted a more fine-grained comparison with appropriate tuning to address this concern.

---

> > ### Author Response · Authors · 2025-11-26
> > **Gentle Reminder on Rebuttal Discussion**
> >
> > Dear Reviewer 8FgG,
> >
> > Thank you again for your detailed review. As the discussion deadline is approaching, we believe that our rebuttal addressed your main concerns by (i) adding full-model experiments with multi-layer, multi-head Mamba2, (ii) providing justification for the width/noise assumptions, and (iii) running all requested ablations and stronger attention baselines.
> >
> > If there are any remaining concerns that you feel were not fully addressed, please let us know, we would be happy to clarify before the discussion concludes.
> >
> > Thank you for your time.

---

> > > ### Comment · Reviewer_8FgG · 2025-11-27
> > >
> > > I have read all comments from the other reviewers as well as the authors’ responses. My concerns have been satisfactorily addressed. Although the assumptions used in the theory are rather strong and thus limit the practical scope, I believe the analysis of generalization and learning dynamics in this paper is valuable and worth publishing. Therefore, I am raising my score to support acceptance of this paper.

---

> > > > ### Author Response · Authors · 2025-11-27
> > > >
> > > > Dear Reviewer 8FgG,
> > > >
> > > > Thank you very much for raising your score from 6 to 8. We sincerely appreciate your thoughtful reassessment of the paper and your recognition of the value of our theoretical analysis. We are grateful for the time and care you dedicated to reading the reviews, engaging with our rebuttal, and supporting the acceptance of our work.
> > > >
> > > > Thank you again for your time and consideration.

---

> ### Author Response · Authors · 2025-11-21
> **References**
>
> [R1] Li, Hongkang, et al. “A Theoretical Understanding of Shallow Vision Transformers: Learning,
> Generalization, and Sample Complexity,” International Conference on Learning Representations
> (ICLR), 2023.
>
> [R2] Zhang, Shuai, et al. “Joint Edge-Model Sparse Learning is Provably Efficient for Graph Neural
> Networks,” International Conference on Learning Representations (ICLR), 2023.
>
> [R4] Li, Hongkang, et al. “How Do Nonlinear Transformers Learn and Generalize in In-Context
> Learning?” International Conference on Machine Learning (ICML), 2024.
>
> [R8] Zhu, Lianghui, et al. “Vision Mamba: Efficient Visual Representation Learning with Bidirec-
> tional State Space Model,” Proceedings of the 41st International Conference on Machine Learning
> (ICML), PMLR 235, pp. 62429–62442, 2024.
>
> [R9] Wang, Chloe, et al. “Graph-Mamba: Towards Long-Range Graph Sequence Modeling with
> Selective State Spaces,” arXiv preprint arXiv:2402.00789, 2024.
>
> [R10] Liu, Yue, et al. “VMamba: Visual State Space Model,” Advances in Neural Information
> Processing Systems (NeurIPS), vol. 37, pp. 103031–103063, 2024.
>
> [R11] Behrouz, Ali, et al. “Graph Mamba: Towards Learning on Graphs with State Space Mod-
> els,” Proceedings of the 30th ACM SIGKDD Conference on Knowledge Discovery and Data Mining
> (KDD), pp. 119–130, 2024.
>
> [R12] Gu, Albert, et al. “Mamba: Linear-Time Sequence Modeling with Selective State Spaces,”
> arXiv preprint arXiv:2312.00752, 2023.

---

### Official Review · Reviewer_Sfca · 2025-11-04

**Soundness:** 3
**Presentation:** 3
**Contribution:** 2
**Rating:** 2
**Confidence:** 4

**Summary:**

This paper theoretically analyzes the training dynamics of a single Mamba layer wrapped with a two-layer MLP trained via gradient descent. Under assumptions like majority-voting or locally structured data, the authors establish generalization guarantees with improved sample complexity and convergence bounds derived from the characterization of the final solution. The analysis centers on the gating mechanism, a key innovation in modern recurrent models, and is supported by empirical evidence validating the theory.

**Strengths:**

W.1. Expanding the theoretical understanding of Mamba and its gating mechanism is important, as these components are popular in both transformers and modern RNNs.

W.2. The analysis is novel, non-trivial, and sheds light on the training dynamics and the role of critical components in modern architectures.

**Weaknesses:**

**W.1. Connection between theoretical settings and the real world:**

While the analysis of the simplified model is interesting, its connection to standard practices in the domain remains vague. I understand that providing theoretical proofs for the full model is challenging and that simplifications (such as using a single-layer model or specific datasets) are standard. I’m not expecting a full theoretical proof for the complete model, but I believe the authors should better link key aspects of the theoretical analysis to real models.

For example, they could run synthetic tasks inspired by the theoretical analysis on the full model and investigate whether the behavior in which MLP weights align with class-relevant features also appears in the complete models. Similarly, they could examine whether the cosine similarity between the gating vector and both class-relevant and class-irrelevant features is consistent with the theoretical (or single layers)  findings.

**W.2. Related work is missing.**

 Several studies analyzing the training dynamics, optimization issues, and generalization properties of Mamba are not mentioned. Examples include [1, 2, 3, 4, 5]. In this sense, some of the claims made in the paper (see ’To the best of our knowledge, this work presents the first theoretical generalization analysis of the Mamba architecture.’ and ‘To the best of our knowledge, we are the first to theoretically analyze the learning and generalization performance of the trained model with gradient descent (GD)’) may be overstated.

**W.3. The experimental analysis is limited:**

(i) Some information is missing or insufficiently detailed. For example, it is unclear which model was used (simplified, single-layer, with or without convolution, and whether parameters were frozen as in Eq. 15).

(ii) The experiments could be improved, for instance, one could empirically analyze multiple models across different data regimes (varying numbers of features, data distribution parameters, etc.). In addition, several architectural ablations could be performed to better understand the dynamics. For example, training the model without gating to see whether it struggles, as the theory predicts, could strengthen the connection between theory and practice.

W.4. Minor: The visibility of Figure 1 should be improved. For example, it would be better to use figures of the same size (the one on the far right is smaller), and the space between the first two figures is too narrow, causing the caption text to almost overlap.

**W.5. Generality can be improved:**

 Some parts of the theoretical analysis could be applied to other modern gated RNNs, such as Gated Linear Attention, Gate Delta-Net, RWKV, and RetNet. It would be good to explain which parts of the analysis are relevant to each architecture, and which components in those architectures improve or negatively impact the optimization dynamics or generalization.

___

**References**

[1] Generalization Error Analysis for Selective State-Space Models Through the Lens of Attention  . Honarpisheh. Nips 2024

[2] The Implicit Bias of Structured State Space Models Can Be Poisoned With Clean Labels. Slutzky∗ et al. NIps 2025.

[3] REVISITING ASSOCIATIVE RECALL IN MODERN RECURRENT MODEL. Okpekpe et al.

[4] Repeat after me transformers are better than state space models at copying. Jelassi et al.

**Questions:**

Q.1. I think $v_i$​ in Equation 6 is not introduced before it appears in the equation, and the authors should make this clearer.

Q.2. The visibility of Figure 1 should be improved. For example, it would be better to use figures of the same size (the one on the far right is smaller), and the space between the first two figures is too narrow, causing the caption text to almost overlap.

Q.3. Which aspects of the proof are not relevant for gated linear attention but are relevant only to Mamba layers?

Q.4. Are $W_B$​ and $W_C$​ trained, or is there a typo with the superscript 0 in Eq. 15 for those weights? It seems that the generalization guarantees are derived for a model where some parameters are frozen.

---

> ### Author Response · Authors · 2025-11-21
> **Point-to-point response to Reviewer Sfca - part 1**
>
> # Point-to-point response to Reviewer Sfca
>
>
> >(W1) "I believe the authors should better link key aspects of the theoretical analysis to real models. For example, they could run synthetic tasks inspired by the theoretical analysis on the full model and investigate whether the behavior is consistent with the theoretical (or single-layer) findings."
>
> **Response to W1 (Synthetic experiments on the full models).**
> We appreciate the reviewer's suggestions. As detailed in [GR2](https://openreview.net/forum?id=hvpKqEYJjj&noteId=rm0XF7CM4I), we conducted new experiments on practical multi-layer, multi-head Mamba architectures. We believe these experiments directly address your concern, and the learned gating vectors and MLP weights in deeper Mamba models exhibit the same alignment behavior predicted by our theory.
> Please see [GR2](https://openreview.net/forum?id=hvpKqEYJjj&noteId=rm0XF7CM4I) for our comprehensive response.
>
>
> >(W2) "Several studies analyzing the training dynamics, optimization issues, and generalization properties of Mamba are not mentioned. In this sense, some of the claims made in the paper may be overstated."
>
> **Response to W2 (Related work is missing).**
> We thank the reviewer for pointing out these relevant papers. We have now cited them in the appropriate places in the revised manuscript. **We would like to clarify that our intended claim is that, to the best of our knowledge, this work presents the first training-dynamics analysis of the Mamba architecture (with a gating mechanism) with generalization guarantees, rather than the first generalization analysis of Mamba overall.** We sincerely apologize for the oversight and any confusion caused. We also emphasize that none of the referenced works provides a training-dynamics analysis for Mamba with a gating mechanism.
>
> Specifically, [1] analyzes the generalization gap of Mamba using Rademacher complexity. The most relevant related work is [2], which studies the training dynamics of SSMs under a teacher–student setting and shows that gradient flow converges to a low-rank solution with improved generalization. There are several fundamental differences from our work. The most important distinction is that [2] does not consider the gating mechanism that is unique to Mamba. In [2],
> the matrices $A$, $B$, and $C$ are independent of the input and, without the introduction of $\Delta$, they do not function as a gating mechanism. [3] compares the empirical learning dynamics of SSMs and Transformers on two benchmark tasks (associative recall and copying), but it does not offer theoretical guarantees on convergence or generalization. However, we also would like to emphasize that  [2] shows Mamba’s success is highly sensitive to hyperparameter tuning, which further motivates fundamental questions about why and when Mamba succeeds, i.e., the focus of this work. [4] shows that Transformers outperform SSMs on the copying task, and its theoretical analysis centers on approximation power rather than training dynamics.
>
> >(W3) "The experiments could be improved, for instance, one could empirically analyze multiple models across different data regimes (varying numbers of features, data distribution parameters, etc.). In addition, several architectural ablations could be performed to better understand the dynamics."
>
> **Response to W3 (Ablations across data regimes and gating).**
>
> We appreciate the reviewer’s suggestion that broader empirical analysis would strengthen the paper, which was indeed underemphasized in the initial draft. After incorporating the requested experiments, we are pleased to report that the numerical results validate and align with our theoretical insights.
>
> **Specifically,   we conducted the recommended experiments, including varying the feature dimension and distribution parameters, as well as training models *without* the gating mechanism.** These additional evaluations consistently support our theoretical findings and show that ungated models struggle to learn the structured data, as predicted. Details of these additional experiments, along with the corresponding plots and observations, are included in the
> general response section of this rebuttal (Please see [GR1](https://openreview.net/forum?id=hvpKqEYJjj&noteId=8HXIkJ7rSq)).
>
> **In the end, our Experiment Settings are provided in the appendix with additional details (Section A.3).** In general, the original experiments were designed to directly verify the theoretical results presented in the paper and therefore followed the exact setup of our proposed theoretical setting. We additionally provide several ablations and the full-model experiments, which offer further support that our insights extend beyond the simplified setup.

---

> ### Author Response · Authors · 2025-11-21
> **Point-to-point response to Reviewer Sfca - part 2**
>
> >(W4 \& Q2) "Minor: The visibility of Figure 1 should be improved."
>
> **Response to W4 and Q2 (Minor).**
> Thank you for pointing out the visibility issues in Figure 1.
> We have revised the figure accordingly by adjusting the spacing and improving the caption placement to ensure readability.
>
> >(W5 \& Q3) "Some parts of the theoretical analysis could be applied to other modern gated RNNs. It would be good to explain which parts of the analysis are relevant to each architecture."
>
> **Response to W5 and Q3 (Generality to other gated architectures).**
> We appreciate the reviewer’s suggestion regarding improving the generality of our theoretical analysis. **However, we would like to clarify that training-dynamics analyses inherently require careful, case-by-case investigation.** Due to this model-specific nature, it is generally not feasible to establish a fully generic analysis framework that universally applies to different architectures, even when they appear similar at first glance. Nevertheless, we provide intuitive guidance on how our framework can be extended to different models.
> Furthermore, we would also like to gently note that this limitation should not be considered a major weakness of our work, as existing studies on training dynamics are likewise performed on a case-by-case basis rather than through a single unified theory.
>
> **First, we note that these models (RetNet, RWKV, GLA, and DeltaNet) can all be viewed as linear recurrent updates involving a weighted accumulation of key–value representations.** However, their decay mechanisms differ substantially, leading to distinct gating behaviors. RetNet and RWKV adopt a fixed decay factor, in contrast to Mamba, whose gating mechanism is input-dependent and therefore yields a richer and more adaptive recurrent behavior. GLA is the closest to our framework, but its gating is not applied multiplicatively in the recurrent state update. We suspect that the resulting dynamics may eventually evolve in a manner similar to Mamba, and that many of our insights would still transfer, although potentially with different trajectories or convergence rates. In contrast, DeltaNet implements a fundamentally different update rule based on the difference between actual and predicted values. We believe this induces training dynamics that prioritize token-wise change tracking rather than the “majority-voting” or "locality" behavior observed in Mamba-style recurrent updates. As a result, the model exhibits a stronger locality effect and can adaptively capture residual errors and local changes. Mamba captures local similarity, as we proved, whereas we suspect that DeltaNet may be better suited for capturing local dissimilarity. As a result, certain parts of our proof, particularly those concerning the evolution of the MLP weights, could potentially carry over to these models. However, the training dynamics of the gating vector, as well as its interaction with the MLP weights, would require substantial modification.
>
> **Second, prior theoretical studies on training dynamics (e.g., [R1, R2, R6]) likewise analyze one architecture at a time, because even small architectural differences lead to fundamentally different gradient structures and optimization behaviors.** In our case, several key steps of the proof rely on Mamba-specific components, most notably the input-dependent gating on the value pathway and the multiplicative recurrent update. These mechanisms create the non-linear token–interaction terms that appear in the gradient expressions used in the proofs of Lemmas in Appendix E. 5 and E.6. Since these components do not appear in the same form in other gated models, extending the analysis would require re-deriving the alignment and convergence arguments separately for each architecture. We view this as an interesting direction for future research, but one that lies beyond the scope of the current paper.
>
> >(Q1) "I think $v_i$ in Equation 6 is not introduced before it appears in the equation, and the authors should make this clearer."
>
> **Response to Q1 (Clarifying notation in Eq.6).**
> We apologize for the oversight.  The term $v_i$ corresponds to the output-layer weight, and we have clarified this notation in the revised version of the paper.
> Thank you for pointing this out.
>
> [R1] Li, Hongkang, et al. “A Theoretical Understanding of Shallow Vision Transformers: Learning,
> Generalization, and Sample Complexity,” International Conference on Learning Representations
> (ICLR), 2023.
>
> [R2] Zhang, Shuai, et al. “Joint Edge-Model Sparse Learning is Provably Efficient for Graph Neural
> Networks,” International Conference on Learning Representations (ICLR), 2023.
>
> [R6] Brutzkus, Alon, et al. “An Optimization and Generalization Analysis for Max-Pooling Net-
> works,” Uncertainty in Artificial Intelligence (UAI), pp. 1650–1660, 2021.

---

> ### Author Response · Authors · 2025-11-21
> **Point-to-point response to Reviewer Sfca - part 3**
>
> >(Q4) "Are $W_B$ and $W_C$ trained, or is there a typo with the superscript 0 in Eq. 15 for those weights? "
>
> **Response to Q4 (Why $W_B$ and $W_C$ are frozen).**
> In our analysis, we intentionally keep $W_B$ and $W_C$ frozen. **This design choice allows us to isolate and study the learning dynamics of the gating vector in the Mamba block without interference from additional trainable components.** We believe that keeping $W_B$ and $W_C$ fully trainable is neither necessary nor feasible, as doing so would significantly complicate the proofs without providing additional conceptual insight.
>
> More specifically, because the focus of this work is on analyzing training dynamics, the gating vector is a core component of the Mamba block, and our objective is to understand how it evolves under gradient descent.
> Freezing $W_B$ and $W_C$ makes it possible to cleanly examine this evolution.
>
> In addition, $W_B$ and $W_C$ play roles analogous to $W_K$ and $W_Q$ in attention mechanisms [R7]. Prior work [R1] on ViT imposes a related but stronger assumption that $W_B$ and $W_C$ are initialized close to a good feature-space mapping, and it is shown that they gradually converge to the corresponding feature subspaces during training, achieving a denoising effect while preserving the useful input features. Following this insight, since $W_B$ and $W_C$ do not fundamentally change the input architecture but primarily serve to denoise the representations, we keep the setup simple by fixing them as identity matrices. This allows all input features (including noise) to pass through, enabling us to isolate and analyze the role of the gating mechanism itself in shaping the learning process.
>
> Therefore, incorporating a joint analysis of $W_B$ and $W_C$ would not alter the main theoretical insights. We expect that they would also converge to the desired feature subspaces to provide a denoising effect, which would reinforce rather than change our theoretical findings. However, extending the proof to include their dynamics would require substantial additional technical work. Specifically, if $W_B$ and $W_C$ were trainable, the input projections would continually change during optimization, altering the token interactions at every step, and each intermediate bound would need to track this coupled drift in the feature mappings. Given that the current proof already spans a long length, such an extension is beyond the scope of this paper.
>
> [R1] Li, Hongkang, et al. “A Theoretical Understanding of Shallow Vision Transformers: Learning,
> Generalization, and Sample Complexity,” International Conference on Learning Representations
> (ICLR), 2023.
>
> [R7] Dao, Tri, et al. “Transformers are SSMs: Generalized Models and Efficient Algorithms Through
> Structured State Space Duality,” arXiv:2405.21060, 2024.

---

> > ### Comment · Reviewer_Sfca · 2025-11-25
> >
> > While some of my concerns were addressed, others were not. Specifically W.1, W.3, and the response to Q.4 did not fully convince me. In particular, the connection to real-world models remains vague. However, the additional experiments improve the paper, so I am increasing my score from 2 to 4.

---

> > > ### Author Response · Authors · 2025-11-26
> > > **Request for Clarification on W.1, W.3, and Q.4**
> > >
> > > Dear Reviewer Sfca,
> > >
> > > Thank you for increasing your score from 2 to 4.
> > > Regarding your comment that W.1, W.3, and Q.4 were “not fully convincing,” we would appreciate clarification, because we believe we have directly and completely addressed the specific points you raised in these sections.
> > >
> > > **For W.1**, your original suggestion was:
> > > >"For example, they could run synthetic tasks inspired by the theoretical analysis on the full model and investigate whether MLP weights align with class-relevant features… similarly examine cosine similarity of the gating vector…"
> > >
> > > We carried out exactly this:
> > >
> > > -   full-model multi-layer, multi-head Mamba2 experiments,
> > >
> > > -   measured MLP alignment and gating cosine similarities,
> > >
> > > -   and observed behavior consistent with the theoretical predictions.
> > >
> > > This was the precise experiment you proposed, so we would appreciate understanding which part of the connection still appears vague.
> > >
> > > **For W.3**, your points were:
> > > >(i) clarify which model was used;
> > >   (ii) analyze multiple models across data regimes and perform architectural ablations (e.g.,model without gating).
> > >
> > > We updated the appendix with full experimental details, ran all the ablations you listed (varying $d$, varying data distribution parameter​, gating vs. no-gating, etc.), and the results aligned with the theory (please see [GR1](https://openreview.net/forum?id=hvpKqEYJjj&noteId=8HXIkJ7rSq)). If there is a specific missing component beyond the items you suggested, we would be grateful if you could point it out.
> > >
> > > **For Q.4**, you asked whether $W_B$ and $W_C$ are trained, or whether the superscript "0" in Eq. 15 is a typo?
> > >
> > > We clarified that it is **not** a typo, the parameters are intentionally frozen during the proof, and the purpose of this simplification is to isolate the effect of the gating vector for analyzing training dynamics. We suspect that when the reviewer stated that this explanation is “not convincing,” they were referring to the motivation rather than the correctness of the assumption. We therefore provide additional details here. In particular, the key intuition is that during training, $W_B$ and $W_C$ converge toward their corresponding feature subspaces, effectively denoising irrelevant components while preserving class-relevant input features. As a result, freezing them does not materially change the theoretical insights we aim to convey, nor does it alter the conclusions drawn from our analysis. To enhance the rigor of our statement, we have attached more detailed technical explanations for your reference in [Figure A10] (https://imgur.com/a/xN9IkDC). Moreover, this type of simplification is standard. For example, to explain the majority-voting effects of transformers, prior work [R1] assumes that $W_Q$ and $W_K$ are initialized near a good feature-space mapping, which simplifies the proof. Consequently, the underlying intuition is well-established for Transformers and carries over naturally to Mamba. Including additional technical arguments would substantially increase the length without providing further conceptual insight.
> > >
> > > Given that all requested clarifications and experiments were carried out exactly as suggested, we would appreciate guidance on which remaining aspects you find unconvincing so that we can address them precisely during the discussion phase.
> > >
> > > Thank you again for the follow-up.

---

### Author Response · Authors · 2025-11-21
**GR1: general Response on "additional experimental results." [@Reviewer Sfca, @Reviewer 8FgG, @Reviewer rcjx, @Reviewer T7xH]**

We thank all reviewers for their careful reading of our paper and their constructive feedback. We have revised the manuscript accordingly. We are encouraged that all four reviewers highlighted the novelty and significance of our theoretical contributions, particularly the generalization guarantees, the characterization of Mamba’s gating mechanism, and our framework for understanding feature selection in selective state-space models. We also appreciate the reviewers' feedback that our empirical experiments are closely tied to the theory  and provide supporting evidence for our analytical findings.

Below, we first provide general responses (GRs) addressing the main concerns raised across reviews. We begin with a unified GR summarizing **all additional experiments** requested by the reviewers. Subsequent GRs then address concerns regarding **the simplified model and its connection to real-world Mamba architectures** and the **validity of our data assumptions**.

Point-by-point responses are provided after each reviewer’s comment.

# GR1: general Response on "additional experimental results." [@Reviewer Sfca, @Reviewer 8FgG, @Reviewer rcjx, @Reviewer T7xH]

We thank all reviewers for emphasizing the importance of strengthening the empirical connection between our theoretical analysis and practical Mamba architectures. In response, we conducted additional experiments using the multi-layer, multi-head Mamba model from [R7] trained on synthetic datasets that follow the same structured data regimes as in our theory. We emphasize that these experiments are conducted to further support the theoretical insights developed in this work and do not alter the main contributions of the study.

**First, we conducted experiments using the Mamba2 block, which includes residual connections and RMSNorm.** Our tests included a stacked multi-layer Mamba block ($2$ and $5$ blocks), both configured with $4$ heads. Overall, the empirical results support our theoretical findings: (i) the cosine similarity between the learned gating vectors and both class-relevant and class-irrelevant features matches the predicted behavior from our theory [Figure A1](https://imgur.com/a/MrYJSR6), and (ii) the MLP weights in practical Mamba blocks consistently align with class-relevant feature directions [Figure A2](https://imgur.com/u4NmIXI), exactly as our analysis suggests.
Additional experiment results on $5$ blocks Mamba model are summarized in the following table.


**Table 1: Cosine similarity alignment in the 5-layer Mamba model.**


| |Class-relevant |Class-irrelevant |
|----------------|-------------------------------|-----------------------------|
|Gating vector |0.53 |0.00 |
|MLP weights |0.73|0.00|


**Next, we evaluated the effect of the gating mechanism by comparing models trained with and without gating on both structured data regimes.** For the majority-voting data, the gated model consistently outperforms the ungated variant [Figure A3](https://imgur.com/a/an2yH9c). For the locality-structured data, gating becomes essential, since the ungated model fails to learn the task while the gated model converges reliably [Figure A4](https://imgur.com/a/5o3GVpq).

**Thirdly, following the suggestion of Reviewer@Sfca, we conducted two additional ablations.** First, we varied the feature dimension $d \in \{32, 64, 128\}$ and observed that the results are consistent with those reported in the paper [Figure A5](https://imgur.com/a/POgId2e). Second, we varied the data distribution parameter $\alpha_c$, the fraction of confusion features in the majority-voting data, across three settings, and the empirical results again align closely with our theoretical predictions [Figure A6](https://imgur.com/a/i4Am2nk).

**Finally, to address the concern raised by Reviewer@8FgG regarding the comparison with attention-based baselines, we extended our empirical study to ensure that the attention models operate under their strongest possible configuration.** In the original experimental setup, all architectures, including Mamba, Transformer, and local-attention models, were trained under identical optimizer, learning-rate, and hyperparameter settings. Following the reviewer’s suggestion, we further incorporated positional encodings and tuned the relevant hyperparameters for the Transformer and local-attention baselines to give them the best possible performance. The updated results can be seen in [Figure A7](https://imgur.com/a/3I8Pmy5). The findings remain consistent with our earlier figure, and the Mamba model continues to perform favorably when the class-relevant features are more locally concentrated.

[R7] Dao, Tri, et al. “Transformers are SSMs: Generalized Models and Efficient Algorithms Through
Structured State Space Duality,” arXiv:2405.21060, 2024.

---

### Author Response · Authors · 2025-11-21
**GR2: general Response to "the simplified model and the connection between our theoretical setting and real-world Mamba models.'' [@Reviewer Sfca, @Reviewer 8FgG, @Reviewer rcjx, @Reviewer T7xH]**

# GR2: general Response to "the simplified model and the connection between our theoretical setting and real-world Mamba models.'' [@Reviewer Sfca, @Reviewer 8FgG, @Reviewer rcjx, @Reviewer T7xH]

We sincerely thank the reviewers for their thoughtful comments and suggestions. We appreciate that all reviewers recognized the difficulty of providing a full theoretical analysis of the complete Mamba architecture, which is precisely why simplified models are commonly used in theoretical studies. We also agree that experiments on the full model would strengthen the paper and offer additional support for our theoretical insights. Following your suggestions, we have carried out additional experiments on the full model and observed behavior fully consistent with our theoretical predictions, further reinforcing our findings.


**First, we conducted additional experiments on multi-layer, multi-head Mamba models, and the results were fully consistent with our theoretical findings.** As suggested by Reviewer@Sfca, we evaluated a multi-layer, multi-head Mamba model under our synthetic data setting. In these experiments, (i) the learned gating vectors aligned with class-relevant features, and (ii) the hidden-layer (MLP) weights likewise converged toward the class-relevant feature directions. These findings further demonstrate that the theoretical mechanisms identified in our simplified analysis extend to practical Mamba architectures. Please see [GR1](https://openreview.net/forum?id=hvpKqEYJjj&noteId=8HXIkJ7rSq) for the detailed results.


**Second, we clarify that working with a simplified shallow neural network model is standard practice in the theoretical analysis
of generalization and convergence.** Most existing results in this area rely on simplified architectures, precisely because full practical models are too complex to admit rigorous, non-asymptotic analysis. This does not prevent these works from offering many interesting and insightful conclusions. Please refer to some recent works [R1, R2, R3, R4, R5] for examples. As highlighted in [R1] and [R2], even in these simplified settings, the optimization landscape remains highly non-convex, and establishing convergence and generalization guarantees is still technically challenging and non-trivial.


**Finally, some degree of simplification is not only common but necessary in the theoretical study of modern neural networks.** The training dynamics analysis of the complete Mamba architecture is currently infeasible due to its substantial architectural complexity, including multi-head gating, multiplicative recurrence, discretization, residual pathways, and normalization layers. As noted by Reviewer@Sfca and Reviewer@8FgG, removing these simplifications is practically impossible at present. Similar abstractions are adopted in prior theoretical works on Transformers, CNNs, and GNNs to make the analysis tractable while preserving the essential mechanisms under investigation.

[R1] Li, Hongkang, et al. “A Theoretical Understanding of Shallow Vision Transformers: Learning,
Generalization, and Sample Complexity,” International Conference on Learning Representations
(ICLR), 2023.

[R2] Zhang, Shuai, et al. “Joint Edge-Model Sparse Learning is Provably Efficient for Graph Neural
Networks,” International Conference on Learning Representations (ICLR), 2023.

[R3] Shi, Zhenmei, et al. “A Theoretical Analysis on Feature Learning in Neural Networks: Emer-
gence from Inputs and Advantage over Fixed Features,” International Conference on Learning
Representations (ICLR), 2022.

[R4] Li, Hongkang, et al. “How Do Nonlinear Transformers Learn and Generalize in In-Context
Learning?” International Conference on Machine Learning (ICML), 2024.

[R5] Allen-Zhu, Zeyuan, et al. “A Convergence Theory for Deep Learning via Over-Parameterization” International Conference on Machine Learning (ICML), pp. 242–252, 2019.

---

### Author Response · Authors · 2025-11-21
**GR3: general Response to "validity of our data assumptions'' [@Reviewer rcjx, @Reviewer T7xH]**

# GR3: general Response to "validity of our data assumptions'' [@Reviewer rcjx, @Reviewer T7xH]


We thank the reviewers for this important question regarding the practicality of our data models.

**First, we would like to clarify a misunderstanding from Reviewer@T7xH.** The paper does not assume that the data itself follows a Gaussian distribution since our framework is based on a structured data model. Only the initialized weights follow a Gaussian distribution, which is standard and natural in practice. In general, the structured data model is widely used in the theoretical analysis of deep learning and has been shown to capture key properties of real data, as elaborated below.

**Second, structured data assumptions are widely used in theoretical analyses of modern architectures.** These data models can be viewed as abstractions of real data that capture the essence of the studied problem and help formalize important hypotheses observed in practice. The majority-voting model in particular is a standard assumption in recent theoretical studies (e.g., [R1, R2, R3]), including analyses of Transformers and attention mechanisms.


**Third, we believe our majority-voting and locality-structured data models introduced in this work capture the structure of many real-world tasks.** On the one hand, the majority-voting data model reflects a common practical pattern in which the label is determined by the aggregate contribution of multiple discriminative sources. For example, in image classification, the class label often reflects evidence from several foreground patches (class-relevant tokens), in contrast to background patches that may contain confusing or irrelevant patterns. On the other hand, the locality-structured data model corresponds to tasks where semantic meaning is concentrated in spatially or temporally localized clusters, while background features are more dispersed. This structure is characteristic of vision tasks such as object detection, localization, and image captioning, as well as audio, speech, and genomics, where decisive information is confined to short, contiguous regions that strongly correlate with the label. **We would like to clarify further that this content was originally presented in Section 4.2 of the paper and is explicitly highlighted in the revised version.**

[R1] Li, Hongkang, et al. “A Theoretical Understanding of Shallow Vision Transformers: Learning,
Generalization, and Sample Complexity,” International Conference on Learning Representations
(ICLR), 2023.

[R2] Zhang, Shuai, et al. “Joint Edge-Model Sparse Learning is Provably Efficient for Graph Neural
Networks,” International Conference on Learning Representations (ICLR), 2023.

[R3] Shi, Zhenmei, et al. “A Theoretical Analysis on Feature Learning in Neural Networks: Emer-
gence from Inputs and Advantage over Fixed Features,” International Conference on Learning
Representations (ICLR), 2022.

---

### Comment · Area_Chair_rkBk · 2025-11-23
**Next Steps Following Authors’ Rebuttal: Review Rebuttal and Participate in Discussion**

Dear Reviewers,

Thank you very much for your thoughtful evaluations of this paper.

Now that the authors have submitted their rebuttal, I kindly ask you to take the following steps (if you have not done so already):

- Read the other reviews as well as the authors’ response.
- Consider whether the rebuttal and additional comments affect your assessment of the paper.
- Engage in interactive discussion with the authors **before November 25**, encouraging a dynamic exchange rather than a one-sided rebuttal.

The current reviews for this paper are mixed. Your contributions at this stage are essential for forming a well-informed final decision. I therefore ask that you reassess your views in light of the authors’ responses and the broader discussion among reviewers.

I am happy to join and support the discussions between you and the authors. Please feel free to share your thoughts and participate actively in the discussion.

Thank you once again for your service to ICLR 2026.

Best regards,

 AC

---

### Meta-Review · Area_Chair_8hL9 · 2026-01-07

**Summary:**

The paper presents a theoretical analysis of the training dynamics and generalization properties of a simplified Mamba block under two structured synthetic data regimes.

The original review scores were 6/6/6/2. Reviewers generally recognized the novelty and theoretical significance of the work. The main concerns focused on the gap between the simplified theoretical setting (e.g., single-layer, single-head architecture and idealized data models) and real-world Mamba architectures and practical applications, as well as missing related work and limited empirical validation. The most fundamental concernl, raised in different forms by all reviewers and shared by the AC, was the strength of the assumptions and the limited connection between the theoretical results and practical models and settings.

In the rebuttal, the authors provided substantial additional discussion and new experimental results, including experiments on more complex multi-layer, multi-head Mamba blocks and the requested ablation studies, to strengthen the connection between theory and practice.

As reflected in the responses, Reviewer Sfca increased their score from 2 to 4, and Reviewer 8FgG explicitly stated support for acceptance, raising their score from 6. While some concerns regarding the remaining gap between theory and realistic models and the necessity of certain modeling simplifications persist, these assumptions are standard in theoretical analyses and are required to make the problem tractable.

Overall, the theoretical results are considered valuable by the reviewers.
Based on the improved evaluations and the strength of the theoretical contribution, the AC recommends acceptance and encourages the authors to carefully incorporate all clarifications, additional experiments, and discussions from the rebuttal and responses into the final version of the paper.

**Reviewer Concerns:**

The most significant concern raised by all reviewers is the limited connection between the simplified theoretical setting and practical applications. This concern manifests in several aspects, including the strong simplifications of the Mamba architecture and data structure, as well as the strong assumptions required by the theoretical analysis. Additional concerns relate to the largely synthetic nature of the empirical validation and the absence of direct comparisons with other gated architectures.

The rebuttal addresses many of these points through additional discussion and new experiments, although some concerns remain partially outstanding. In particular, the gap between the theoretical assumptions and the simplified model architecture versus real-world Mamba models remains the most substantial issue (Sfca, 8FgG, and T7xH[-on data structure and modeling assumptions]). While such simplifications and assumptions are common and often unavoidable in theoretical work, and while the added experiments on more complex architectures strengthen the empirical support, they do not fully resolve the fundamental limitations inherent in the theoretical setting.

The AC therefore suggests that the authors carefully refine their claims and presentation in the final version to avoid potential overstatements.

**Reviewer Scores:**

The original scores are 6/6/6/2. The rebuttal may change the scores to 8/6/6/4.

- Reviewer Sfca. The reviewer stated to improve the score from 2 to 4 in responses. With fuller discussion, the most realistic outcome is staying at 4 because they explicitly indicated remaining doubt about the connection between the theory and real-world practice and the frozen-parameter justification (Q4), and those are not easy to “fully” resolve within rebuttal constraints.

- Reviewer 8FgG. Although the reviewer thinks the assumptions in the theory are rather strong and thus limit the practical scope, they acknowledged the value of analysis. As reflected in the responses, they would like to raise the score from 6 to probably 8 to support acceptance of this paper.

- Reviewer rcjx. Likely maintain the original score 6. rcjx’s critique is about practical relevance, idealized data assumptions, and minimal empirical validation beyond synthetic. The rebuttal provided more experiments, but it still stays in synthetic regimes, so I’d predict no change is likely.

- Reviewer T7xH. Likely maintain the original score 6 or increase the score if satisfied by the provided experiments. Their major issues are about  the scope of the model, the binary classification, and Gaussian assumption, plus limitations. The authors directly implemented multi-class extension + experiment and clarified the misunderstanding, while some criticisms on the fundamental setup are hard to address.

---

### Decision · Program_Chairs · 2026-01-26

Accept (Poster)